# Tumour-retained activated CCR7$^+$ dendritic cells are heterogeneous and regulate local anti-tumour cytolytic activity

Colin Y. C. Lee [1,2], Bethany C. Kennedy [3], Nathan Richoz[1], Isaac Dean [3], Zewen K. Tuong[1,2], Fabrina Gaspal[3], Zhi Li [3], Claire Willis[3], Tetsuo Hasegawa[1], Sarah K. Whiteside[4], David A. Posner[1], Gianluca Carlesso[5], Scott A. Hammond [5], Simon J. Dovedi[6], Rahul Roychoudhuri [4], David R. Withers [3] ✉ & Menna R. Clatworthy [1,2] ✉

Tumour dendritic cells (DCs) internalise antigen and upregulate CCR7, which directs their migration to tumour-draining lymph nodes (dLN). CCR7 expression is coupled to an activation programme enriched in regulatory molecule expression, including PD-L1. However, the spatio-temporal dynamics of CCR7$^+$ DCs in anti-tumour immune responses remain unclear. Here, we use photo-convertible mice to precisely track DC migration. We report that CCR7$^+$ DCs are the dominant DC population that migrate to the dLN, but a subset remains tumour-resident despite CCR7 expression. These tumour-retained CCR7$^+$ DCs are phenotypically and transcriptionally distinct from their dLN counterparts and heterogeneous. Moreover, they progressively downregulate the expression of antigen presentation and pro-inflammatory transcripts with more prolonged tumour dwell-time. Tumour-residing CCR7$^+$ DCs co-localise with PD-1$^+$CD8$^+$ T cells in human and murine solid tumours, and following anti-PD-L1 treatment, upregulate stimulatory molecules including OX40L, thereby augmenting anti-tumour cytolytic activity. Altogether, these data uncover previously unappreciated heterogeneity in CCR7$^+$ DCs that may underpin a variable capacity to support intratumoural cytotoxic T cells.

Dendritic cells (DCs) capture tumour antigens and upregulate the chemokine receptor CCR7, which directs their migration to secondary lymphoid organs where they present antigens to T cells[1-3]. Two major conventional DC (cDC) subsets have been identified in tumours; cDC1 that specialise in cross-presenting tumour antigens to CD8$^+$ T cells and are associated with improved survival[4], and cDC2 that present exogenous antigens to CD4$^+$ T cells and have variable associations with cancer prognosis and treatment responses[1]. The recent application of single-cell technologies to tissue samples has enabled the distinction of an activated DC state that may arise from both cDC1 and cDC2 but demonstrates a conserved phenotype and transcriptional programme characterised by the expression of LAMP3, genes consistent with a maturation and migration (CCR7, FSCN1, CD40), and immunoregulatory molecules including PD-L1 and PD-L2[5]. This DC population has been variously labelled as "migratory DCs", "mature DCs enriched in immunoregulatory molecules" (mRegDCs), as well as "LAMP3$^+$ DCs",

[1]Molecular Immunity Unit, Department of Medicine, Medical Research Council Laboratory of Molecular Biology, University of Cambridge, Cambridge, UK. [2]Cellular Genetics, Wellcome Sanger Institute, Wellcome Genome Campus, Hinxton, Cambridge, UK. [3]Institute of Immunology and Immunotherapy, College of Medical and Dental Sciences, University of Birmingham, Birmingham, UK. [4]Department of Pathology, University of Cambridge, Cambridge, UK. [5]Early Oncology R&D, AstraZeneca, Gaithersburg, MD, USA. [6]Early Oncology R&D, AstraZeneca, Cambridge, UK. ✉e-mail: d.withers@bham.ac.uk; mrc38@medschl.cam.ac.uk

"mature DCs", or "activated DCs"[6–8]. Regardless of nomenclature, single-cell atlasing efforts have identified activated CCR7+ DCs in multiple human cancer types[9–11], and the acquisition of this maturation programme appears dependent on the uptake of tumour antigens[5].

Intriguingly, despite the conserved expression of co-inhibitory molecules such as PD-L1 in CCR7+ DCs, the expression of the DC activation-associated marker LAMP3 is associated with improved prognosis in breast, lung cancer and metastatic melanoma[12–15]. However, the precise contribution of CCR7+ DCs to anti-tumour responses, and whether they have a positive or negative effect on disease outcomes remains unclear. Certainly, CCR7+ DCs are likely to be key targets of immune checkpoint blockade (ICB) by virtue of their high expression of co-inhibitory molecules. Consistent with this, murine studies have demonstrated the importance of PD-L1 expression by DCs in regulating anti-tumour T cell activation and response to ICB in subcutaneous tumours[16,17].

Activated DCs express high levels of CCR7, enabling trafficking from the tumour to draining lymph nodes (dLN)[2,3,5]. Indeed, LAMP3+ DCs have been identified in tumour-dLNs, where they may activate tumour antigen-specific T cells[18]. However, LAMP3-expressing DCs have also been described within tertiary lymphoid structures (TLS) in tumours[12,15], suggesting that some activated DCs may be retained in certain niches for prolonged periods. Thus, the precise temporal dynamic behaviour of CCR7+ DCs, the extent to which they act within the tumour versus dLN, and how these dynamics and site-specific cellular interactions are influenced by ICB remain to be clarified. The concept that prolonged residence within tumours might influence CCR7+ DC fate and function is worthy of consideration given the known effects of the tumour microenvironment (TME) on other immune cell populations. For example, CD8+ T cells transition to a so-called 'exhausted' state with prolonged dwell-time in the tumour[19], with three defining characteristics; reduced effector function, sustained expression of inhibitory molecules such as PD-1, and a transcriptional state distinct from that of functional effector cells[20]. These features have also been observed in other tumour-resident immune cells such as natural killer (NK) cells[21,22].

Here, we use a photo-tracking mouse model, combined with single-cell RNA sequencing (scRNA-seq), confocal imaging and spatial transcriptomics in mouse and human tissue samples, to interrogate the spatio-temporal dynamics of CCR7+ DCs, their roles within the tumour, and the effects of ICB. We found that CCR7+ DCs were heterogeneous, influenced by duration of tumour residence, location (tumour versus dLN), and anti-PDL1 treatment. Strikingly, tumour-retained CCR7+ DCs were phenotypically distinct from those that migrated to the dLN and took on an increasingly "exhausted" transcriptional profile with more prolonged tumour dwell-time. These intratumoural CCR7+ DCs co-localised with cytotoxic CD8+ T cells, and anti-PD-L1 treatment enhanced their expression of several important T cell-stimulatory molecules. CCR7+ DC-CD8+ T cell engagement and their augmentation by ICB was confirmed across a range of human cancers. Altogether, these data provide further insight to tumour CCR7+ DC dynamics and their role in cancer immunotherapy.

## Results

### CCR7+ DC signatures are associated with improved survival in human cancers

The presence of mature LAMP3+ DCs within lymphoid follicles has been associated with improved prognosis in non-small cell lung cancer (NSCLC)[12], but the effect of tumour CCR7+ DCs on prognosis in human cancers more broadly has not been examined. To address this, we analysed the transcriptomes of 4,045 human solid tumours from the cancer genome atlas (TCGA)[23]. Enrichment of a CCR7+ DC signature[5] was associated with improved survival not only in lung cancer, but also in cutaneous melanoma, breast, and colorectal cancer (CRC, Supplementary Fig. 1a), all of which harbour CCR7+CD274+PDCD1LG2+ DCs

(Supplementary Fig. 1b–g)[24–26]. Further analysis of 1,853 human breast tumours from METABRIC[27] revealed cancer subtype-specific associations with survival (Supplementary Fig. 1h). These data suggest that CCR7+ DCs contribute to anti-tumour responses across a range of human cancers.

### CCR7+ DCs are transcriptionally heterogeneous, and some remain within the tumour despite CCR7 expression

To investigate the mechanisms by which CCR7+ DCs promote anti-tumour immunity, we sought to characterise their spatio-temporal dynamics. The DC activation programme is thought to be driven by acquisition of tumour antigen[5], and we asked whether establishment of this programme inevitably leads to the trafficking of CCR7+ DCs to the dLN, or whether some cells remain in the tumour. To address this, we established multiple syngeneic subcutaneous colorectal tumours (MC38, MC38-Ova, CT26) in a photoconvertible Kaede transgenic model that enables site-specific temporal labelling of cells within the tumour[28]. Tumours were transcutaneously photoconverted on day 13, converting all infiltrated cells in the tumour only from the default green fluorescence (Kaede-green) to a red fluorescent profile (Kaede-red)[19]. Tumours were harvested 24–72 h following photoconversion, enabling the distinction of newly-infiltrating Kaede-green cells and Kaede-red cells retained in the tumour from the point of photo-labelling (Fig. 1a). To address the effect of ICB on CCR7+ DCs, we administered anti-PD-L1 antibodies in this model (Supplementary Fig. 2a), selecting this target because CCR7+ DCs in CRC showed the highest expression of PD-L1 among immune cells (Supplementary Fig. 1c).

scRNA-seq of FACS-isolated Kaede-green+ or Kaede-red+ immune cells (Kaede+CD45+Ter119-) 48 h after photoconversion (Supplementary Fig. 2b) generated 80,556 single cell transcriptomes following rigorous quality control, including 32,191 myeloid cells (Fig. 1b, Supplementary Fig. 2c). Unbiased clustering of DCs revealed 8 distinct clusters, including cycling DCs, cDC1, cDC2s, and activated DCs (Ccr7_DC), assigned based on high expression of canonical marker genes, including Ccr7, Cd274 and Pdcd1lg2, and similarity to published transcriptomes (Fig. 1c, d, Supplementary Fig. 2d). Indeed, CCR7+ DCs expressed higher levels of surface PD-L1 than other immune cells (Supplementary Fig. 2e, f).

Activated DCs (defined in this study as surface CCR7+PD-L2+, hereafter just "CCR7+ DC") were a prominent tumour DC population, and this was confirmed by flow cytometry (Fig. 1c, Supplementary Fig. 2e, g). Surprisingly, tumour Ccr7_DCs were predominantly Kaede-red indicating that these cells have resided in the tumour for over 48 h (Fig. 1e, Supplementary Fig. 2h). In contrast, cDC1 and cDC2 were mostly Kaede-green, consistent with the conclusion that these newly infiltrating cDCs differentiate into Ccr7_DCs following uptake of tumour antigen[5]. The prevalence of tumour-retained Kaede-red CCR7+ DCs was validated over a time-course in all tumour models, by flow cytometry, where CCR7+ DCs were the major Kaede-red DC cell-type up to 72 h post-photoconversion (Fig. 1f, g, Supplementary Fig. 2i). Using immunofluorescence (IF) microscopy, we further confirmed the presence of Kaede-red MHC-II+CCR7+ DCs up to 72 h post-photoconversion (Fig. 1h, Supplementary Fig. 2j–l).

To explore the relationship between tumour DC subsets, we performed pseudotime analysis rooted in the Kaede-green dominant cluster (cDC2.1). This revealed a trajectory terminating in Ccr7_DC.2/3 that transitioned through an intermediate Ccr7_DC.1 state (Fig. 1i, Supplementary Fig. 3a), consistent with the Kaede-green/red ratio of clusters that marks tumour dwell in real-time. RNA velocity analysis confirmed the cell-state transition from cDC, through Ccr7_DC.1, to Ccr7_DC.2 or Ccr7_DC.3, which were the endpoints of the velocity trajectory (Fig. 1j, Supplementary Fig. 3b).

To further demonstrate that tumour CCR7+ DCs arise from cDC precursors, we examined Kaede-green DCs 5 h post-photoconversion, which should harbour few CCR7+ DCs because newly-entered DCs

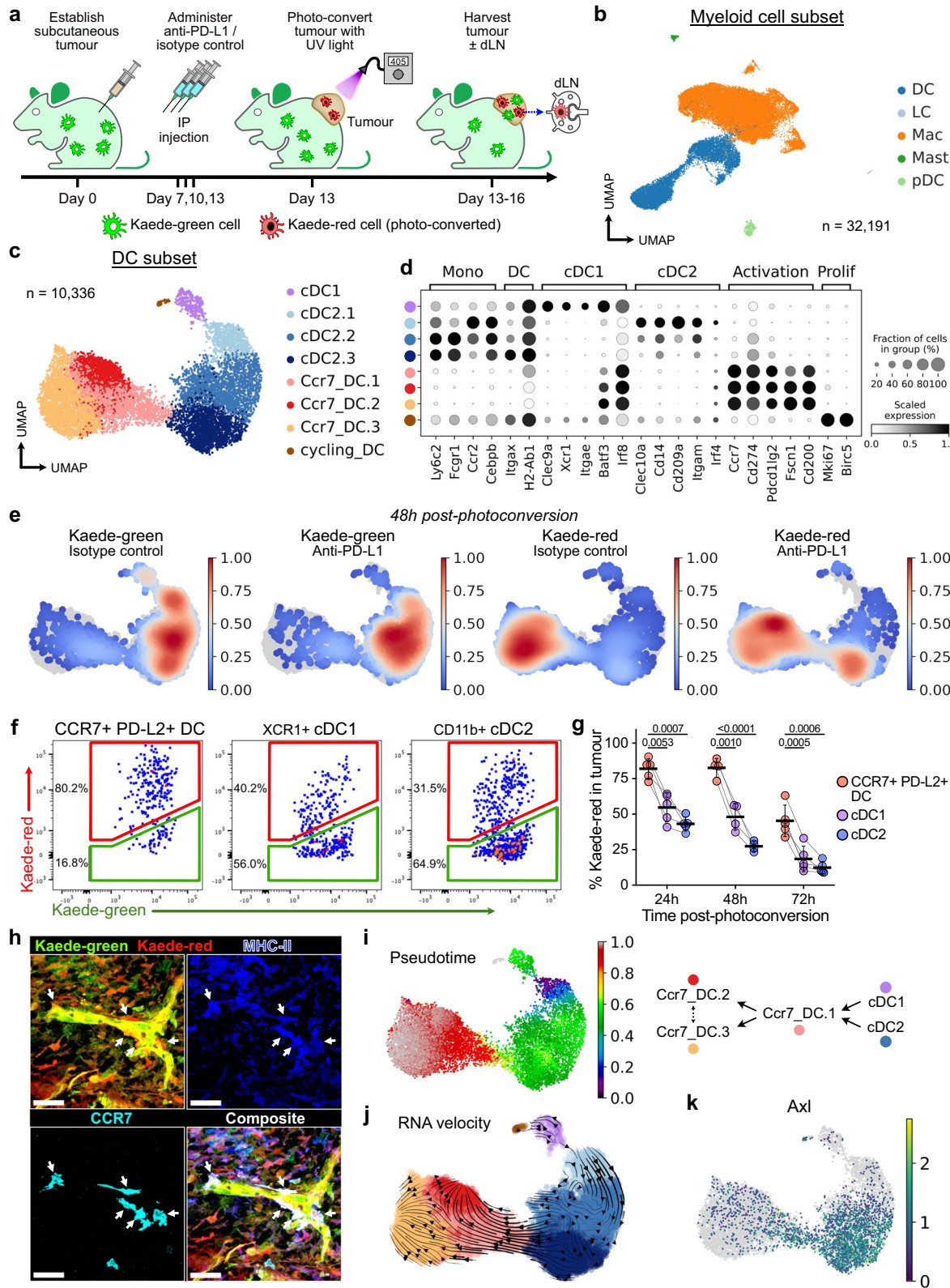

would have little time to acquire the CCR7⁺ activation programme. Indeed, at 5 h post-photoconversion the Kaede-green CCR7⁺ DC:cDC ratio was only 1/100, but this increased to 1/5 72 h post-photoconversion (Supplementary Fig. 3c). Hence, the CCR7⁺ DC state emerges over time in the tumour and only after the influx of cDCs. cDC1 or cDC2-defining transcripts were not conserved upon acquisition of the DC activation programme, for example, *Clec9a* or *Clec10a* expression was lost but *Irf8* and *Batf3* are upregulated ubiquitously (Fig. 1d).

However, maintained cDC subset-specific surface marker expression indicated a mixed ontogeny of CCR7⁺ DCs, with expression of XCR1 (cDC1 origin) and CD11b (cDC2 origin) detectable, and notably, XCR1⁺CCR7⁺ DCs were higher among the Kaede-red fraction (Supplementary Fig. 3d). These data support the conclusion that both cDC1 and cDC2 contribute to tumour-retained CCR7⁺ DCs.

Kaede-red cDC1 and cDC2 showed higher expression of migratory transcripts and *Ciita*, a master regulator of antigen presentation[29], than

**Fig. 1 | Landscape and temporal dynamics of DCs in murine tumours.**
**a** Experiment design. Tumours from Kaede transgenic mice were harvested 5, 24, 48, or 72 h after photoconversion. **b** UMAP of myeloid cells from scRNA-seq of FACS-sorted CD45[+]TER119[-] Kaede-green[+]/Kaede-red[+] cells 48 h after photoconversion of subcutaneous MC38-Ova tumours. DC, dendritic cell; LC, Langerhans cell; Mac, macrophage; Mast, mast cell; pDC, plasmacytoid DC. **c** UMAP of DCs from (**b**) and canonical marker gene expression in DC clusters (**d**). **e** Kernel density embedding of DCs by Kaede fluorescence and treatment group. **f, g** Flow cytometry of Kaede fluorescence in DCs subsets/states from MC38-Ova tumours: **f** Representative plots 48 h post-photoconversion, and (**g**) Time-course and quantification. **h** Confocal microscopy of MC38 tumours 72 h post-photoconversion. Arrows point to tumour-residing CCR7[+] DCs (Kaede-red[+]CCR7[+]MHC-II[+], dendritic morphology). Scale bar, 40 μm. **i** Tumour DCs ordered by pseudotime, rooted in the cluster with highest proportion of Kaede-green DCs, and proposed maturation trajectory. **j** RNA velocity trajectory in tumour DCs. **k** Expression of *Axl*. Paired two-sided student's t-test was used, data are shown as means ± s.d., and points represent independent mice (**g**). The results shown in (**g**) are from one experiment (24 h/72 h *n* = 5; 48 h *n* = 4 animals), representative of three independent experiments; and (**h**) are representative of three independent experiments (*n* = 7 animals).

Kaede-green counterparts, further suggesting that cDCs mature with duration in the TME (Supplementary Fig. 3e). Principal component (PC) analysis showed that Kaede fluorescence accounted for most of the transcriptional variance in cDC2s (PC1, Supplementary Fig. 3f). Genes driving PC1 were enriched in pathways relating to antigen presentation and myeloid differentiation (Supplementary Fig. 3g). cDC2.3, the most mature cluster in the cDC2-to-CCR7[+] DC trajectory, upregulated *Ciita*, class-II MHC (MHC-II) transcripts, and *Ccr7* (Supplementary Fig. 3i). The receptor tyrosine kinase *Axl*, which recognises apoptotic cells and is associated with acquisition of the CCR7[+] DC programme[5], was also upregulated as cDC2.1 transitioned to cDC2.3 (Fig. 1k). Altogether, these data suggest time-associated maturation of cDCs in the TME towards a CCR7[+] state, and importantly, antigen-charged activated DCs may reside in the tumour for several days despite the expression of genes involved in tumour egress, including CCR7.

## Migrated DCs are phenotypically distinct from tumour-residing CCR7[+] DCs

DCs expressing activation transcripts have been identified in tumour-dLNs[5,30] and CCR7 expression directs migration to the dLN. However, given our demonstration that many CCR7[+] DCs acquire prolonged tumour residence, we sought to definitively track tumour DC egress to dLNs. Kaede-red, activated CCR7[+]PD-L2[+] DCs, which were photo-labelled within the tumour and hence tumour emigrants, were readily detectable in the dLN, but not in the contralateral non-draining LN (ndLN, Fig. 2a, b). Indeed, among Kaede-red DCs in the dLN, essentially all were CCR7[+] DCs, up to 72 h post-photoconversion (Fig. 2c). scRNA-seq of Kaede-red CD45[+] cells in tumour-dLNs (integrated with CD45[+] cells from control LNs) included a prominent *Ccr7[+]Cd274[+]Pdcd1lg2[+]* DC cluster among myeloid cells (Fig. 2d, Supplementary Fig. 4a). In this data, 94% of Kaede-red[+] myeloid cells in the dLN were Ccr7_DCs (Fig. 2e). Hence, CCR7[+] DCs are the dominant myeloid cell tumour emigrants arriving in the dLN.

Compared to Kaede-red tumour emigrants in the dLN, CCR7[+] DCs in tumours were enriched in '*interferon gamma response*' genes (Fig. 2f). While there was no difference in '*lymphocyte co-stimulation*' geneset enrichment between sites, expression of precise co-stimulatory molecules differed (Fig. 2f, Supplementary Fig. 4b, c). Specifically, CD80 and CD86 were higher in tumour-residing CCR7[+] DCs, confirmed at a protein level, while *Icosl* was significantly higher in migrated CCR7[+] DCs in the dLN (Fig. 2g, h, Supplementary Fig. 4b). Of note, CD80/86 and ICOSL have differing effects on T cell activation, where signalling through CD28 but not ICOS induces IL-2 production to support the clonal expansion of T cells[31]. *Il12b*, which drives anti-tumour cytotoxic T cell activity[32], was enriched in tumour CCR7[+] DCs, while *Il15ra*, associated with DC maturation[33], was higher in CCR7[+] DC in the dLN. Notably, scRNA-seq of B16-F10 tumours and their dLNs[34] demonstrated the same tissue-associated heterogeneity in CCR7[+] DCs (Supplementary Fig. 4d–g).

We next asked whether the three transcriptionally distinct tumour CCR7[+] DC states contributed equally to the dLN emigrants. To address this, we integrated the single-cell transcriptomes of tumour-originating DCs in the dLN with the tumour DC landscape (Fig. 2i). Strikingly, over 80% of Kaede-red DCs from the dLN mapped to the tumour Ccr7_DC.1 cluster (Fig. 2j). Only 9% of DCs in the dLN resembled Ccr7_DC.2/3, despite Ccr7_DC.2/3 being the dominant Kaede-red tumour CCR7[+] DC states. Although CCR7 expression levels were similar in both tumour-residing or migrated CCR7[+] DCs, we observed a decrease in transcripts associated with DC chemotaxis and CCR7 signalling in Ccr7_DC.2/3 (Supplementary Fig. 4h–j). Moreover, CD80 and CD86 expression was upregulated on Kaede-red versus Kaede-green tumour CCR7[+] DCs, with expression levels on tumour Kaede-green CCR7[+] DCs similar to migrated CCR7[+] DCs in matched dLNs (Fig. 2h). Therefore, successful LN emigrants resemble newly-formed CCR7[+] DC, and tumour-retained CCR7[+] DCs acquire a distinct phenotype with prolonged tumour dwell-time. Altogether, these data suggest that Ccr7_DC.1, which have most recently acquired the DC activation programme, are the main population to seed the dLN. Hence, we propose that Ccr7_DC.2 and Ccr7_DC.3 are terminal, tumour-residing CCR7[+] DC states, and DCs that have transitioned beyond the intermediate Ccr7_DC.1 state become increasingly unlikely to egress.

## Tumour-retained CCR7[+] DCs progress towards an "exhausted"-like state but is attenuated by anti-PD-L1 treatment

Since tumour-retained CCR7[+] DCs are reduced in their migratory capacity, we sought to assess how their transcriptional programmes changed with increasing time in the tumour. During the progression from Ccr7_DC.1 to Ccr7_DC.3, there was a decrease in MHC-II expression, including class-II invariant chain *Cd74* (Fig. 3a, Supplementary Fig. 5a). Concomitantly, DC antigen presentation genes decreased as cells transitioned towards the tumour-retained Ccr7_DC.2/3 states (Fig. 3b, c). This included downregulation of molecules involved in antigen processing (*Ctsb, Ctsl, Ifi30, Lgmn, Psme2*), transport (*Tap1, Tap2*), chaperones (*Hsp1a1, Hspa2, Hspa8*), MHC loading (*H2-DMa, H2-Oa, Tapbp, Calr*) and *Ciita* (Supplementary Fig. 5b, c).

Moreover, genes involved in innate immune function and response to inflammatory stimuli, including '*interferon gamma/alpha response*', '*inflammatory response*', '*cytokine-cytokine receptor interaction*' and '*Toll-like receptor signalling*' pathways, significantly decreased as cells transitioned from Ccr7_DC.1 to Ccr7_DC.3 (Fig. 3d). Specifically, Ccr7_DC.3 showed reduced expression of several innate immune response genes (*Nlrp3, Tlr1, Cd14*), genes involved in DC activation and migration (*Axl, Ccr7, Rgs1, Icam1*), lymphocyte co-stimulation (*Cd40, Tnfsf9, Pvr*) and cytokines which may recruit and activate immune cells (*Cxcl9, Il1b, Tnf*) compared to Ccr7_DC.1 (Fig. 3e, Supplementary Fig. 5d). To identify transcription factors (TF) that accompany the tumour-retained CCR7[+] DC state, we performed TF regulon analysis, which revealed that *Tcf7* expression and activity were upregulated in terminal CCR7[+] DC states (Supplementary Fig. 5e). Altogether, these data suggest that CCR7[+] DCs retained in the tumour undergo a transition, acquiring the transcriptional hallmarks of "exhaustion", as defined in T cells[20], namely reduced expression of molecules enabling effector function (e.g. antigen presentation), sustained expression of inhibitory molecules (such as PD-L1/L2), and a transcriptional state distinct from that of functional effector cells (i.e. successful dLN emigrants).

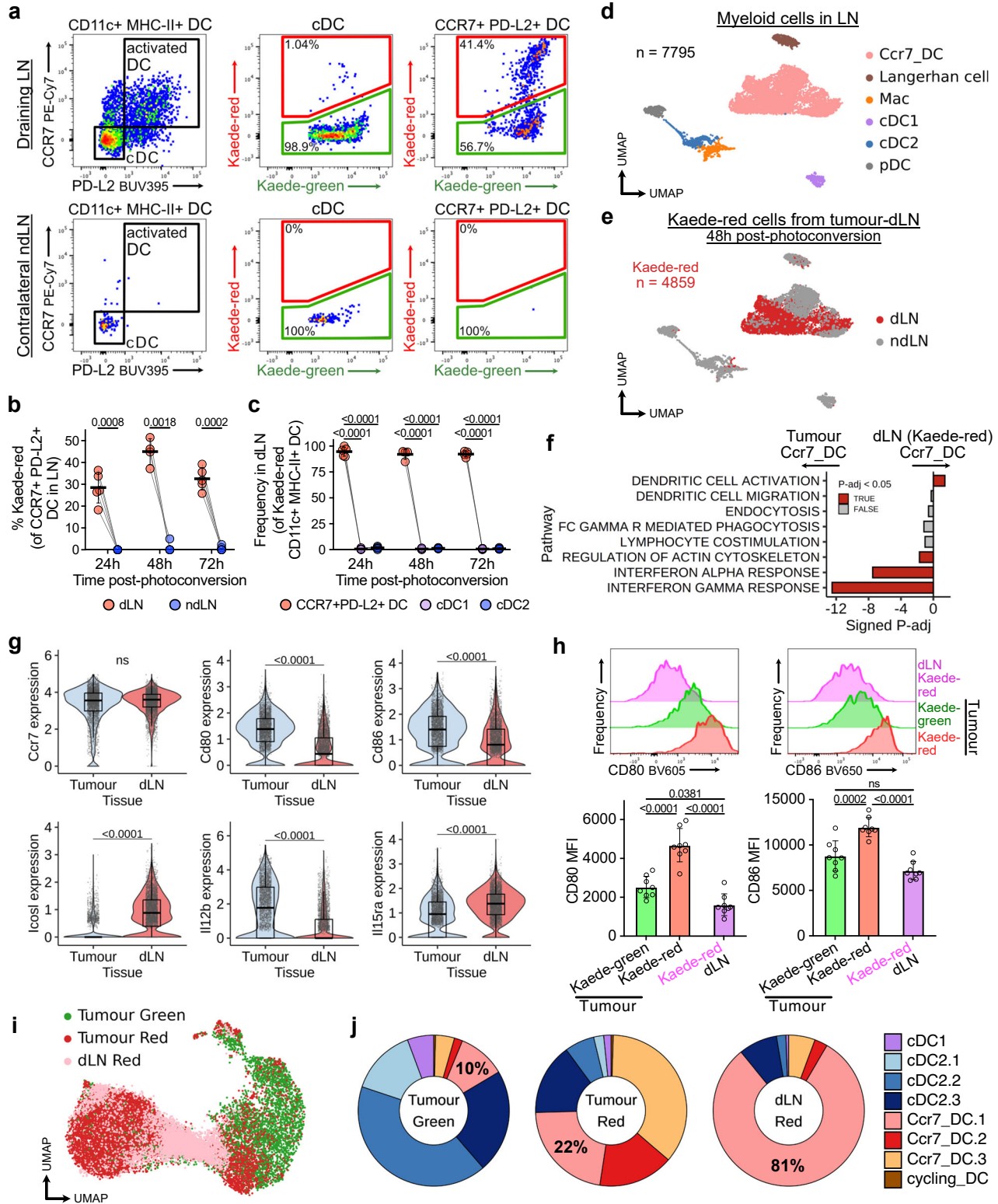

Next, we considered the effects of anti-PD-L1 treatment on tumour DCs, since PD-L1 expression by DCs is essential for effective responses to anti-PD-L1 antibodies[16,17]. In cDC2s from anti-PD-L1-treated tumours, genes involved in inflammation and adaptive immunity were upregulated and expression of *Axl* increased (Supplementary Fig. 3h, j). Across all Ccr7_DC clusters, differential gene expression following anti-PD-L1 drove enrichment of similar pathways, including "*interferon gamma response*", necessary for effective

response to ICB[32], and "*TNFα signalling* via *NFκB*" (Supplementary Fig. 6a, b).

We then asked whether anti-PD-L1 treatment influenced tumour CCR7+ DC heterogeneity. Assessment of differential abundance showed significant enrichment within Ccr7_DC.2 neighbourhoods following anti-PD-L1 treatment, but relative depletion of Ccr7_DC.3 (Fig. 3f), consistent with kernel density embeddings (Fig. 1e). RNA velocity analysis suggested that this was driven by a preferential

**Fig. 2 | CCR7$^+$ DCs that migrate to the dLN become phenotypically and transcriptionally distinct from tumour-residing populations. a–c** Flow cytometry of draining lymph nodes (dLN) and contralateral non-draining lymph nodes (ndLN) of MC38-Ova tumours; DCs that originate from photo-flashed tumours (tumour emigrants) carry the Kaede-red fluorescent profile, which enables tracking to LNs. **a** Representative plots 48 h after tumour photoconversion, **b** Time-course and quantification of Kaede-red cells among CCR7$^+$ DCs, and **c** composition of Kaede-red DCs in dLNs. **d, e** UMAP of myeloid cells from scRNA-seq of FACS-sorted CD45$^+$ Kaede-red cells from tumour-dLNs (MC38-Ova) and CD45$^+$ cells from control LNs. **f** Gene set enrichment analysis (GSEA) of Kaede-red CCR7$^+$ DCs in dLNs versus tumours (isotype control-treated). Signed P-adj indicate log$_{10}$(Benjamini-Hochberg adjusted $P$ value) with the direction of enrichment. **g** Selected gene expression in scRNA-seq of tumour CCR7$^+$ DCs (isotype control-treated) and Kaede-red CCR7$^+$ DC

tumour emigrants in the dLN. **h** Flow cytometry of CD80 and CD86 on CCR7$^+$ DCs from MC38-Ova tumours or dLNs 48 h after photoconversion, with representative histograms. **i** Integration of scRNA-seq of Kaede-red DCs from tumour-dLNs with tumour DCs and label transfer. **j** Proportion of DC clusters by tissue and Kaede fluorescence from **i**. Paired two-sided student's t-test (**b, c**), two-sided Wilcoxon rank-sum test with Benjamini-Hochberg multiple-testing correction (**g**), or one-way analysis of variance (ANOVA) and Šidák's multiple comparisons test (**h**) were used. Points represent independent mice (**b, c, h**). Data are shown as means ± s.d. (**b, c, h**), or box (median; box, 25$^{th}$ and 75$^{th}$ percentile; whiskers, 1.5*inter-quartile range) and violin plots (**g**). The results shown in **a–c** are from one experiment (24 h/72 h $n$ = 5; 48 h $n$ = 4 animals), representative of three independent experiments; and **h** from two independent experiments ($n$ = 8 animals).

transition from Ccr7_DC.1 to Ccr7_DC.2 in anti-PD-L1-treated tumours (Supplementary Fig. 6c). Strikingly, the transcriptome of Ccr7_DC.2 was highly activated compared to Ccr7_DC.3, with increased expression of many genes involved in immune signalling, including *Cd40*, *Il1b*, *Tnf*, *Nfkbia*, etc. (Fig. 3g). This was reflected in the pathway enrichment analysis, which showed multiple genesets relating to immune activation enriched in Ccr7_DC.2 (Supplementary Fig. 6d), potentially augmenting their capacity to promote anti-tumour responses.

Importantly, we identified several lymphocyte activation ligands[32,35,36] that were significantly upregulated on Ccr7_DC.2 versus other tumour CCR7$^+$ DC states and Kaede-red dLN CCR7$^+$ DCs, such as *Il12b*, *Tnfsf4*, *Tnfsf9*, *Cd70* and *Pvr* (Supplementary Fig. 6e). These molecules were also preferentially expressed on tumour-residing CCR7$^+$ DCs in B16-F10 tumours (Supplementary Fig. 4g). Using OX40L$^{Hu-CD4}$ (*Tnfsf4*)-reporter mice, we confirmed that some tumour CCR7$^+$ DCs express OX40L, but it was not expressed by tumour cDCs or CCR7$^+$ DCs in the dLN (Fig. 3h, Supplementary Fig. 6f, g). The absence of OX40L expression in the dLN suggests that it is only upregulated by tumour-retained CCR7$^+$ DCs, and that OX40L$^+$CCR7$^+$ DC do not emigrate to the dLN, or at least rapidly downregulate it upon tumour exit. Altogether, these data underline that a subset of tumour-retained CCR7$^+$ DCs exhibit a specific, "activated" state, which is enriched following anti-PD-L1 treatment.

### CCR7$^+$ DCs interact with CD8$^+$ T cells in human tumours

Given that a substantial proportion of CCR7$^+$ DCs in the tumour failed to migrate to dLN, and that tumour CCR7$^+$ DCs may associate with outcomes, we sought to identify which immune cells CCR7$^+$ DCs may communicate with in the tumour. We deconvolved 521 human CRC transcriptomes[23] and found the highest correlation between CCR7$^+$ DC and CD8$^+$ T cell transcripts (Fig. 4a), an observation replicated in melanoma, breast, and lung tumour biopsies (Fig. 4b). Furthermore, CCR7$^+$ DCs had stronger predicted interactions with effector CD8$^+$ T cells than other myeloid cells, including via CD28, CTLA-4 and PD-1 (*PDCD1*) engagement (Fig. 4c). Similar observations were confirmed in scRNA-seq of human breast tumours and melanoma (Supplementary Fig. 1d–g).

Effective engagement of cell-surface ligand-receptor pairs require co-localisation of CCR7$^+$ DCs and effector CD8$^+$ T cells. Analysis of spatial transcriptomics data from human CRC tumours[37] revealed hotspots with co-localised expression of CCR7$^+$ DC and effector CD8$^+$ T cell genes (Fig. 4d, Supplementary Fig. 7a, b). Spatial correlation of CCR7$^+$ DC and effector CD8$^+$ T cell transcripts was also evident in human melanoma and breast tumours (Fig. 4e, f).

Finally, we analysed data from RNA-seq of physically-interacting cells (PICs), consisting of sorted myeloid-T cell doublets from NSCLC[38]. Of note, CCR7_DC-T cell PICs were more frequent in tumours than normal tissue[38]. We found that CCR7_DC-CD8$^+$ T cell doublets were more frequent than other CCR7_DC-T cell combinations (Supplementary Fig. 7c, d), and PICs containing activated DC (*CCR7$^+$FSCN1$^+$CD274$^+$*) highly co-expressed *CD8B*, *PRF1*, *GZMB* and *PDCD1* (Fig. 4g),

confirming that in NSCLC, CCR7$^+$ DC and effector CD8$^+$ T cells physically interact. Altogether, these data suggest that CCR7$^+$ DC-CD8$^+$ T cell interaction is conserved across multiple human solid tumours. The prolonged tumour dwell-time of CCR7$^+$ DCs, which maintain high levels of PD-1 ligand expression but downregulate expression of genes enabling effector function, suggests that these cellular interactions are potentially deleterious. Specifically, tumour-retained CCR7$^+$ DCs may regulate the activation and expansion of anti-tumour cytotoxic T cells, but they could also be important targets of cancer immunotherapy.

### Anti-PD-L1 promotes immunogenic CCR7$^+$ DC-CD8$^+$ T cell interactions

To investigate the effects of anti-PD-L1 on CCR7$^+$ DC-CD8$^+$ T cell interactions in tumours, we first analysed scRNA-seq of TILs (Supplementary Fig. 8a, b), focussing on PD-1 (*Pdcd1*)-expressing CD8$^+$ T cells which are the target of PD-L1–PD-1 checkpoint blockade. These included a major *Prf1$^{high}$Gzm$^{high}$Pdcd1$^+$Havcr2$^+$* cluster resembling "exhausted" T (T$_{EX}$) cells and a *Pdcd1$^+$Tcf7$^+$Slamf6$^+$* cluster resembling "stem-like" T cells[39,40] (Fig. 5a, Supplementary Fig. 8c, d), that were predominantly tumour-resident and expanded with anti-PD-L1 (Fig. 5b, Supplementary Fig. 8e). *Pdcd1$^+$* cells were evident among the cycling_CD8T cluster (Supplementary Fig. 8f, g) and increased in proliferation following anti-PD-L1, potentially underpinning their increased number (Supplementary Fig. 8h, i). Moreover, the T$_{EX}$ cluster showed the largest transcriptional response to anti-PD-L1 (Fig. 5c), including upregulation of genes involved in "*TCR signalling*" and "*IL2-STAT5 signalling*" with potential anti-tumour benefits (Supplementary Fig. 8j). Notably, Kaede-red$^+$PD-1$^+$ CD8$^+$ T cells showed significant increases in Granzyme B and IFNγ protein expression following anti-PD-L1 treatment (Supplementary Fig. 8k, l).

Given CCR7$^+$ DCs are a major source of PD-L1 in tumours, and the marked response of CD8$^+$ T$_{EX}$ cells to anti-PD-L1 treatment, we sought to identify CCR7$^+$ DC-mediated interactions that might promote their activation and proliferation (Fig. 5d). Cell-cell communication analysis showed several previously described interaction pairs, including *Cxcl16* and *Il15* from activated DCs, which may recruit and sustain cytotoxic T cells (Supplementary Fig. 9a)[41]. Importantly, we identified TNF-superfamily interactions, including *TNFSF4–TNFRSF4* (OX40L–OX40), *TNFSF9–TNFRSF9* (4-1BBL–4-1BB), *CD70–CD27*, and *PVR*-mediated interactions upregulated between Ccr7_DC.2, which were enriched with anti-PD-L1 treatment, and CD8$^+$ T$_{EX}$ cells (Fig. 5d). These TNF-superfamily ligands are known to promote T cell survival, proliferation, and activation[35], and PVR engages the activating receptor CD226, or its competing inhibitory receptor TIGIT, to control CD8$^+$ T cell effector function[36]. Similar interactions were observed between Ccr7_DC.2 and other PD-1-expressing CD8$^+$ T cells, including *Tcf7$^+$* stem-like cells (Supplementary Fig. 9b). Overall, Ccr7_DC.3 had fewer predicted activating interactions with CD8$^+$ T cells than Ccr7_DC.1/2, supporting their status as an "exhausted" CCR7$^+$ DC state (Supplementary Fig. 9c), but all CCR7$^+$ DC states were a consistently high source of inhibitory PD-1 or CTLA-4 signals (Supplementary Fig. 9c, d).

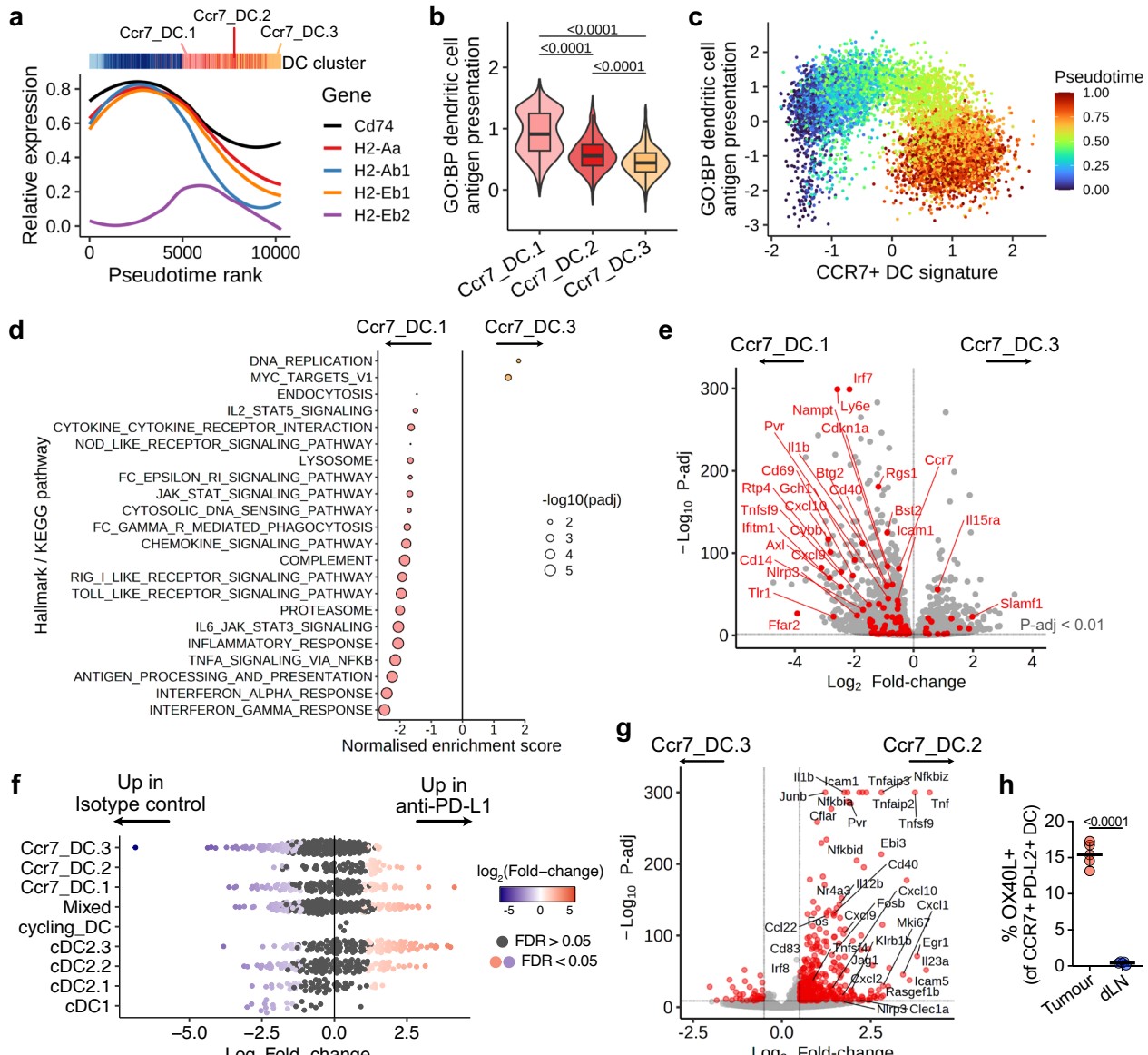

**Fig. 3 | Tumour-residing CCR7+ DCs undertake an "exhausted" state with duration in the tumour, attenuated by anti-PD-L1 treatment. a** Expression of MHC-II transcripts and *Cd74* over pseudotime in tumour DCs. Local regression (loess) was fit to scaled expression values. **b** Gene signature scores for *"GO:BP dendritic cell antigen processing and presentation (GO:0002468)"* by CCR7+ DC cluster. **c** Scaled enrichment scores of scRNA-seq of all DCs for *"GO:BP DC antigen processing and presentation"* and CCR7+ DC signature genes, coloured by pseudotime. **d** GSEA of Ccr7_DC.3 versus Ccr7_DC.1. Only significant pathways (Benjamini-Hochberg-adjusted p values (*P*-adj) < 0.05) are shown. **e** Differential gene expression between Ccr7_DC.1 and Ccr7_DC.3. Significant differentially expressed genes (DEG, *P*-adj < 0.01) from *"Hallmark inflammatory response"* are highlighted in red. **f** Milo differential abundance analysis. Bee-swarm plot shows treatment-associated

differences in overlapping cellular neighbourhoods (individual points). Differentially abundant neighbourhoods at FDR < 0.05 are coloured. 'Mixed' refers to neighbourhoods where cells do not predominantly (>70%) belong to a single cluster. **g** Differential gene expression between Ccr7_DC.2 and Ccr7_DC.3. DEGs (*P*-adj < 0.01, log₂Fold-change > 0.5) are coloured. **h** Flow cytometry of OX40L expression (OX40L+/Hu-CD4 reporter mice) on CCR7+ DC from MC38-Ova tumours and their dLN. Two-sided Wilcoxon rank-sum test (**b**), two-sided Wald test with Benjamini-Hochberg multiple testing correction (**e, g**), or paired two-sided student's t-test (**h**) were used. Data are shown as box (median; box, 25th and 75th percentile; whiskers, 1.5*inter-quartile range) and violin plots (**b**), or means ± s.d. (**h**). Points represent independent mice (**h**). The results shown in **h** are from one experiment (*n* = 5 animals), representative of two independent experiments.

To visualise the cellular interactions predicted by our analysis, we used IF microscopy to confirm that CCR7+ DCs and CD8+ T cells spatially co-localise (Supplementary Fig. 9e, f). There were frequent interactions between tumour-residing CCR7+ DCs and Kaede-red+CD3+CD8+ T cells, often within a perivascular niche, including proliferating (Ki-67+)4-1BB+ cytotoxic cells (Fig. 5e, Supplementary Fig. 9g–i), consistent with a role for CCR7+ DCs in regulating anti-tumour cytolytic activity.

To test the functional importance of these predicted CCR7+ DC-CD8+ T cell interactions, we generated CCR7+ DCs from bone marrow-

derived dendritic cells (BMDC) by culturing with apoptotic MC38-Ova cells, as previously described[5]. This led to a robust expression of DC activation markers, including PD-L1, PD-L2 and CCR7 (Fig. 5f, Supplementary Fig. 10a, b). Of note, the activated, tumour-experienced BMDCs generated by this system also expressed OX40L and PVR (Supplementary Fig. 10c), resembling the phenotype of tumour-retained CCR7+ DC in vivo.

Co-culture of FACS-sorted tumour-antigen (Ova)-experienced CCR7+ BMDC with naïve Ova-specific CD8+ T cells (OT-I) for 3 days

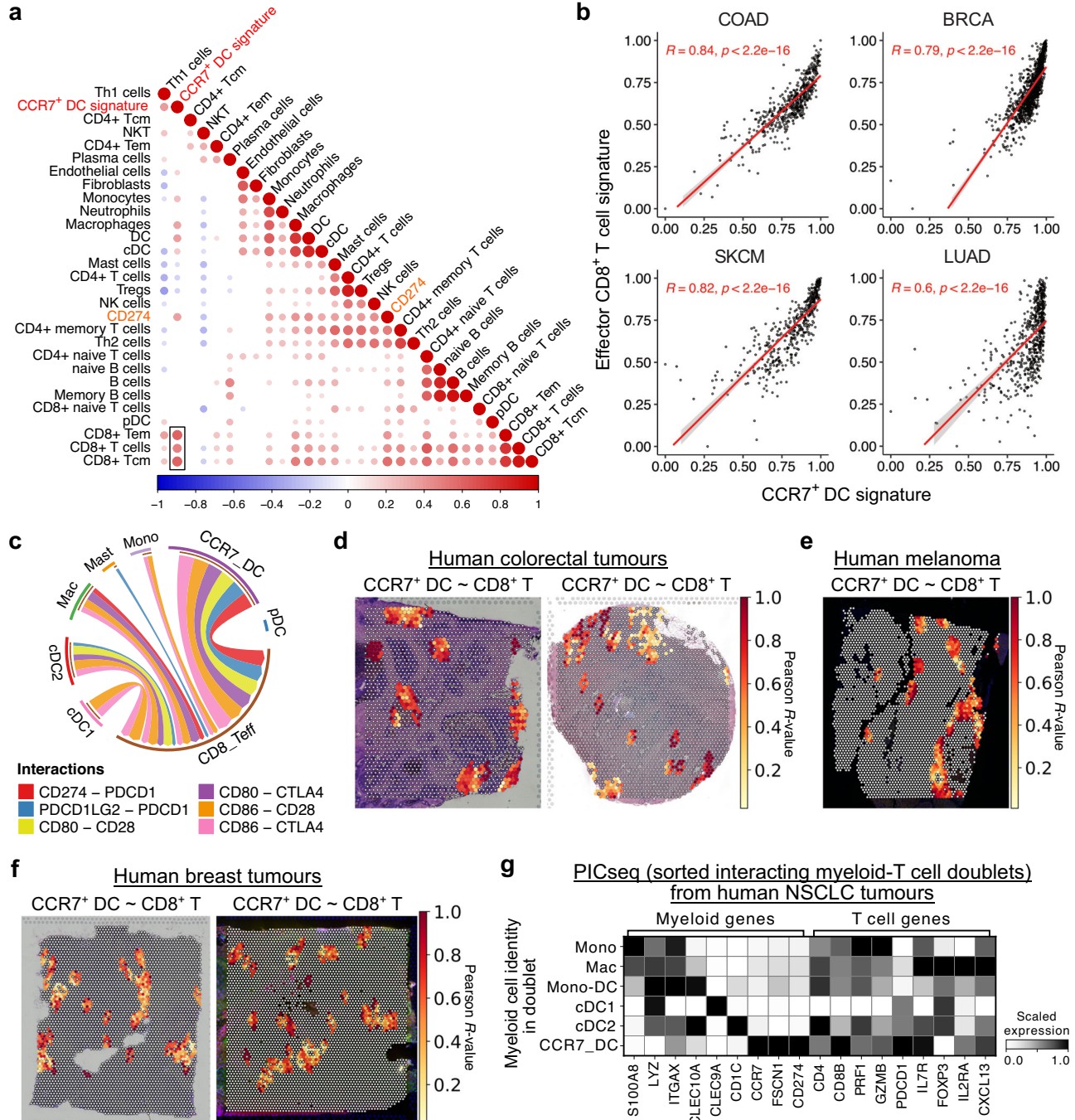

**Fig. 4 | CCR7+ DCs interact with CD8+ T cells in human tumours. a** Pearson correlation between cell proportions, from deconvolution of 521 bulk transcriptomes of colorectal adenocarcinoma biopsies (TCGA), *CD274*, and a CCR7+ DC gene signature. Box highlights the high correlation between CCR7+ DC and CD8+ T cell transcripts. Only significant correlations (*p* < 0.05) shown. **b** Pearson correlation between CCR7+ DC signature genes and effector CD8+ T cell signature genes in colorectal adenocarcinoma (COAD), breast cancer (BRCA), skin cutaneous melanoma (SKCM) and lung adenocarcinoma (LUAD) from TCGA. Points represent individual patient samples. **c** CellPhoneDB cell-cell communication analysis between myeloid cells and effector CD8+ T cells in scRNA-seq of human CRC (*n* = 62 patients). Edge width scaled to standardised interaction scores. Only significant

interactions (*p* < 0.05) shown. Mono, monocyte; Mac, macrophage; Mast, mast cell; pDC, plasmacytoid DC; CD8_Teff, effector CD8+ T cell. Spatial correlation (Pearson R-value) of CCR7+ DC and effector CD8+ T cell signature scores in spatial transcriptomics (10X Genomics Visium) of independent human CRC tumour sections (**d**, *n* = 2), a human melanoma section (**e**, *n* = 1), and independent human breast tumour sections (**f**, *n* = 2). **g** Expression of selected genes in myeloid-T cell doublets from sequencing of physically interacting cells (PICseq) of non-small cell lung cancer (NSCLC) tumours (*n* = 10 patients), grouped by the myeloid cell identity in each myeloid-T cell doublet. Two-sided Pearson correlation was used (**a-b**). Data are shown as linear regression with 95% confidence interval (**b**).

(Fig. 5g) led to OT-I proliferation, which was not observed unless DCs were exposed to Ova (Supplementary Fig. 10d–f). Notably, the addition of anti-PD-L1 antibodies to the CCR7+ BMDC:OT-I co-culture resulted in an increase in activated (CD44+CD25+), clonally-expanded

OT-I cells, and enhanced the production of granzyme B among activated T cells (Fig. 5h, i, Supplementary Fig. 10g).

To assess the role of *Tnfsf4* (OX40L), a ligand expressed by the activated Ccr7_DC.2 state increased in anti-PD-L1 treatment and

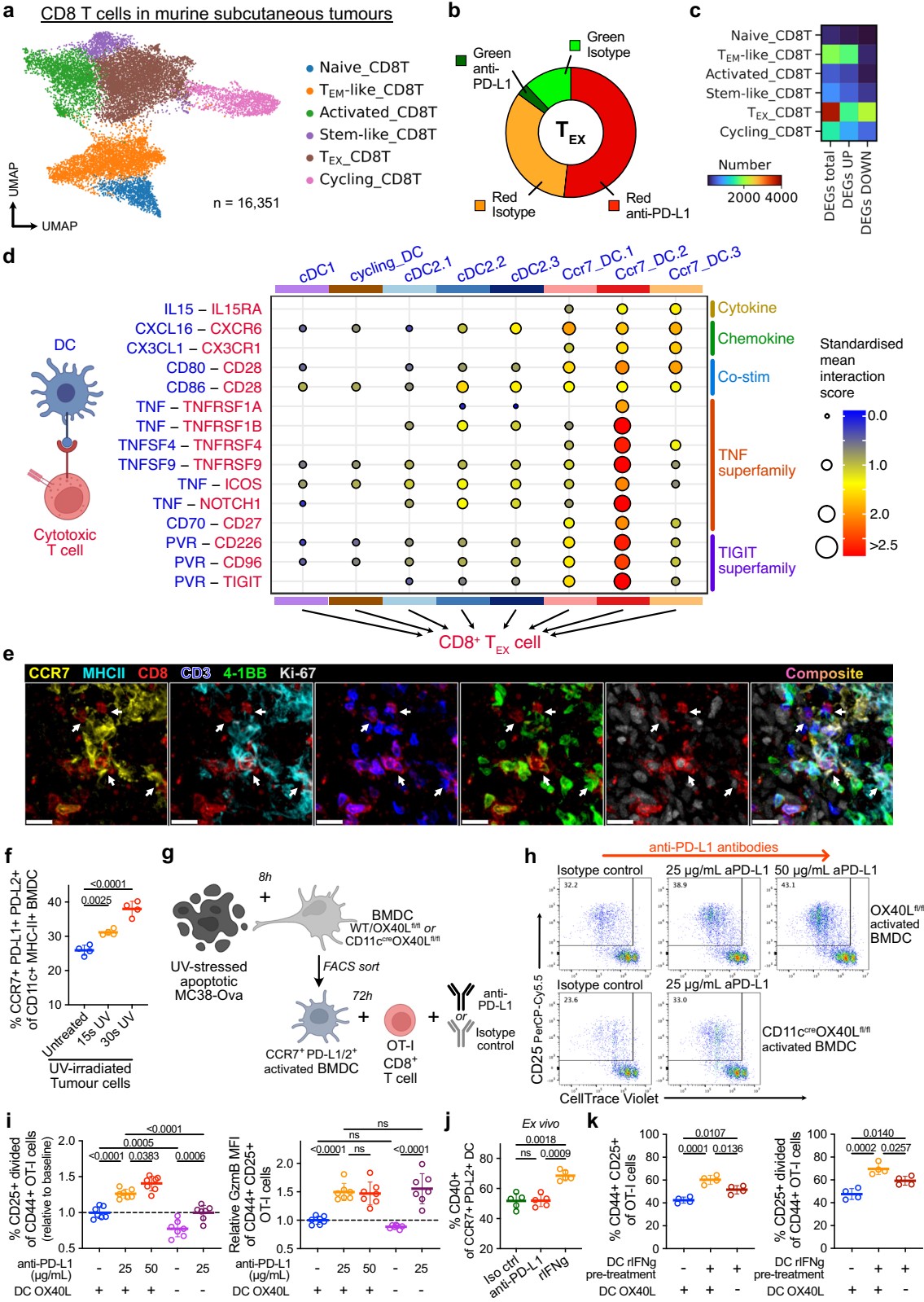

enables interactions with CD8$^+$ T cells via OX40 (Fig. 5d), we generated OX40L-deficient CCR7$^+$ BMDC from CD11c$^{cre}$*Tnfsf4*$^{fl/fl}$ mice. OT-I cells co-cultured with OX40L-deficient CCR7$^+$ BMDC showed reduced activation and proliferation (Fig. 5h, i, Supplementary Fig. 10h, i), suggesting that the OX40L:OX40 axis may be important for CCR7$^+$ DC function in vivo.

Finally, we asked whether anti-PD-L1 treatment directly influences the tumour CCR7$^+$ DC state, or whether this is driven by DC-extrinsic factors. For example, immune checkpoint therapy increases interferon gamma (IFNγ) production by PD-1$^+$CD8$^+$ T cells that co-localise with CCR7$^+$ DCs in tumours (Supplementary Fig. 8k), which may activate tumour DCs[32]. Administration of anti-PD-L1 antibodies to isolated,

**Fig. 5 | anti-PD-L1 promotes activating CCR7⁺ DC-cytotoxic CD8⁺ T cell interactions in the TME. a** UMAP of scRNA-seq of CD8⁺ T cells, from FACS-sorted CD45⁺ tumour-infiltrating lymphocytes 48 h after photoconversion of subcutaneous MC38-Ova tumours. **b** Proportion of $T_{EX}$_CD8T cells by Kaede fluorescence and treatment group. **c** Number of significant DEGs in anti-PD-L1 versus isotype control-treated tumours by cluster. DEGs calculated using Wilcoxon rank-sum test with Benjamini-Hochberg multiple-testing correction. **d** CellPhoneDB ligand-receptor analysis between tumour DCs and CD8⁺ $T_{EX}$ cells. Only significant interactions ($p < 0.05$) shown. **e** Representative confocal microscopy images of MC38 tumours, showing co-localisation of CCR7⁺MHC-II⁺ DCs and CD3⁺CD8⁺ 4-1BB⁺/Ki-67⁺ T cells (arrows). Scale bar, 20 μm. (**f**–**k**) In vitro and ex vivo cultures. **f** Phenotype of bone marrow-derived DCs (BMDC) following culture with UV-irradiated MC38-Ova tumour cells. **g** Experiment set-up of DC:OT-I co-culture. **h** Representative flow cytometry of CD44⁺ OT-I cell activation and proliferation following co-culture with MC38-Ova-experienced CCR7⁺ BMDC; +/− anti-PD-L1, OX40L-expressing (+, OX40L^fl/fl) or OX40L-deficient (−, CD11c^cre OX40L^fl/fl) BMDCs. **i** Quantification of

h and granzyme B (GzmB) expression. Values shown are relative to cultures with isotype control antibodies and OX40L-expressing DCs (baseline), to facilitate comparisons across experiments. **j** Flow cytometry of CCR7⁺ DCs from subcutaneous MC38-Ova tumours following 8 h culture ex vivo with antibodies or recombinant IFNγ (rIFNg). **k** Flow cytometry of OT-I cells following co-culture with MC38-Ova-experienced CCR7⁺ BMDC; CCR7⁺ BMDCs pre-treated with rIFNg (+) or PBS (−), OX40L-expressing (+) or OX40L-deficient (−) BMDCs. One-way analysis of variance (ANOVA) and Šidák's multiple comparisons test was used, data are shown as means ± s.d. (**f**, **i**–**k**). The results shown in (**e**) are representative of two independent experiments ($n = 5$ animals); **f** from one experiment ($n = 4$ biological samples) representative of three independent experiments; **h**, **i** from two independent experiments ($n = 7$ biological samples); **j** from one experiment ($n = 5$ animals), representative of two independent experiments; and (**k**) from one experiment ($n = 4$ biological samples), representative of two independent experiments.

tumour antigen-experienced CCR7⁺ BMDC in vitro did not alter their phenotype, but the addition of recombinant IFNγ upregulated expression of OX40L, PVR and CD40, consistent with the activated Ccr7_DC.2 state (Supplementary Fig. 10j). Tumours cultured ex vivo with IFNγ resulted in similar activation of CCR7⁺ DCs (Fig. 5j). To assess the significance of IFNγ-induced changes, we pre-treated CCR7⁺ BMDC with IFNγ, prior to OT-I co-culture. This increased the activation and expansion of OT-I cells but was reduced in cultures containing OX40L-deficient CCR7⁺ DCs (Fig. 5k, Supplementary Fig. 10k). Altogether, these data support the conclusion that tumour CCR7⁺ DC states can be manipulated to promote antigen-specific CD8⁺ T cell responses.

## Conserved CCR7⁺ DC heterogeneity and CD8⁺ T cell crosstalk in human cancers

We asked if the time, tissue, and treatment-associated heterogeneity in CCR7⁺ DCs observed in our murine model was pertinent to human cancers. In CCR7⁺ DCs from human CRC (Supplementary Fig. 1b), there was a gradient of MHC-II expression (Fig. 6a). Importantly, markers upregulated on tumour-residing *MHC-II*^low Ccr7_DCs in murine tumours were also preferentially expressed in the human *MHC-II*^low CCR7_DCs, including *IL15, PVR, TNFSF4, TNFSF9,* and *CD70*.

To assess tissue-associated differences in CCR7⁺ DCs, we analysed an independent human CRC dataset with paired scRNA-seq of tumour, dLN and normal adjacent tissue[42] (Supplementary Fig. 11a–c). Consistent with our findings, there was no difference in *CCR7* expression, but *CD80, CD86, TNFSF4, TNFSF9, CD70* and *PVR* were higher in tumour-residing CCR7_DCs versus dLN and normal tissue, while *ICOSLG* was enriched in the dLN (Fig. 6b, Supplementary Fig. 11d, e). Therefore, the heterogeneity of CCR7⁺ DCs in murine tumours are paralleled in human CRC.

Next, we sought to address whether intra-tumour CCR7⁺ DCs, and their precise phenotype, influences clinical response to ICB. Atezolizumab (anti-PD-L1 antibody) is widely used in the treatment of metastatic urothelial carcinoma (mUC), but its efficacy is variable[43]. We analysed 208 bulk transcriptomes of tumour biopsies from the IMvigor210 trial for mUC[44,45]. In both responders and non-responders, CCR7⁺ DC gene expression was positively correlated with effector CD8⁺ T cell transcripts, and was higher in responders (Supplementary Fig. 11f). To leverage the clinical response data from this cohort, we integrated the bulk transcriptomes with scRNA-seq of myeloid cells in mUC[46] (Fig. 6c, Supplementary Fig. 11g) using *Scissor*[47], which enables identification of clinically relevant cell subpopulations and gene expression profiles. This analysis revealed that CCR7_DCs were enriched in responders (Fig. 6d). Indeed, myeloid cells associated with a favourable clinical response expressed transcripts associated with tumour-residing CCR7⁺ DCs, including *CCR7, CD274, IL15, PVR* and *CD70* (Fig. 6e).

Moreover, in scRNA-seq of breast cancer (Supplementary Fig. 1d, 11h)[24], CCR7_DCs from tumours that successfully underwent clonal T cell expansion following treatment with anti-PD-1 antibodies were enriched in transcripts associated with our ICB-induced Ccr7_DC.2 cluster, versus non-responders (Fig. 6f). CCR7_DCs from T cell clonotype expanders also upregulated genes contributing to '*interferon gamma response*', '*FcγR-mediated phagocytosis*', '*lymphocyte co-stimulation*', etc. (Supplementary Fig. 11i). Altogether, data from these treatment cohorts support the importance of an ICB-activated tumour-residing CCR7⁺ DC state.

Next, we asked if the molecular crosstalk between CCR7⁺ DCs and CD8⁺ T cells in mice is conserved in humans. In PICseq data from NSCLC[38], we found that doublets containing CCR7_DCs highly expressed ligands we identified in murine tumours (Fig. 5d), and their corresponding T cell receptors were highly expressed in the same CCR7_DC-T cell conjugates (Fig. 6g). Importantly, CCR7_DC-CD8⁺ T cell doublets were most enriched for T cell cytotoxicity genes and '*TNFα response* via *NFκB*' compared to other myeloid-CD8⁺ T cell combinations (Fig. 6h, Supplementary Fig. 11j, k).

Finally, we re-examined hotspots of CCR7⁺ DC and effector CD8⁺ T cell co-localisation in spatial transcriptomics of CRC, breast cancer and melanoma (Fig. 4d–f). Molecules involved in CCR7⁺ DC-CD8⁺ T cell crosstalk were expressed in these voxels, with spatial correlation of CCR7⁺ DC-CD8⁺ T cell ligand-receptor pairs in all 3 tumour types (Fig. 6i, j, Supplementary Fig. 12a–c). Altogether, these data demonstrate that CCR7⁺ DCs are critically positioned to regulate anti-tumour cytolytic activity, including via TNF-superfamily ligands such as OX40L, and PVR.

## Discussion

In this study, we combined scRNA-seq with a photoconvertible murine tumour model to unravel the spatio-temporal dynamics of tumour CCR7⁺ DCs. Contrary to current assumptions, CCR7⁺ DCs are heterogeneous, including sub-populations that either primarily contribute to LN migration or are retained in the tumour. Since CCR7⁺ DCs do not uniformly and instantaneously migrate to the dLN, and influence local tumour immunity, this activated DC subset cannot be unequivocally labelled as "migratory DC". We found that tumour-retained CCR7⁺ DCs acquire transcriptional features consistent with functional "exhaustion" with prolonged tumour residence, but following anti-PD-L1 treatment, are skewed towards a state enriched in T cell stimulatory molecules, capable of augmenting anti-tumour cytotoxic T cell responses. Their heterogeneity, crosstalk with CD8⁺ T cells, and association with response to ICB was conserved across human cancers.

Our work suggests that CCR7⁺ DCs serve as a critical cellular immunoregulatory hub within the TME, through the provision of

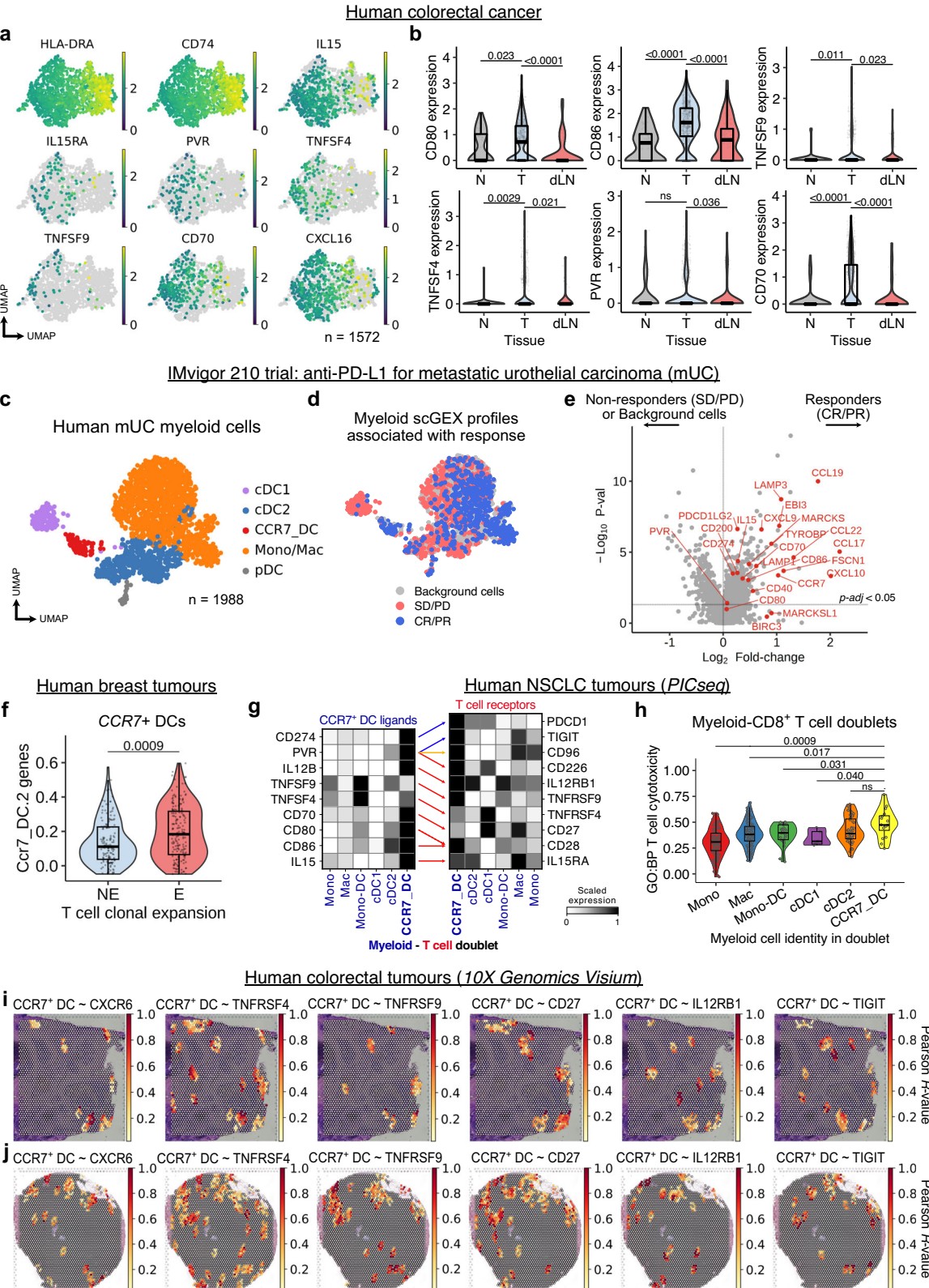

**Human colorectal cancer** (a, b)

**IMvigor 210 trial: anti-PD-L1 for metastatic urothelial carcinoma (mUC)** (c, d, e)

**Human breast tumours** (f)

**Human NSCLC tumours (PICseq)** (g, h)

**Human colorectal tumours (10X Genomics Visium)** (i, j)

chemotactic, survival, activating, and inhibitory factors. We identified a CCR7+ DC-CD8+ T cell axis, including specific interactions which control anti-tumour cytolytic activity. Consistent with this, acquisition of the full effector CD8+ T cell program requires engagement of CD80high DCs in the local TME[48], which we propose are CCR7+ DCs. However, while tumour-retained CCR7+ DCs increase CD80/86 expression, concomitant downregulation of immunogenic transcriptional programmes

obscures its functional significance, analogous to the increased expression of granzymes by terminally-exhausted CD8+ T cells[49]. Recent studies also report that CCR7+ DCs may engage other immune cells in tumours, including CXCL13+ CD4+ T cells, regulatory T cells and NK cells[38,50–52], and may reside in TLS[12,15]. How heterogeneous CCR7+ DC states influence the survival or activation of other immune cells, including within TLS, warrants further investigation.

**Fig. 6 | Conserved CCR7+ DC heterogeneity and CD8+ T cell crosstalk in human cancer. a** UMAP of scRNA-seq of CCR7+ DCs from human CRC (GSE178341, *n* = 62 patients). **b** Selected gene expression in CCR7+ DCs from tumour (T), normal adjacent tissue (N) and tumour-dLN from independent scRNA-seq data of human CRC (syn26844071, *n* = 63 patients). **c** UMAP of myeloid cells from scRNA-seq of human metastatic urothelial carcinoma (mUC, *n* = 11 patients). **d** single-cell gene expression (scGEX) profiles associated with responders (complete responder, CR; partial responder, PR) or non-responders (stable disease, SD; progressive disease, PD) in the IMvigor 210 trial (atezolizumab) for treatment of mUC; Scissor integration of bulk RNA-seq of tumour samples from IMvigor210 (*n* = 208 patients) and scRNA-seq from **c**. **e** Differential gene expression in myeloid cells associated with responders (CR/PR) versus non-responders (SD/PD) from **d**. **f** Gene signature scores for Ccr7_DC.2 transcripts in scRNA-seq of CCR7+ DCs from human breast cancer following treatment with anti-PD-1 antibodies; T cell clonotype expanders (E, i.e. responders, *n* = 9 patients) versus non-expanders (NE, i.e. non-responders, *n* = 20 patients). **g** PICseq (myeloid-T cell doublets) of human NSCLC; expression of CCR7+ DC-T cell ligand-receptor pairs, grouped by myeloid cell identity in each myeloid-T cell doublet (*n* = 10 patients). **h** Gene signature scores for "*GO:BP T cell mediated cytotoxicity (GO:0001913)*" (HLA genes removed), in myeloid-CD8+ T cell PICs grouped by the myeloid cell identity. **i, j** Spatial correlation (Pearson R-value) of CCR7+ DC signature scores and selected CCR7+ DC-ligand receptors expressed by CD8+ T cells, in spatial transcriptomics (10X Genomics Visium) of independent human CRC tumour sections (*n* = 2). Two-sided Wilcoxon rank-sum test with Benjamini-Hochberg multiple-testing correction (**b, e**), or two-sided Wilcoxon rank-sum test (**f, h**) were used. Data are shown as box (median; box, 25th and 75th percentile; whiskers, 1.5*inter-quartile range) and violin plots (**b, f, h**).

The success of ICB have led to combination immunotherapies[53], exemplified by the recent success of tiragolumab (anti-TIGIT) + atezolizumab for NSCLC[54]. In CD8+ T cells, mechanistic convergence of PD-1 and TIGIT inhibitory pathways powerfully regulates cytotoxic function[55]. We found that CCR7+ DCs are the major suppliers of both PD-1 and TIGIT ligands to CD8+ T cells, and importantly, there was an increase in PVR expression following anti-PD-L1 treatment, potentially regulating T cell activation via CD226/TIGIT engagement[36]. Independently, combining OX40 or 4-1BB agonists with anti-PD-L1 or anti-PD-1 antibodies respectively improved treatment outcomes in pre-clinical cancer models[56,57]. Given their diverse ligand profile, our data suggest that CCR7+ DCs are a molecular hub through which these combinatorial immunotherapies synergise.

Several questions remain; While provision of PD-1 ligands by DCs are essential for effective ICB[16,17], it remains unclear whether anti-PD-L1 antibodies directly alter DC function. Interestingly, 'reverse signalling' through PD-L1 has been described[58], including effects on DC migration[59]. Our in vitro data does not support a DC-intrinsic effect of anti-PD-L1 on tumour DC phenotype, but a cell type-specific deletion of PD-L1 in vivo would be needed to definitively address this, as in vitro systems may not fully recapitulate the complexity of the TME. Next, knowledge of CCR7+ DC ontogeny remains incomplete[6,8], particularly the relative contribution of cDC subsets or monocytes to heterogeneous tumour-residing CCR7+ DC states, which may influence their ability to support CD8+ versus CD4+ T cell responses. Our analyses suggest both cDC1 and cDC2 precursors contribute, but fate mapping experiments would be required to support this. Finally, whether the retention of CCR7+ DCs in tumours is due to the acquisition of aberrant trafficking behaviour or due to intratumoral interactions, such as chemoattraction to stromal cells or tumour cells expressing CCR7 ligands[60,61] remains to be resolved. Indeed, the presence of CCR7+ DC in TLS suggest that local cues facilitate their retention.

Previous studies have drawn conflicting conclusions on the role of CCR7+ DCs in tumours; Loss of TIM-3 in DCs prevents the acquisition of this DC activation programme, but facilitates maintenance of the effector CD8+ T cell pool[62]. Conversely, CXCL16 and IL-15 expressing CCR7+ DCs may recruit and sustain CXCR6+ cytotoxic T cells crucial in cancer immunosurveillance[41]. Our data help to resolve these contradictions; We propose that tumour CCR7+ DCs are heterogeneous, with subset-specific capacities to both support and inhibit the cytotoxic T cell niche, presenting new opportunities for intervention.

## Methods
### Mice
Wild-type C57BL/6 (Charles River, strain code 632), transgenic C57BL/6 Kaede[28], BALB/c Kaede, OX40L[fl/fl 63], CD11c[cre]OX40L[fl/fl] and OX40L[+/Hu-CD4] reporter mice are maintained and bred at the University of Birmingham Biomedical Services Unit. Wild-type C57BL/6 (Charles River, strain code 632) and OT-I × Rag1[-/-] mice (Taconic, strain code 4175) were maintained and bred at the University of Cambridge Biomedial Services Gurdon

Institute animal facilities. OX40L[fl/fl] mice were crossed with CD11c-cre (JAX, strain code 008068) to obtain CD11c[cre]OX40L[fl/fl]. OX40L[+/Human-CD4] mice were generated by knock-in of human CD4 to the *Tnfsf4* locus (MRC Harwell GEMM scheme). Before experiments, mice were genotyped by PCR. Both male and female mice were used, and within experiments, mice were age- and sex-matched, and co-housed between experimental and control groups. All mice used were between the ages of 8 and 14 weeks and sacrificed by cervical dislocation. All animal experiments were conducted in accordance with Home Office guidelines under project licenses awarded to D.R. Withers and R. Roychoudhuri and were approved by the University of Birmingham Animal Welfare and Ethical Review Body or the University of Cambridge Animal Welfare and Ethics Review Board. Mice were housed at 21 °C, 55% humidity, with 12-h light-dark cycles in 7–7 individually ventilated caging, under specific pathogen-free conditions with environmental enrichment of plastic houses plus paper bedding. Mice were monitored daily.

### Mouse subcutaneous tumour model
MC38 (kindly provided by Dr. Gregory Sonnenberg; Weill Cornell Medicine, New York, NY), CT26 (kindly provided by Professor Tim Elliot, University of Oxford, Oxford, UK) and ovalbumin-expressing MC38 (MC38-Ova, obtained from AstraZeneca) murine colon adenocarcinoma cells were cultured in DMEM and supplemented with 2mM L-glutamine (Thermo Fisher Scientific), 10% FBS (F9665; Sigma-Aldrich), and penicillin-streptomycin (Sigma-Aldrich) at 37 °C with 5% $CO_2$. Cells grown in the log-phase were harvested and resuspended to $2.5 \times 10^6$ cells/ml in Dulbecco's PBS (Sigma-Aldrich) for tumour injection.

$2.5 \times 10^5$ tumour cells in 100 μl were subcutaneously injected into mice in the pre-shaved left flank area under anaesthesia via 2% gaseous isoflurane. For MC38 and MC38-Ova experiments, mice on the C57BL/6 background were used; for CT26 experiments, mice on the BALB/c background were used. For experiments involving anti-PD-L1 treatment, including scRNA-seq, MC38-Ova tumours were used; for DC phenotyping, the specific tumour model used for each experiment are included in figure legends. Tumour size was measured every 2–3 days from day 7 (when tumours were first observed) to sacrifice, with a digital Vernier calliper, and the volume was calculated using the formula $V = 0.5 \times a \times b^2$ in cubic millimetres, where a and b are the long and short diameters of the tumour, respectively. Mice were sacrificed on day 13, 14, 15, or 16 (5 h, 24 h, 48 h, 72 h post-photoconversion respectively, where performed) and tumours were harvested for analysis. Where indicated for specific experiments, tumour-dLNs (i.e. left inguinal LN) and contralateral non-draining lymph nodes were also harvested. The following humane end-points were adhered to: experiments were terminated and animals sacrificed by cervical dislocation if tumour size exceeded 12 mm in diameter or 1.25 $cm^3$ in volume (institutional ethics maximum tumour size), or if animals demonstrated weight loss > 15%, body condition score < 2.0, reduced

activity, failure to respond to gentle stimulation, lethargy, piloerection, continuous hunched posture, or tumour ulceration.

## Administration of anti-PD-L1 antibodies

Anti-PD-L1 mouse IgG1 (Clone 80, SP16–260; AstraZeneca) or NIP228 isotype control mouse IgG1 (SP16-017; Astra Zeneca) were administered on day 7, 10, and 13 after tumour injection by intraperitoneal injection. Each dose consisted of 200 µg antibodies diluted in 200 µl PBS (10 mg/kg body weight).

## Labelling of tumour compartment by photoconversion

Photoconversion was performed as previously described[19,28]. Briefly, on day 13 after tumour injection, the subcutaneous tumour was exposed to a 405-nm wavelength focussed LED light (Dymax BlueWave QX4 outfitted with 8 mm focussing length, DYM41572; Intertronics) for 3 min, with a 5-second break every 20 s, at a fixed distance of 1 cm. Black cardboard was used to shield the remainder of the mouse. We previously showed that this method resulted in complete conversion (99.9%) of host cells within the tumour from the default green fluorescence of the Kaede protein (Kaede-green) to the altered red fluorescence (Kaede-red), and that the cells in the dLN were fully protected from tumour photoconversion[19]. Of note, while converted cells express Kaede-red fluorescence, they also retain a weak Kaede-green signal. Overall, this enabled the discrimination of newly infiltrating (Kaede-green) and resident (Kaede-red) cell populations in the tumour, or cells that have egressed the tumour (Kaede-red) by their fluorescence profile. Moreover, using i.v. administration of anti-CD45 antibodies prior to culling, we previously showed that majority of cells (>95%) were in tumour tissue and were not intravascular contaminants[19].

## Confocal microscopy and analysis

Tumour or LN samples were fixed in 1% paraformaldehyde (Electron Microscopy Services) for 24 h at 4 °C followed by 12 h in 30% sucrose in PBS. 20 µm sections were permeabilized and blocked in 0.1 M TRIS, containing 0.1% Triton (Sigma), 1% normal mouse serum, 1% normal rat serum and 1% BSA (R&D Systems). Samples were stained for 2 h at room temperature in a humid chamber with the appropriate antibodies, listed in Supplementary table 1, washed 3 times in PBS and mounted in Fluoromount-G® (Southern Biotech). Images were acquired using a TCS SP8 (Leica microsystems) confocal microscope. Raw imaging data were processed using Imaris (v9.7.2, Bitplane).

Iterative staining of sections was performed as previously described[64]. Samples were prepared and stained as detailed above. Following acquisition, the coverslips were removed, and slides were washed 3 times in PBS to remove residual mounting medium. Bleaching of the fluorochromes were achieved by submerging the slide in a 1 mg/mL solution of lithium borohydride in water (Acros Organics) for 15 min at room temperature. The slides were then washed 3 times in PBS prior to staining with a different set of antibodies. The process was repeated twice. Raw imaging data were processed using Imaris using CD31 as fiducial for the alignment of acquired images.

For quantification and co-localisation analysis, processed fluorescence imaging data from Imaris were further analysed using QuPath (v0.3.2)[65]. Hoechst nuclear staining was first used to perform automated cell detection with the nucleus diameter setting at 3–10 µm. Thereafter, detections were manually annotated to identify 100 cells each for CCR7+ DCs or CD8+ T cells per image. CCR7+ DCs were annotated based on CD45+MHC-II+CCR7+ staining and dendritic morphology. CD8+ T cells were annotated based on CD45+CD3+CD8+CCR7+/- staining and spherical morphology. Vessels (CD31+) and tumour or stromal cells (CD45-) were annotated as detections to 'ignore'. The manual annotations were used to train a semi-automated random trees object classifier, to automate annotation of remaining cells, using the following parameters: nuclear and

cellular morphology, CD45, CD3, CD8, MHC-II, CCR7, and CD31 staining intensity. The classification output and centroid positions for all cell detections were exported for further analysis in R (v4.0.4). For correlation analysis, tumour sections were divided into grids approximately 20–30 cell detections wide (200 µm) across multiple sections. Number of cell detections of each class were counted per grid and Pearson correlation was applied to quantify spatial co-localisation.

## Tissue dissociation

Tumours were cut into small pieces using surgical scissors, and incubated with 1 mg/ml collagenase D (Roche) and 0.1 mg/ml DNase I (Roche) in a volume of 1.2 ml RPMI media at 37 °C on a thermomixer (Eppendorf) for 20 min; or tumours were digested using Tumour Dissociation Kit (Miltenyi Biotec) and gentleMACS Dissociator (Miltenyi Biotec) for 40 min at 37 °C according to the manufacturer's protocol. The gentleMACS protocol was used for scRNA-seq experiments. Subsequently, the sample was filtered through a 70 µm strainer to remove undigested tissue debris. Next, dead cells were removed using Dead Cell Removal Kit and LS Columns (Miltenyi Biotec), according to the manufacturer's instructions. Lymph nodes were cleaned and dissected in RPMI 1640 medium (Thermo Fisher Scientific) and crushed through a 70 µm strainer. Thereafter, cells were centrifuged at 400 g at 4 °C for 5 min and resuspended in FACS staining buffer (2% FBS; 2 mM EDTA in PBS) for flow cytometry.

## Flow cytometry

Cell suspensions were subjected to Fc block with anti-CD16/32 (Bio-Legend) diluted in FACS staining buffer on ice for 15 min before staining with surface markers, listed in Supplementary table 1, diluted in FACS staining buffer on ice for 30 min, and subsequently, a live/dead stain. Where applicable, cells were then fixed with eBioscience intracellular fixation buffer (Thermo Fisher) for 30 min and stained for intracellular markers diluted in eBioscience permeabilization buffer (Thermo Fisher) at 4 °C overnight. $1 \times 10^4$ counting beads (Spherotech) were added to stained samples at the final step, to calculate absolute cell numbers. Data were acquired on the LSR Fortessa X-20 (BD) using FACSDiva software (v8.0.2, BD) or CytoFLEX (Beckman Coulter) using CytExpert (v2.5, Beckman Coulter) and analysed with FlowJo (v10.8.1, BD).

## Single-cell isolation for RNA sequencing

MC38-Ova tumours were injected in age-matched female Kaede C57BL/6 mice, treated with anti-PD-L1 antibodies, and photoconverted on day 13 as described above. Mice with tumours of similar sizes were collected 48 h after tumour photoconversion. After tumour digestion, as described above, cell suspensions were stained for CD45 BV786, TER119 PE-Cy7, CD11b BV421, NK1.1 BV650, Live/dead APC-Cy7 on ice for 30 min (Supplementary table 1). Subsequently, cells were centrifuged at 400 g at 4 °C for 5 min and resuspended in FACS staining buffer for sorting. Tumour-infiltrating lymphocytes (TIL; Live CD45+TER119-Kaede+CD11b-/lowNK1.1low/hi) and tumour-infiltrating myeloid cells (Live CD45+TER119-Kaede+CD11b+NK1.1-) from anti-PD-L1 or isotype-control treated tumours were sorted with a FACS Aria II Cell Sorter (BD) into two groups per cell type, based on the presence or absence of Kaede-red signal. CD45+ cells were only sorted to 'myeloid' or 'TIL' (CD11b+ or CD11b-/low respectively) fractions to ensure appropriate representation of various cell types in the scRNA-seq data. All single cell transcriptomes were combined at the analysis stage, before cell type annotation, to ensure all CD45+ immune cells, regardless of surface CD11b or NK1.1 expression, are included in the final analysis and annotated based on their transcriptome. Hence, both CD11b+/- DCs are represented in the scRNA-seq data.

Tumour-dLNs from mice treated with the same experimental protocol were harvested and digested as described in the preceding

paragraph. Cell suspensions were stained for CD45 BV785, TER119 PE-Cy7, Live/dead APC-Cy7 on ice for 30 min. Live CD45[+]Ter119[-]Kaede-red[+] cells were sorted, to identify immune cells that have migrated from photoconverted tumours (Kaede-red) to the tumour-dLNs. Inguinal lymph nodes from control mice (no tumour cells injected) were also processed for scRNA-seq to obtain a representation of homeostatic cell populations in the LN. This was done to facilitate accurate annotation and integration of Kaede-red cells in the tumour-dLN, where specific populations arriving from the tumour would be disproportionately over-represented, with the LN cellular landscape. Control LN cell suspensions were stained for B220 BV421, CD11c BV786, TER119 PE-Cy7, Live/dead APC-Cy7 on ice for 30 min. Thereafter, a 3-way sort was performed consisting of Live Kaede[+]TER119[-]CD11c[+]B220[-] (DC), Kaede[+]TER119[-]CD11c[-]B220[+] (B cells), and Kaede[+]TER119[-]CD11c[-]B220[-] fractions. The three subsets were then mixed at a ratio of 1:1:1 to generate a cellular suspension that captured all LN cell populations but was enriched for DCs and with a reduced frequency of B cells. scRNA-seq of control LNs and tumour-dLNs were sequenced and analysed together.

### Single-cell library construction and sequencing

Gene expression libraries from were prepared from FACS-sorted populations of single cells using the Chromium Controller and Chromium Single Cell 3′ GEM Reagent Kits v3 (10x genomics, Inc.) according to the manufacturer's protocol. The resulting sequencing libraries comprised of standard Illumina paired-end constructs flanked with P5 and P7 sequences. The 16 bp 10x barcode and 10 bp UMI were encoded in read 1, while read 2 was used to sequence the cDNA fragment. Sample index sequences were incorporated as the i7 index read. Paired-end sequencing (2 × 150 bp) was performed on the Illumina NovaSeq 6000 platform. The resulting.bcl sequence data were processed for QC purposes using bcl2fastq software (v2.20.0.422) and the resulting fastq files were assessed using FastQC (v0.11.3), FastqScreen (v0.9.2) and FastqStrand (v0.0.5) prior to alignment and processing with the CellRanger (v6.1.2) pipeline.

### Processing of scRNA-seq

Single-cell gene expression data from CellRanger count output (filtered features, barcodes, and matrices) were analysed using the Scanpy[66] (v1.8.2) workflow in Python (v3.8.12). Raw count data from the "myeloid" and "TIL" sorts were concatenated. Doublet detection was performed using Scrublet[67] (v0.2.1), with cells from iterative sub-clustering flagged with outlier Scrublet scores labelled as potential doublets. Cells with counts mapped to > 6000 or < 1000 genes were filtered. The percentage mitochondrial content cut-off was set at < 7.5%. Genes detected in fewer than 3 cells were filtered. Total gene counts for each cell were normalised to a target sum of $10^4$ and log1p transformed. This resulted in a working dataset of 80,556 cells. Next, highly variable features were selected based on a minimum and maximum mean expression of ≥ 0.0125 and ≤ 3 respectively, with a minimum dispersion of 0.5. Total feature counts and mitochondrial percentage were regressed. The number of principal components used for neighbourhood graph construction was set to 50 initially, and subsequently 30 for myeloid and TIL subgroup processing. Clustering was performed using the Leiden algorithm with resolution set at 1.5 for initial annotations, but subsequently sub-clustering was performed at lower resolutions (0.7–1.0) for analysis of myeloid cell, DC, T cell, and CD8[+] T cell subsets. Uniform manifold approximation and projection (UMAP, v0.5.1) was used for dimensional reduction and visualisation, with a minimum distance of 0.3, and all other parameters according to the default settings in Scanpy.

### Analysis of scRNA-seq from murine tumours

Broad cell types of interest were subset (eg. DCs from myeloid cells) and re-clustered as described above. Resulting clusters were annotated using canonical marker gene expression and published transcriptomic signatures. Unless otherwise indicated, log-transformed expression values were used for plotting. Gene set scoring was performed using Scanpy's tl.score_genes tool. Gene sets were obtained from the Molecular Signature Database (MSigDB) inventory, specifically Hallmark, KEGG, or Gene Ontology (GO), using the R package msigdbr (v7.5.1) or published RNAseq data (Supplementary data 1). CCR7[+] DC signature genes used for scoring and identification of activated DCs were obtained from a published tumour mRegDC signature gene list[5], for both murine and human data. Differential gene testing was performed using the Wilcoxon rank-sum test with Benjamini-Hochberg multiple-testing correction, implemented in Scanpy's tl.rank_genes_groups. Analysis of LN scRNA-seq, B16-F10 tumours and their dLNs followed the same methods. Gene regulatory network and transcription factor regulon activity analyses was performed in pyScenic (v0.12)[68].

Trajectory analysis was performed using partition-based graph abstraction (PAGA, implemented in Scanpy v1.8.2)[69] and the Palantir algorithm (v1.0.0)[70]. For Palantir pseudo-time analysis, the differentiation trajectory was rooted in the Kaede-green dominant cluster, which represents the cellular subset most associated with newly-infiltrating cells. The specific root cell was selected based on the extrema of diffusion components and no end-points were specified. RNA velocity analysis was performed using velocyto (v0.17.15) scVelo (v0.2.5)[71], following the default pipeline. Integration and label transfer of lymph node DC scRNA-seq data and tumour DCs, or cycling CD8[+] T cells and non-cycling CD8[+] T cells was performed using Scanpy's tl.ingest tool. For analysis of cycling CD8[+] T cells, cell cycle regression was first performed on the isolated cluster using Scanpy's pp.regress_out function, before re-integration.

Gaussian kernel density estimation to compute density of cells in the UMAP embedding was performed using Scanpy's tl.embedding_density. Differential abundance analysis of $k$-nearest neighbour (kNN) defined cellular neighbourhoods was performed using Milo (v0.1.0)[72]. Spcifically, the kNN graph was constructed with a $k$ parameter of 30 and initial random sampling rate of 0.1. Cellular neighbourhoods with constituent cells comprising less than 70% of a previously defined cluster were designated as mixed neighbourhoods. CellPhoneDB[73] (v2.1.7) was used for cell-cell communication analysis, using the default parameters and normalised expression values as the input. CellPhoneDB output was visualised using ktplots (https://github.com/zktuong/ktplots).

For gene set enrichment analysis (GSEA), and principal component analysis of cDC2 in the scRNA-seq data, a pseudo-bulk approach was first applied (https://github.com/colin-leeyc/CLpseudobulk), to increase robustness for pathway analysis and overcome limitations associated with differential expression testing and gene ranking of single cell data[74]. Briefly, filtered, raw count data (prior to normalisation, transformation, or scaling) from each group were randomly sorted into pseudo-replicates, with iterative bootstrapping applied to random sampling ($n > 10$). Pseudo-bulked count matrices were normalised using the median-of-ratios method, implemented in DESeq2[75] (v1.34), and differential gene expression testing was performed using the Wald test. Pre-ranked GSEA[76] was implemented in fgsea (v1.24), using the mean Wald statistic over $n > 10$ iterations of random sorting into pseudo-bulked replicates, as the gene rank metric. Leading edge genes were identified for further analysis. GO term over-representation analysis was implemented in topGO[77] (v2.50), using the top 100 genes with the highest loadings for each principal component (PC) of interest, following PC analysis.

### Published sequencing data

Published data was accessed and downloaded from public repositories using the following links or accession codes: scRNA-seq of human CRC (GEO: GSE178341 and Synapse: syn26844071)[26,42]; scRNA-seq of human

breast cancer (https://lambrechtslab.sites.vib.be/en/data-access)[24]; scRNA-seq of human melanoma (GEO: GSE123139)[25]; scRNA-seq of murine B16-F10 melanoma (https://www.ebi.ac.uk/gxa/sc/experiments/E-EHCA-2)[34]; scRNA-seq of human mUC[46]; scRNA-seq and PICseq of human NSCLC (GEO: GSE160903)[38]; TCGA (https://portal.gdc.cancer.gov, via TCGAbiolinks)[23]; METABRIC (https://www.cbioportal.org/study/summary?id=brca_metabric)[27]; IMvigor210 ate-zolizumab trial bulk RNA-seq (EGA: EGAS00001004343)[44]; 10x Visium (https://www.10xgenomics.com/resources/datasets)[37]. Where necessary, data access permission was sought and approved prior to download.

## Analysis of scRNA-seq from human tumours

scRNA-seq data from human solid tumours were downloaded as above. Data was analysed using the Scanpy (v1.8.2) workflow, as outlined in the sections above. Filtering for quality control was performed according to the parameters outlined in the original publications. scRNA-seq integration was performed using the batch-balanced KNN (with ridge regression) (bbknn v1.5.1) or Harmony algorithm (harmonypy v0.0.5) with sequencing batch as the batch term, where available and sufficient, or patient ID if sequencing batch was not available. Cell annotations from original publications were checked and refined, particularly where myeloid cell annotations were not complete or the focus of the original publication.

Myeloid cells were subset and used for further analysis. In total, this included re-analysis of 41,624 and 43,193 myeloid cells from two independent CRC datasets, 16,688 myeloid cells from breast tumours, 8555 myeloid cells from cutaneous melanoma, 11,663 myeloid or T cells from NSCLC including 901 myeloid-T cell doublets, and 1,988 myeloid cells from mUC. CCR7+ DCs were identified using expression of canonical marker genes and a CCR7+ DC signature[5].

The only exception to the Scanpy workflow was for analysis of NSCLC PICseq data, where the MetaCell (v0.3.41) workflow was utilised, as documented in the original publication, for analysis and annotation of single-cells[38]. This was to ensure compatibility with the PICseq deconvolution algorithm[78], which was used to analyse sorted doublets. Gene expression raw count data from PICs and deconvolution results for the contributing myeloid cell or T cell identity in each PIC were obtained, and visualisation and further analysis were performed in Scanpy.

Subsequently, analysis of gene expression profiles, gene signature scoring, cell-cell communication, were as described in earlier sections. For analysis of enrichment of Ccr7_DC.2 genes in the breast cancer dataset, where anti-PD-1 response data was available, Ccr7_DC.2 genes were first identified as DEGs between Ccr7_DC.2 versus remaining Ccr7_DCs (Wilcoxon rank-sum test with Benjamini-Hochberg correction) in scRNA-seq from murine tumours. 85 differentially upregulated genes ($p$-adj < 0.05, $\log_2$Fold-change > 1) were converted to equivalent human gene symbols using Ensembl and used to score CCR7+ DCs from human breast cancer (Fig. 6f).

## Analysis of bulk transcriptomics

Bulk RNA-seq data were downloaded as above. TCGA data was accessed via TCGAbiolinks, using STAR-aligned reads[23]. For bulk transcriptomics data from both TCGA and the IMvigor 210 trial, transcripts-per-million (tpm) normalised values were used for analysis. Cellular deconvolution was performed using xCell[79], implemented in the webtool using default parameters. Gene signatures scoring was performed using single-sample GSEA[80] (ssGSEA, v10.1.0), implemented in GenePattern, followed by normalisation (scaled between 0 and 1) of gene set enrichment scores. Cell type signatures for ssGSEA were derived from cell-specific marker genes in scRNA-seq of human cancers. Pearson correlation was used to assess for correlation of cellular proportions or cell-specific signatures of interests in biopsy samples. Survival analysis was performed on overall survival (months), using the

median CCR7+ DC signature score for stratification to 'high' and 'low' groups, and log-rank test was applied to survival curves. For integration of bulk transcriptomic data from the IMvigor210 trial with accompanying clinical response data with scRNA-seq of myeloid cells in mUC, the Scissor pipeline (v2.1.0) was used[47]. Scissor enables identification of single-cells and gene expression profiles within cell sub-populations that are significantly associated with phenotypes obtained from bulk expression data. A logistic regression model using the binary outcome of clinical responders (CR/PR) vs non-responders (SD/PD) was used, and the alpha parameter was set at 0.05, per default settings.

## Analysis of spatial transcriptomics

10x Visium spatial transcriptomics were analysed using the standard Scanpy (v1.9.3) workflow, using the default SpaceRanger (v1.2.0) outputs including spot alignments to corresponding tissue haematoxylin and eosin or immune-fluorescence images. Spots with fewer < 500 counts, > 80000 counts, and percentage mitochondrial content > 15% were filtered. Genes detected in fewer than 5 spots were filtered. Total feature counts, mitochondrial percentage, and cell cycle scores were regressed out. Expression values plotted are log1p-transformed $10^4$-sum-normalised values. Gene set scoring was performed using Scanpy's tl.score_genes tool. Calculation of spatial correlation was implemented using the correlationSpot function in ktplots (https://github.com/zktuong/ktplots), as previously described[81]. Briefly, k = 6 nearest neighbourhoods were extracted from a kNN graph computed from the spatial location of each Visium spot. Pearson correlation was performed on each neighbourhood using gene expression values or gene signature scores and averaged across the neighbourhoods. Correlation values were only returned if expression value or signature scores were detected in all spots, and above a significance threshold of $p < 0.05$.

## Bone marrow derived dendritic cells

Bone marrow from wild-type C57BL/6, OX40L$^{fl/fl}$, CD11c$^{cre}$ OX40L$^{fl/fl}$, or OX40L$^{+/Hu-CD4}$ reporter mice were isolated by flushing femurs and tibias with RPMI, supplemented with 10% fetal calf serum, 1% penicillin/streptomycin, 1% L-glutamine and 1% sodium pyruvate (cRPMI). Bone marrow cells were strained through a 70 μm filter, centrifuged, and resuspended in RBC lysis buffer for 2 min on ice. Cells were plated in cRPMI in tissue-culture-treated 10 cm dishes. 20 ng/mL murine recombinant GM-CSF (Peprotech) and 5 ng/mL murine recombinant IL-4 (Peprotech) was added to generate bone-marrow derived dendritic cells (BMDC). Half of the medium was removed on day 2 of differentiation and new pre-warmed medium supplemented with GM-CSF and IL-4 (2X concentrations) were added. The culture medium was entirely replaced on day 3 with fresh warmed cRPMI + GM-CSF (20 ng/mL) only. On day 6, non-adherent cells in the culture supernatant were harvested for tumour cell-line co-culture.

## In vitro cultures

Ovalbumin-expressing MC38 (MC38-Ova) cells were plated and incubated in tissue-culture-treated dishes in FCS-supplemented DMEM media ( + 5 mM HEPES), as above, and allowed to settle. 24 h after (80% confluence), the MC38-Ova monolayer was exposed to UV light (302 nm, 30 s, 200 J m$^{-2}$ s$^{-1}$) to induce apoptosis, and further incubated for 24 h. Day 6 BMDC were added to apoptotic MC38-Ova cells, and non-adherent cells were collected after 8 h of co-culture for analysis by flow cytometry or further application. Either $1.5 \times 10^5$ MC38-Ova cells were plated in 24-well plates and $0.5 \times 10^5$ BMDC were added 24 h after MC38-Ova irradiation, or $4.5 \times 10^6$ MC38-Ova were plated in 10 cm dishes and $1.5 \times 10^6$ BMDC were added. Where indicated, 10 ng/mL recombinant murine interferon gamma (Peprotech) or 25 μg/mL of anti-PD-L1 antibodies or isotype control antibodies were added to the tumour-DC cultures for 8 h.

For OT-I T cell co-culture, 30 s UV exposure was used to induce MC38-Ova apoptosis for culture with BMDCs. Naïve CD8+ OT-I T cells

(Live CD3$^+$CD8b$^+$CD62L$^+$CD44$^-$) were FACS sorted from the spleens of OT-I × *Rag1*$^{-/-}$ mice. Tumour antigen-experienced BMDC (Live CD45$^+$CD11c$^+$MHC-II$^+$CCR7$^+$PD-L2$^+$) were FACS sorted from MC38-Ova tumour-DC cultures after 8 h, as above. Sorted naïve OT-I T cells were labelled with CellTrace Violet (CTV, Thermo Fisher) at 37 °C for 20 min, washed, and counted before use. 30,000 BMDC were incubated with 120,000 OT-I T cells in cRPMI in a 48 well tissue culture plate. For a positive control for OT-I stimulation, day 6 BMDCs were incubated with endotoxin-free ovalbumin protein (100 μg/mL) and LPS (20 ng/mL), before FACS sorting and culture with OT-I cells. For negative controls, BMDCs stimulated with apoptotic MC38 cells (not expressing ovalbumin antigen), OT-I cultured with ovalbumin and LPS (no DC), or OT-I only were used. Anti-PD-L1 antibodies or isotype control antibodies were added on day 0 and day 2 as indicated. T cell activation and proliferation was assessed by flow cytometry after 72 h. For IFNγ pre-treatment of BMDC, 10 ng/mL recombinant murine interferon gamma (Peprotech) was added to tumour-DC cultures for 8 h, washed twice to remove free cytokine, and FACS sorted, before culturing with OT-I cells. For analyses where data from multiple independent experiments were combined, values were normalised to respective replicates of cultures containing isotype control antibodies and OX40L-expressing DCs (baseline), to facilitate comparisons across experiments.

### Ex vivo cultures

For ex vivo tumour cultures, tumours were digested as described above, and centrifuged in 30% Percoll for 20 min to remove tissue debris, prior to dead cell removal. Tumour cell suspensions were resuspended in pre-warmed cRPMI and split equally into 3 wells for paired stimulation, to enable intra-tumour comparisons. Ex vivo cultures were performed at 37 °C in 48w plates with 300 μL per well containing 10$^7$ cells / mL. Isotype control or anti-PD-L1 antibodies (25 μg/mL each), or recombinant murine interferon gamma (10 ng/mL, Peprotech) was added for 8 h. Cells were collected, culture plates washed twice with ice-cold PBS supplemented with 2 mM EDTA and 10% FBS, and filtered before analysis by flow cytometry.

### Statistical analysis

Mice were age and sex-matched. Analysis for RNA sequencing is as described above, including statistical frameworks used. Statistical tests were implemented for two principal purposes; to compare expression values (RNA or protein) between samples, or to compare proportions of defined cell subsets. All experiments were performed with biological replicates, and the specific statistical tests applied are indicated in the figure legends. All statistical tests applied were two-tailed. Paired statistical tests were used where different populations from the same animal were compared. Statistical analyses were performed in R (v4.0.4) or GraphPad Prism (v10.0.3). Exact p-values are shown unless $P > 0.05$ (not significant, ns).

### Online supplementary material

Supplementary Figs 1–12. Supplementary table 1 list antibodies and reagents used. Supplementary data 1 list gene sets used for enrichment scoring. Supplementary data 2 contain cluster DEGs in scRNA-seq of tumour DCs (Fig. 1).

### Reporting summary

Further information on research design is available in the Nature Portfolio Reporting Summary linked to this article.

## Data availability

The scRNA-seq data generated in this study have been deposited on the GEO repository under accession numbers GSE221513 and GSE221064. Source data are provided with this paper.

## Code availability

All analysis performed are described under *Methods*. Source code for softwares or computational pipelines used are referenced. Additional code used for analysis, as described above, have been made available on GitHub repositories (https://github.com/zktuong/ktplots, https://github.com/colin-leeyc/CLpseudobulk).

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

## Acknowledgements

CYCL is funded by the Gates Cambridge scholarship trust and University of Cambridge School of Clinical Medicine Elmore fund. BCK was supported by Cancer Research UK (SEBSTF-2021/100002). ID was supported by an iCASE Studentship with AstraZeneca. DRW was funded by Cancer Research UK Immunology Project Award C54019/A27535, Cancer Research Institute CLIP grant CRI3128, Worldwide Cancer Research Grant 21-0073 and a Medical Research Council IMPACT iCASE Studentship with AstraZeneca. MRC is supported by the National Institute of Health Research (NIHR) Cambridge Biomedical Research Centre and the NIHR Blood and Transplant Research Unit and a Wellcome Investigator Award (220268/Z/20/Z). We thank A. Ptasinska and other members of Genomics Birmingham at University of Birmingham for their help with single-cell RNA-sequencing experiments, and the University of Birmingham Flow Cytometry Facility. We thank Dr. Y. Miwa (Tsukuba University, Tsukuba, Japan) and Dr. O. Kanagawa (RIKEN Research Center for Allergy and Immunology, Yokohama, Japan) and Dr. M. Tomura (Osaka Ohtani University, Tondabayashi, Japan) for the Kaede mice. We thank the Mary Lyon Centre at MRC Harwell for OX40L reporter mice used in this study (award MC_UP_2201/2). We thank Marina Botto (Imperial College London) for the OX40L^fl/fl mice. Icons in Figs. 1a, 5d and g were created with BioRender.com.

## Author contributions

C.Y.C.L. designed and performed experiments, analysed data, performed computational analyses, and wrote the manuscript. B.C.K. and I.D. performed experiments and analysed data. Z.K.T. and T.H. provided project guidance. N.R., F.G., Z.L., C.W., S.W. and D.P. performed experiments. G.C., S.A.H. and S.J.D. reviewed the manuscript. R.R. supported the project. D.R.W. conceived the project, designed experiments, provided guidance, and edited the manuscript. M.R.C. conceived the project, designed experiments, provided guidance, and wrote the manuscript.

## Competing interests

AstraZeneca provided therapeutic anti-PD-L1 antibodies, isotype control antibodies, and MC38-Ova cells. GC, SAH and SJD are full employees and shareholders in AstraZeneca. No other disclosures or conflicts of interest are reported.
