## [Peer Review File · Nature Communications]

This manuscript has been previously reviewed at another journal that is not operating a transparent peer review scheme. This document only contains reviewer comments and rebuttal letters for versions considered at *Nature Communications*.

Response letter to review #1 for:
**Time-, tissue- and treatment-associated heterogeneity in
tumour-residing migratory DCs**
Colin YC Lee *et al.*

Accompanying documents:
Manuscript text; revised v2
Figures; revised v2

Reviewer #1:

Although the existence of a conserved DC subset with maturation, migration and regulatory features in human and mouse tumors, named mRegDCs (also known as LAMP3+ DCs, mregDCs or DC3s), has been described in many studies, their contribution to anti-tumor immunity remains controversial. In this manuscript, using photoconvertible mice and single-cell sequencing, the authors describe the intrinsic heterogeneity of migratory capacity and tissue distribution of mRegDCs for the first time. The authors highlight that tumor-resident mRegDCs exhibit a more exhausted phenotype compared to migratory mRegDCs, and that PD-L1 treatment attenuates this exhaustion feature. Although some conclusions require more robust experimental support, this manuscript still provided meaningful insights to address the functional heterogeneity of mRegDCs.

We thank the reviewer for their careful consideration of the manuscript, and their very helpful suggestions. We believe that their comments have helped to improve the manuscript significantly. Please see below our point-for-point response to each suggestion, including revisions to the manuscript. Major revisions to the manuscript text have been highlighted in yellow.

Major suggestions:

1. In Fig. 1, CD11b+ Myeloid cells were sorted for unbiased clustering analysis. However, CD11b expression was heterogeneous across DC subsets, where pDCs and cDC1s only express low or no CD11b, while cDC2s universally express CD11b. Also, whether mouse mRegDCs uniformly express CD11b still needs to be confirmed. It is certain that this sorting strategy can lead to the absence of particular DC subsets, hence misleading the subsequent analysis of transcriptomic data.

The reviewer raises an important point with regards to CD11b expression on cDC1 vs cDC2. However, this comment arises from a lack of clarity in our original text/methods, for which we apologise. In fact, the scRNA-seq experiment was performed on **all CD45+** cells. To obtain information on Kaede fluorescence, we performed FACS sorting on tumour immune cells prior to scRNA-seq. Since FACS was necessary, we also sorted for CD11b+ and CD11b- fractions (**Ext. Data Fig. 2B**), so that we could obtain adequate representation of all immune cells (since CD11b+ cells are much more frequent in MC38 tumours). However, during analysis, all cells were 're-combined' to form the scRNA-seq data object. Myeloid cells and subsequently DCs were identified from the scRNA-seq solely based on transcriptional profiles, meaning that both CD11b+ and CD11b- cells are represented in the DC data shown in Figure 1. In addition, we have included new data on surface expression of CD11b on tumour mRegDCs in response to suggestion 4 (see below); we hope this addresses the reviewer's concerns regarding CD11b expression on mRegDC.

We have now edited **Ext. Data Fig. 2B**, the **Results** text and **Figure legends** to state that scRNA-seq was performed on Kaede⁺CD45⁺Ter119⁻ cells. In addition, we have included a detailed explanation under **Methods**, to highlight that all CD45⁺ cells, regardless of surface CD11b or NK1.1 expression, are included in the analysis.

2. The authors defined 3 mRegDC subsets based on transcriptome features but did not provide distinct and reliable markers or transcription factors to distinguish them. Their conservation and interconversion should be characterized by *in vitro*, *ex-vivo* or adoptive transfer experiments.

We thank the reviewer for their comments and have performed additional experiments and analysis to characterise mRegDC heterogeneity. Importantly, we believe that it is more appropriate to refer to the clusters in the scRNA-seq data as variable mRegDC **states** rather than distinct **subsets**, because the expression profiles of the 3 clusters are part of a continuous spectrum, and there is no binary expression of specific markers that definitively separate the mRegDC states into discrete subsets. We have now checked and edited the manuscript text, where applicable, to ensure that the clusters are referred to as mRegDC states.

Transcription factors: We investigated transcription factors (TFs) that were differentially expressed between newly-formed mRegDC_1 and terminal mRegDC_2/3. In addition, since TF expression does not reflect TF activity *per se*, we further performed gene regulatory network analysis using SCENIC (Aibar et al., 2017), to compute TF regulon activity scores and identify congruent changes with TF expression. This led to the identification of *Tcf7*/Tcf7 regulon which was particularly enriched in Kaede-red tumour-retained mRegDCs. While the role of TCF-1 (*Tcf7*) is well described in memory or stem-like T cells, their activity has not been described in myeloid cells, and warrants investigation in future work. This data has now been included in **Ext. Data Fig. 5E**.

Surface markers: CD80 and CD86 expression differs between mRegDC that have migrated to the dLN, which resemble mRegDC_1 (**Fig. 2I**), and tumour-mRegDCs; and is also different between newly-formed Kaede-green mRegDCs (representing mRegDC_1) and the terminal/resident Kaede-red mRegDC states (representing mRegDC_2/3). This data (below) has now been added to **Fig. 2H** and the accompanying **Results** text, and is further elaborated below in response to comment 6. Importantly it demonstrates variable co-stimulatory marker expression relating to time and tissue, supporting the presence of heterogeneous mRegDC states in the tumour.

We further assessed tumour DCs using OX40L^{Hu-CD4} (*Tnfsf4*)-reporter mice newly generated for these revisions, since *Tnfsf4* is higher on tumour-retained mRegDC, particularly mRegDC_2. This

data is shown below and is now included in **Ext. Data Fig. 6F-G / Fig. 3H**. This data confirms that a fraction of tumour mRegDCs express OX40L, but is not expressed by cDC or migrated mRegDC in the dLN, further supporting the presence of variable mRegDC states. Importantly, the absence of OX40L expression in the dLN implies that this molecule is only upregulated by CCR7+ DCs that fail to egress, and that OX40L+ mRegDC do not emigrate to the dLN, or at least rapidly downregulate it upon tumour exit.

Interconversion between states: The conversion of mRegDC_1 to mRegDC_2/3 is part of the maturation trajectory that occurs naturally in tumours and is discussed in response to comment 4 and 6 below, and differences between newly-formed and more terminal mRegDC have been highlighted above. We sought to further investigate the conversion to terminal mRegDC_2 or mRegDC_3 states using an *in vitro* system, as the reviewer suggested, where mRegDC were treated with various factors in isolation to identify a specific mechanism that may underlie acquisition of various states. Of note, these *in vitro* experiments are elaborated in detail in response to comment 14. mRegDC were generated from BMDC by tumour cell-line co-culture, and isotype-control, anti-PD-L1 antibodies, or IFN γ was added. We wondered whether anti-PD-L1 antibodies directly affected mRegDCs, or whether some DC-extrinsic factor was responsible. Some of the transcripts expressed by the activated mRegDC_2 state are regulated by IFN γ (based on literature), so we tested our hypothesis that administration of recombinant IFN γ (rIFN γ) can induce the mRegDC_2 state which is enriched following anti-PD-L1 treatment *in vivo*. The data (below) showed that IFN γ increased expression of CD40, OX40L and PVR, which are expressed by mRegDC_2 (**Ext. Data Fig. 6E**). Treatment of dissected tumours *ex vivo* with rIFN γ resulted in similar activation. This data has now been added to **Fig. 5J / Ext. Data Fig. 10I**. Of note, mRegDC pre-treated with IFN γ facilitated increased activation and expansion of antigen-specific CD8 $^+$ T cells (see comment 14). A possible source of IFN γ *in vivo* are the CD8 $^+$ T cells which mRegDC co-localise with (**Fig. 5E, Ext. Data Fig. 9E-I**), particularly given that CD8 $^+$ T cells increase IFN γ production in anti-PD-L1-treated tumours (**Ext. Data Fig. 8H**).

3. On page 5, line 130, the proportion of mRegDCs to tumor tumor-infiltrating DCs should be

further confirmed by flow cytometry, given that the absence of CD11b-DCs may have been artificially introduced in the single-cell sequencing data.

As clarified in our response above (comment 1), we thank the reviewer for this comment, but the absence of CD11b- DCs were not artificially introduced to the single-cell data. In addition, we have performed the analysis that the reviewer has requested, and compared the proportion of mRegDC to cDC1 and cDC2 by flow cytometry. The data is shown below and is now added to **Ext. Data Fig. 2G**.

4. On page 5, line 133, The majority of mRegDCs expressed IRF8, presuming their cDC1 origin. This is inconsistent with the results of pseudotime analysis in Fig. 1I and Fig. S3C, where mRegDCs and cDC2ss appear to have a closer developmental correlation.

We thank the reviewer for their comment on the relationship between cDCs and mRegDCs. We recognise that there is room to address this with increased clarity in our data and have now done so with the following experiments/analyses.

First, to prove that mRegDC indeed arise from cDC, we decided to examine timepoints very early after photoconversion (i.e. 5 hours). The hypothesis was that, if indeed mRegDC originate from cDC, most of the Kaede-green DCs 5 hours after photo-conversion would be cDC, because newly-infiltrated DCs in this timespan would have had little time to acquire the mRegDC programme, but would be higher at later timepoints. Indeed, we found this to be true. At 5h post-photoconversion, the Kaede-green mRegDC:cDC ratio was approximately 1/100, but increased to 1/5 72h post-photoconversion. Hence, mRegDC emerge only after cDC influx. The data is shown below and is now included in **Ext. Data Fig. 3C**.

Second, we do not believe that our data support that mRegDC and cDC2 have a closer developmental relationship. While it is true that in the UMAP embedding, mRegDC are embedded closer to cDC2 than cDC1, the UMAP embedding is computed based on neighbourhood graph, and may imply transcriptional similarity but not developmental relationships. Conversely, if a correlation matrix computed on the DC clusters, with hierarchical clustering, it is evident that cDC1 are more

‘similar’ to mRegDC than cDC2 (see below). Of note, in **Fig. 11**, cDC1 have similar pseudotime values to cDC2_2 and cDC2_3.

Indeed, difficulties in transcriptionally distinguishing mRegDC based on cDC1/2 origin is recognised (Gerhard et al., 2021; Ginhoux et al., 2022; Maier et al., 2020). Specifically, upregulation of the common mRegDC programme upon DC maturation, and downregulation of cDC1- or cDC2-associated transcripts make it difficult to tell cDC1- or cDC2- derived mRegDC apart transcriptionally, but they may maintain some surface expression of cDC1/2 markers. Hence, we chose to address the question of mRegDC origin by investigating surface XCR1 or CD11b expression (data below). These data indicate that XCR1+ and CD11b+ mRegDC both exist, where XCR1+ mRegDC are enriched in the Kaede-red fraction. Hence, we concluded that mRegDC arise from both cDC1 and cDC2, and similar to tumour cDC2:cDC1 ratios, CD11b+ mRegDC are more frequent than XCR1+ mRegDC. This conclusion could not be made based on the transcript data alone, and is now included in **Ext. Data Fig. 3D**.

mRegDC express high levels of *Irf8* but lower levels of *Irf4* (below), and as the reviewer points out, we highlighted *Irf8* expression in mRegDCs as an indication of cDC1 origin in the original manuscript submitted. However, in light of this new surface phenotyping, it is likely that a large proportion of tumour mRegDC are cDC2 derived, and that *Irf8* may be expressed as part of the mRegDC programme. Hence, we have removed this sentence in the revised manuscript, instead highlighting the surface phenotyping above to understand mRegDC ontogeny.

In summary, our data supports that mRegDC arise from both cDC1 and cDC2, and more mRegDC in the tumour are cDC2 derived, as indicated by surface cDC1/2 marker expression. The revised manuscript now has a new section in **Results** addressing cDC to mRegDC maturation, and mRegDC ontogeny.

5. In Fig.2E, as mentioned above, CD11b-DCs may be lost in the single-cell sequencing data. It is difficult to evaluate whether mRegDCs are the majority of DCs arriving in dLN, and more experiments are needed.

As discussed above (response to comment 1), in fact, all CD45⁺ cells from the tumour were sequenced and analysed, and in the scRNA-seq data for LN, Kaede-red CD45⁺ cells were sorted to comprehensively track all CD45⁺ cells that have migrated from the tumour to the dLN (stated in **Results** and **Methods**). Hence, CD11b- DCs are included in our analyses. We apologise for the lack of clarity in our initial methodological descriptions that led to this misunderstanding.

We have now also included new flow cytometric data (**Fig. 2C**, below) which shows that among Kaede-red DCs in the dLN (which were photo-labelled in the tumour, and hence tumour emigrants), essentially all are mRegDCs, and few “immature” cDC successfully migrate to the dLN in the absence of CCR7 expression. This is in line with the current dogma that CCR7 expression drives homing to the LN. Hence, both the flow cytometry data (**Fig. 2C**) and scRNA-seq data (**Fig. 2E**) support mRegDCs as the major DC subset arriving at the dLN.

6. In Fig. 1I, Fig. S3C and Fig. 3, based on pseudotime analysis, the authors speculated a differentiation trajectory of mRegDC_1 to mRegDC_2/3. These results indeed revealed a transcriptomic heterogeneity of mRegDC_1/2/3, but more convincing evidence is needed to demonstrate the existence of the transition from mRegDC_1 to mRegDC_2 or mRegDC_3.

We thank the reviewer for this comment, and offer the following evidence to further support the transition from cDC to mRegDC_1 to mRegDC_2/3.

First, we performed RNA velocity analysis (see below), which computes trajectories based on splicing dynamics rather than gene expression (as used in the pseudotime and PAGA analysis in the original version of the manuscript). This analysis clearly supports our existing proposed trajectory, where mRegDC_1 are an intermediate state on the transition to mRegDC_2 or mRegDC_3, which are terminal tumour mRegDC states. This analysis is now included in **Fig. 1J** and **Ext. Data Fig. 3B**.

Next, we have obtained new phenotypic data to support this trajectory. *Cd80/86* expression is upregulated along the purported trajectory from mRegDC_1 to mRegDC_2/3 (Ext. Data Fig. 4B). In line with this, CD80/86 expression higher on Kaede-red mRegDC than Kaede-green mRegDC, consistent with an upregulation of CD80/86 with tumour dwell time. Hence, increased CD80/86 expression marks the transition from mRegDC_1 to mRegDC_2/3. This data is shown below and now included in Fig. 2H. Of note, migrated mRegDC, which resemble mRegDC_1 transcriptomes (Fig. 2I), express more similar levels of CD80/86 to Kaede-green mRegDC, so this data is in support of our hypothesis that newly-formed mRegDCs are the most likely tumour emigrants.

7. The gating strategy in Fig. 2A should be described. Even if most of the non-DC population has been excluded as in Fig. S2H, it is hasty to simply assume PD-L2+ cells as mRegDCs, and more co-stimulatory molecules and CD11c should be introduced to verify their identity. In addition, a substantial fraction of PD-L2+ cells appear to be CCR7-, which requires more stringent isotype to confirm, and more evidence is needed on whether they belong to mRegDCs.

We thank the reviewer for these comments and have performed the requested revisions.

The gating strategy for identification of DCs from tumours (Fig. 1F) and LNs (Fig. 2A) are the same and has been included in Ext. Data Fig. 2F. The legend for this figure has been updated to reflect this.

The gating strategy has also been altered from the initial submission. Per the reviewer's request, CD11c⁺MHC-II⁺ has now been introduced to the gating strategy as a pan-DC gate, prior to identification of DC subsets (below, new gating). Hence, all cDC and mRegDC are Live, CD45⁺ Lin⁻ (CD3, NK1.1, B220, Ly6G, F4/80) CD11c⁺MHC-II⁺. All relevant flow cytometry has been re-analysed according to the new gating strategy. This did not change our conclusions.

With regards to mRegDC identity, we have further optimised CCR7 surface antibody staining for flow cytometry, but despite this, some PD-L2⁺ DCs remain negative for CCR7. It is difficult to tell if this is an issue with the antibody, since chemokine receptors are often difficult/unreliable to identify by conventional antibody staining. However, given the reviewers concerns, we have revised the gating strategy (new gating, row 3 panel 1), such that only PD-L2⁺CCR7⁺ double-positive cells are considered mRegDC. With this stricter gating strategy, we can be confident that these cells belong to mRegDC. All data in the manuscript has been updated to this gating strategy.

Of note, we re-analysed our earlier experiments to see if gating mRegDCs using CCR7+PD-L2+ or PD-L2+ only made a difference in terms of Kaede-red frequency. As seen below, the proportion of Kaede-red DCs is the same, regardless of the gating strategy in the early experiments, supporting PD-L2+ DCs as a major Kaede-red DC subset.

8. In Fig.2F-H, the transcriptomic differences between tumor-resident mRegDCs and dLN mRegDCs may primarily reflect the inherent differences between mRegDC_2/3 and mRegDC_1, as they are enriched in tumors and dLN, respectively. In contrast, more than 20% of mRegDC1 resides in tumors, and comparing the transcriptomic differences between tumor-resident mRegDC_1 and mRegDC_1 in dLN will better clarify whether migratory mRegDCs and tumor-resident mRegDCs are distinct.

We thank the reviewer for this comment. The data requested is included in **Ext. Data Fig. 4B** (below), where gene expression between tumour mRegDCs and dLN DCs are compared. Several

molecules of interest are expressed at an intermediate level in mRegDC_1 – for example, expression of *Cd80* and *Cd86* is lower on mRegDC_1 than mRegDC_2/3, but higher than mRegDC in the dLN. This is consistent with the reviewer’s comment and our hypothesis that mRegDC_1 are an intermediate state that may become tumour-resident mRegDC_2/3 or migrate to the dLN. In addition, please see our above response to comment 6, specifically, comparison of surface CD80/86 expression between dLN mRegDCs and Kaede-green/red tumour mRegDCs. These data highlight differences between mRegDC_1 and mRegDC_2/3 as well as migratory versus tumour mRegDC, and demonstrate that CD80/86 upregulation is part of the distinct tumour-resident mRegDC phenotype, consistent with the scRNA-seq data.

9. In Fig. 3D-E, the conclusion that "Tumour-retained mRegDCs progress towards an "exhausted" state" is proved by comparing mRegDC_1 and mRegDC_3, but mRegDC_3 should also be considered, since they appear to be tumour-retained. In addition, as mentioned above, mRegDC_1 does not always appear to be migratory, thus comparing the differences between mRegDC_1 in tumors and in dLN will help to clarify whether tumor-resident mRegDCs exhibits exhausted phenotype.

We thank the reviewer for this comment, and we assume the reviewer meant "..., but **mRegDC 2** should also be considered". In any case, while in the earlier version of our manuscript we had discussed mRegDC_2 later in figure 5, we have now brought this data forward to figure 3, per the reviewer’s request. Hence, in the revised manuscript, all three tumour mRegDC subsets in our data are now considered in **Fig. 3**. In response to the reviewer’s second point, mRegDC_1 are early in the proposed maturation trajectory; hence, while some cells in this cluster resemble tumour-retained mRegDC (ref. data in response to comment 8), they are likely on route to attaining features of "exhaustion", which we observe later in the maturation trajectory. Indeed, as shown in **Fig. 3B**, mRegDC_1 are more enriched for antigen presentation genes compared to mRegDC_2/3, but the scores are bimodal, suggesting that a fraction are starting to downregulate expression.

10. In Fig. 4, the authors emphasized the interaction between mRegDCs and effector T cells, but in Fig. 5, PD-L1 treatment instead mainly affected mRegDC-Tex cell interaction. Does this mean that mRegDCs selectively interacted with different T cell populations before and during treatment, respectively? Could the authors clarify the mechanisms involved?

We thank the reviewer for this comment and apologise for the lack of clarity. We do not have evidence that mRegDCs interact with *different* T cell populations before and during treatment, and have not made this claim. Instead, it is the nature of their interactions with CD8 T cells that change with treatment by means of ligands expressed on mRegDC. However, we realise that our decision to focus on Tex cells may be the source of this confusion, which we apologise for. We had initially

decided to focus on Tex cells given that this cluster are major responders to xPD-L1, but acknowledge that the omission of other subsets may not be ideal. We have now added analysis considering interactions between mRegDCs and other effector CD8 T cell populations (**Ext. Data Fig. 9**). Our further response to this comment is also of relevance to the reviewer’s comment 13, and is elaborated below.

11. In the spatial transcriptome data, mRegDC-CD8+/T cell interaction is concentrated in a few niche-like regions. It is desirable to determine if these locations are TLSs or TLS-like regions, as it has been reported that mRegDCs in tumors tend to be enriched in TLSs. If so, it seems inappropriate to compare this interaction with tumor-resident mRegDCs in mouse models, as TLSs are quite rare in mouse subcutaneous tumors.

We thank the reviewer for this comment on TLS – the presence of mRegDC in TLS (Discussion; refs 12, 15, 19) is something we have given extensive thought to, and definitely in the scope of future work, especially since our Kaede model enables the distinction of ‘true’ tumour-residing mRegDC.

TLS in cancer is defined by a concentration of B cells at its core (Schumacher & Thommen, 2022). Below, we show new data on enrichment of B cell transcripts and TLS-associated transcripts/molecules such as *CR2*, *FCER2* (CD23, follicular DCs), *BCL6*, *CXCL13*, *LYVE1*, *LTBR* (LTβ-receptor), in the CRC spatial transcriptomic data (from Fig 4). From this data, it is evident that: while some mRegDC-CD8 T cell hotspots co-localise with B cells and TLS transcripts, and may represent TLS, there are many hotspots where B cells or other TLS transcripts are absent. Hence, mRegDC may interact with CD8 T cells within TLS, but also outside. Indeed, while mRegDC have been identified in TLS, there is no existing data to suggest they are restricted only to TLS.

While it is true that TLS are rare in murine subcutaneous tumours, these models are still useful to investigate immune cell interactions. Indeed, we have shown that similar interactions exist between mRegDC and CD8 T cells in humans and mice, and likely influence T cell function in similar manners. Moreover, to our knowledge, there is little evidence that the *location* at which DCs and T cells interact alters the molecular function of PD-1, TNFSF or PVR-mediated interactions, etc., but the conservation in molecular interactions between humans and mice provide a powerful opportunity to test novel interventions.

We have not included this spatial transcriptomic data in the revised version of the manuscript as we believe it is beyond the primary scope. We have included new text to highlight the need to investigate mRegDCs in TLS in the Discussion. However, should the reviewer/editor advise otherwise, we are happy to reconsider.

12. In Fig. 5B and Fig. S6D-F, the differences between mRegDC_2 and mRegDC_3 do not seem to represent alterations in the mRegDC transcriptome before and after PD-L1 treatment, as the differences in several immune activating gene signatures between mRegDC_2 and mRegDC_3 appear to exist prior to PD-L1 treatment, such as Nfkb1 showed in Fig. S6C. In addition, PD-L1 treatment does not seem to lead to a greater enrichment of inflammatory or IFN responsive genes in mRegDC_2 than in mRegDC_3, as shown in Fig. S6B. Furthermore, the differences between mRegDC_2 and mRegDC_3 do not suggest that PD-L1 treatment increased the "activation" signatures of tumor-resident mRegDCs, as they are both enriched in the tumor. Whether PD-L1 treatment alters the abundance of each mRegDC subset in the tumor microenvironment should be examined. Also, it would be more convincing to compare the alterations in each mRegDC subset before and after PD-L1 treatment (e.g. Fig. S6C), or the differences between mRegDC_2/3 and dLN-enriched mRegDC_1 after treatment.

We thank the reviewer for the helpful comment. We had performed both differential expression and differential abundance analysis (*whether PD-L1 treatment alters abundance of mRegDC subsets in the TME*) in the manuscript, and apologise that this was not described with sufficient clarity.

We first performed differential expressed gene analyses between anti-PD-L1 and isotype-control treated tumours for each mRegDC subset in our data (**Ext. Data Fig. 6A-B**). and as the reviewer points out, there are similar patterns of enrichment and similar genes that are enriched in all 3 clusters (eg. Hallmark inflammatory response), albeit to different extents.

However, as our study focusses on heterogeneous mRegDC states, we turned our attention to differential abundance analyses. Assessment of abundance in treated vs untreated samples using kernel density embeddings (**Fig. 1E**) and MiloR (**Fig. 3F**) both show that there is an increased abundance of mRegDC_2 following anti-PD-L1, at the expense of mRegDC_3. This observation is now further supported by our RNA velocity trajectory analysis (below), which shows that there is a skew towards maturation to the mRegDC_2 state, and is now included in the manuscript (**Ext. Data Fig. 6C**).

The increased abundance of mRegDC_2 is important given the transcriptionally activated state of this subset, as we highlight in **Fig. 3G**. We acknowledge the reviewer's comment that the mRegDC_2 state is present at baseline, suggesting that some tumour-intrinsic cue is responsible for maintaining this state. We have new reason to believe that this state is in part driven by IFN γ (this data is part of our *in vitro* experiments, elaborated below in response to comment 14). In addition, we have selected some of these mRegDC_2-associated genes (*Tnfsf4*, *Tnfsf9*, etc.) for comparison, per the reviewer's request, and showed that these are not changed within mRegDC_2 following treatment (now included in **Ext. Data Fig. 6B**, below). These data suggest that anti-PD-L1 treatment is associated with independent changes to gene expression and cell state abundance.

We have now rewritten the manuscript text for increased clarity, and to include the new data. We hope that these clarify some of the comments raised.

13. As shown in Fig. S7C, PD-1 is expressed in Activated_CD8T, Stem-like_CD8T, and Tex_CD8T, so the enhanced proliferation and effector function detected in Fig. S7H may not represent the effects of PD-L1 treatment on Tex_CD8T. In addition, the increased proportion of Tex after treatment may also be attributed to the proliferation and differentiation of stem-like CD8+ T, as widely reported. Therefore, whether mRegDCs primarily interact with activated_CD8T, stem-like_CD8T or Tex_CD8T during PD-L1 treatment is not clear and should be evaluated.

We thank the reviewer for raising a very pertinent comment on the potential interactions with other PD-1 expressing CD8 T cell types. As the reviewer points out in comment 10, in **Fig. 4** (human cancers) mRegDC may also interact with other subsets beyond Tex. We had initially decided to focus primarily on interactions with Tex cells in the scRNA-seq data, given that this cluster are major responders to xPD-L1, and the importance of “reinvigorating” exhausted T cells in immune checkpoint therapy. Nevertheless, we acknowledge the need to consider other PD-1 expressing T cell subsets which may be pertinent to PD-L1-PD-1 blockade, and new data supporting the role of stem-like cells in CD8 T expansion following checkpoint blockade.

We have therefore re-performed this analysis and now include data on communication between DCs and all PD-1-expressing / effector CD8 T cells, which includes stem-like CD8 T cells. This data is now in **Ext. Data Fig. 9B-C** (below), along with revisions to the **Results** text to clarify the reviewers concerns and emphasise interaction with other effector subsets. Specifically, we have rewritten the **Results** text to include interactions with all effector/PD-1-expressing T cells rather than Tex only.

Moreover, the new analysis would indicate that mRegDCs interact with Tex, Stem-like, and activated CD8 T cells via similar molecules, and have a similar number of significant activating/inhibitory interactions (top panel). Importantly, mRegDC_3 have least activating interactions, regardless of target cell type; and TNFSF and PVR-mediated interactions between mRegDC_2 and effector CD8 T cells are high throughout (bottom panel).

Whether mRegDCs *primarily* interact with a select T cell subset would also likely be dependent on cell location, number of TILs, tumour immune status (eg. T cell states) and immunogenicity, and many context-dependent factors. Rather than delving into the complexity of specific tumour CD8 T cell states, our manuscript focuses on molecular axes through which mRegDCs may support T cell activation and expansion more generically, which we believe our data addresses. Of note, PD-1+ populations express Cxcr6, which may facilitate co-localisation with Cxcl16-expressing mRegDCs.

We thank the reviewer for the productive suggestions and hope the new analyses provides some clarification.

14. In Fig. 5F, inhibitory ligand-receptor pairs such as PD-1-PD-L2, CTLA4-CD80/86 should also be analyzed, as they represent the main types of mRegDCs-CD8+ T cell interactions in human CRC (Fig. 4C). The extensive upregulation of stimulatory and inhibitory signals (e.g. PVR-TIGIT) makes it difficult to define the modulation of T-cell function by mRegDCs and to identify critical ligand-receptor pairs for their interaction. Ex-vivo or in vitro co-culture experiments may provide more evidence and should be considered.

We have now included analysis of the inhibitory ligand-receptor pairs shown in human CRC (**Fig. 4C**) for the mRegDC states found in our murine model, per the reviewer's request. This data has been included in the revised manuscript (**Ext. Data Fig. 9D**, below) and shows that similar inhibitory interactions exist between mRegDC states.

We appreciate the difficulty in making conclusions regarding the functional importance of mRegDC states given extensive upregulation of stimulatory/inhibitory signals, and thank the reviewer for the helpful suggestions regarding *ex vivo* or *in vitro* co-culture experiments. *Ex vivo* experiments (i.e. sorting tumour mRegDCs for culture) would be challenging because only ~1000 tumour-residing mRegDC could be isolated by FACS per tumour after tissue processing in our model (tumours are small to facilitate 100% efficiency of photo-conversion). Hence, the large number of mice needed would prove unfeasible, for which we apologise. As an alternative, we developed an *in vitro* co-culture system by generating mRegDC from bone marrow-derived DCs, since treatment of BMDC with apoptotic tumour cells induces the PD-L1+ mRegDC state (Maier et al., 2020). A graphical summary of the experiment set-up is included below.

Briefly, BMDCs from control or genetically modified (OX40L-deficient) mice were generated and cultured with UV-stressed (apoptotic) Ova-expressing MC38 cells for 8h to induce PD-L1⁺PD-L2⁺CCR7⁺ “mReg-BMDC” (above right). These were subsequently FACS isolated and co-cultured with naïve OT-I CD8⁺ T cells. Key findings were (selected figures below):

- Anti-PD-L1 antibodies increased proliferation and GzmB expression in antigen-specific OT-I cells.
- Loss of OX40L from DCs reduced expansion of OT-I cells.
- Treatment of mRegDC with recombinant IFN γ but not anti-PD-L1 antibodies *in vitro* or *ex vivo* induced an activated / stimulatory state resembling mRegDC₂.
- Pre-treatment of tumour-antigen experienced DCs with IFN γ promoted OT-I T cell expansion, reduced in OX40L-knockout.

We feel that these data provide substantial support for the functional importance of the mRegDC-CD8⁺ T cell ligand-receptor interactions we identified, the involvement of mRegDCs in anti-PD-L1 treatment, and provide some clues to the mechanism through which anti-PD-L1 antibodies promote immunogenic mRegDC-CD8⁺ T cell interactions, and we have therefore included them in the main manuscript. A full description is included under **Results** and **Fig. 5 / Ext. Data Fig. 10**.

15. In Fig.5G, the authors should explain why the MC38 model, which is inconsistent with the MC38-OVA model used in single-cell sequencing, was chosen for IF microscopy and whether the altered tumor immunogenicity was due to OVA affects mRegDC-CD8+T cell interactions. In addition, the proximity of mRegDCs and CD8+ T cells should be quantified, particularly whether mRegDCs affect the effector, proliferation, and exhaustion signatures of proximal CD8+ T cells.

We thank the reviewer for this comment and apologise for the lack of clarity here. The specific tumour model used was selected according to the two aims of our study, rather than the modality (eg. microscopy / scRNA-seq) we were using.

To investigate the phenotype and spatiotemporal dynamics of mRegDCs in solid tumours, we used multiple tumour models, including MC38-Ova, MC38, CT26, and B16 (scRNA-seq only), because we sought to generalise our findings across multiple models. Of note, we have added new data quantifying the Kaede profile of DCs from MC38 and CT26 tumours across a 5h to 72h time course, which is now included in **Ext. Data Fig. 3C**. Hence the spatiotemporal dynamics of tumour DCs have been assessed in all models used.

To investigate the effect of anti-PD-L1 treatment, we used MC38-Ova tumours. As the reviewer points out, MC38-Ova is more immunogenic than MC38, is associated with enhanced T cell responses to checkpoint inhibition, and hence is often a preferred model in immune-oncology research involving ICB intervention. Moreover, this model enables the tracking of Ova antigen-specific responses following ICB, as we have recently demonstrated (Li et al., 2022). Hence, in experiments where investigating the effects of ICB was one of the objectives, we used MC38-Ova tumours, including the scRNA-seq experiment.

We had not used microscopy to study the effects of ICB in the present study. Instead, microscopy was only used to investigate the presence of mRegDCs in tumours, and which cell types they interact with at baseline. Hence, MC38 and CT26 was used for microscopy, where there was co-localisation between mRegDC and CD8⁺ T cells. We have not compared mRegDC-T cell interactions between MC38 and MC38-Ova tumours, as this was not within scope; instead, we have shown that in Ova-expressing tumours, ICB alters the ligands expressed by mRegDCs, which may influence T cell activation.

mRegDC and CD8⁺ T cell co-localisation by IF microscopy has now been repeated, quantified, and added to the revised manuscript (**Ext. Data Fig. 9F**, below). These data indicate that in regions of the tumour where there are more mRegDC, there are also more CD8⁺ T cells. Finally, to investigate how mRegDC affects the effector, proliferation and exhaustion state of CD8⁺ T cells, we do not believe that microscopy of tumours is best suited to address this as multiple factors may influence CD8⁺ T cell state *in vivo* which is challenging to deconvolve, and limited plex. Instead, we have attempted to utilise *in vitro* co-culture systems to address this, as described above. Effector state of T cells engaging mRegDC in human tumours is addressed by analysis of PICseq data in **Fig. 6H** / **Ext. Data Fig. 11I-J**.

Minor suggestions:

1. In Fig. S1A, the figure legend and the graph do not match.

We apologise. This has now been edited.

2. On page 7, line 219, there is no evidence of stronger predicted interactions between mRegDCs and CD8⁺ T cells in human breast tumours and melanoma.

The data is included in **Ext. Data Fig. 1**. Here we are only referring to PD-1, CD28 and CTLA-4 ligand-receptor pairs (**Fig 4C**). Since all comparisons are made respective to *Pdcd1*⁺*Cd28*⁺*Ctla4*⁺ CD8 T cells, the relative expression of the cognate ligands on mRegDCs versus other myeloid cell types in **Ext. Data Fig. 1D-G** enables us to infer stronger mRegDC-mediated interactions via PD-1, CD28 and CTLA-4 ligation in these tumour types.

3. On page 8, line 268, "Since mRegDCs are the major source of PD-L1 in tumours". Although mRegDCs express higher levels of PD-L1 compared to macrophages and other cells, the abundance of mRegDCs appears to be lower. "the major source" is open to debate.

Considering cell frequencies, the reviewer raises a valid point. We have now edited this to "a major source".

Response letter to review #2 for:
**Time-, tissue- and treatment-associated heterogeneity in
tumour-residing migratory DCs**

Colin YC Lee *et al.*

Accompanying documents:
Manuscript text; revised v2
Figures; revised v2

Reviewer #2:

The manuscript by Lee et al analyzes the myeloid and lymphoid compartment in a mouse grafted tumor using single-cell RNAseq in combination with photoconversion at different time points. Based on computational correlations, the authors propose different patterns of DC sub-clusters (largely described in PMID: 32269339) to remain in tumors, migrate to lymph nodes and adopt more cancer-favoring or exhausted states. They confirm this (partially) in human cancers and show a correlation of mregDC states with T cell presence. The authors also show that anti-PD-L1 blocking antibody can cause a differential abundance of DC sub-clusters and alter their activation/maturation state. Finally, based on computational and spatial correlations, Lee et al propose the enhanced interaction of mregDCs with CD8+ T cells compared with other DC subsets.

While the use of state-of-the-art techniques is very much appreciated, almost the entirety of the presented manuscript is descriptive, largely based on correlations and analyses of few tumor samples, providing no mechanistic studies and therapeutically useful insights. Moreover, several interpretations of the authors may be influenced by factors that were not controlled for.

We thank the reviewer for their careful consideration of our manuscript and for providing these helpful comments, which have helped to improve the study. Please see below our point-by-point response to each comment, including revisions to the manuscript. Major revisions to the manuscript text have been highlighted in yellow.

Main comments:

1. The authors sort for analysis tumor-infiltrating myeloid cells (Live CD45+TER119-Kaede+CD11b+ NK1.1-). Classical intratumoral cDC1 lack CD11b expression by these cells (PMID: 32269339 and PMID: 29429633), leading to a biased analysis by the authors of the myeloid component. This questions some of the conclusions as CD11b neg cDC1s are not included in the analysis.

The reviewer raises an important point on CD11b expression on cDC1 vs cDC2. However, this comment arises from a lack of clarity in our original text/methods, for which we apologise. In fact, the scRNA-seq experiment was performed on **all CD45+** cells. To obtain information on Kaede fluorescence, we had to perform FACS sorting on tumour immune cells prior to scRNA-seq. Since FACS was necessary, we also sorted CD11b+ and CD11b- fractions (**Ext. Data Fig. 2B**), so that we could obtain adequate representation of all immune cells (since CD11b+ cells are much more frequent in tumour). However, during analysis, all cells were 're-combined' to form the scRNA-seq data object. Myeloid cells, and subsequently DCs, were identified from the scRNA-seq solely based on transcriptional profiles, meaning that both CD11b+ and CD11b- cells are represented in the DC data shown in Figure 1. In addition, we have included new data on surface expression of CD11b on tumour mRegDCs in response to suggestion 4 (see below); we hope this addresses the reviewer's concerns regarding CD11b expression on mRegDC.

We have now edited **Ext. Data Fig. 2B**, the **Results** text and **Figure legends** to state simply that scRNA-seq was performed on Kaede⁺CD45⁺Ter119⁻ cells. In addition, we have included a detailed explanation under **Methods**, to highlight that all CD45⁺ cells, regardless of surface CD11b or NK1.1 expression, are included in the analysis.

2. How does cell viability influence the interpretation of DC sub-cluster dwelling in tumors/migration to dLNs? The authors measure the percentage of Kaede green/red cells in tumors and the dLNs after different time points and then conclude the largest DC populations that dwell in the tumor or migrate to the LN. Puzzlingly, following the conclusion of the authors, mregDCs appear to be the dominant DC subset to remain in the tumor and also to migrate to the dLN – which sort of seems contradictory. Hence, what is the viability or life-span of DC subsets in tumors or the dLN after migration? Do cDC1s and cDC2s simply die earlier than mregDCs and, therefore, appear less reflected in the analysis of the authors – rather than mregDCs egressing more from the tumor to the dLN? Or, in other words, how do the authors explain that cDC1s and cDC2s remain less time in the tumor and migrate less to dLNs than mregDCs? What happens to cDC1s and cDC2s?

We thank the reviewer for these comments, and hope that the following new data, text amendments, and explanations help to clarify these concerns.

We do not believe that cDC1/2 are dying or have decreased viability compared to mRegDC. Instead, what we and others propose is that cDC become mRegDC (a mature cDC state) within the tumour (Maier et al., 2020). Maier *et al.* propose this is induced by uptake of tumour antigen by AXL. We have now included more data to support the maturation / development of cDC to mRegDC, in response to the reviewer's comment 4 (see below). Importantly, cDC1/2 maintain expression of surface cDC1/2 markers (XCR1/CD11b) after maturation to the mRegDC state (comment 4, below). Hence, they are not dying, but maturing to cDC1-mRegDC or cDC2-mRegDC. To further support this, we do not see any evidence that cDC1/2 have reduced viability, by examining cell death / stress genes in the scRNA-seq data (violin plots below). Indeed, cell death may occur among tumour DCs, but there is no evidence to support one subset dying at increased frequency. Finally, there are Kaede-red cDC1 and cDC2 present up to 72h after photoconversion, so those that have not acquired the mRegDC maturation programme retain their identity.

We do not believe that mRegDC being the major Kaede-red DC fraction in our tumour model, and the predominant DC subset to migrate is contradictory. As discussed, newly-infiltrating cDC may mature to mRegDCs with the appropriate cues. Part of the maturation programme is CCR7 expression, which is known to direct migration to the dLN. We have not challenged this dogma. Among Kaede-red cells in the dLN (that have come from the tumour), essentially all the DCs are mRegDC, since DCs lacking CCR7 expression are unlikely to migrate. We have now added new supporting data to the manuscript, quantifying the frequency of migrated DC subsets to the dLN by flow cytometry, to show this (below, **Fig. 2C**).

However, within the tumour, there are a significant number of CCR7+ mRegDC that are Kaede-red, even up to 72h after photoconversion, and the proportion of Kaede-red mRegDC is higher than Kaede-red cDC1/2 (below, **Fig. 1G**). This means that not all mRegDC egress to the dLN, at least not immediately. We have shown that, with time, tumour-retained mRegDC adopt a phenotype distinct from their migrated counterparts. Hence, it is not that mRegDC are pre-dominantly retained, but rather, not all successfully emigrate, and among tumour-retained DCs, mRegDCs are frequent.

Of note, we have now also included new time-course data from MC38 and CT26 tumours (**Ext. Data Fig. 2I**, below), which show that mRegDC are the major tumour-retained DC subset up to 72h post photo-conversion in these tumour models as well.

3. Are the effects of anti-PD-L1 therapy to the presence/state of DCs in tumors/tDLNs due to blockade of PD-L1 signaling or due to the altered tumor microenvironment/sustained T cell activation? Blocking PD-L1 will likely have cell intrinsic effects on DCs (that express these receptor), but the treatment does also notably alter the tumor microenvironment and behavior of other (immune) cells. Could the authors explore which is the main cause affecting DC sub-clusters? For example, by using PD-L1-deficiency strategies (of tumor cells, myeloid cells (bone marrow chimeras) or by blocking anti-PD1?

We thank the reviewer for this comment, indeed, the precise mechanism through which anti-PD-L1 antibodies influence tumour DCs (and whether mechanisms differ in anti-PD-1 therapy) is of interest to us, but was not within the scope of the present study, although we have included new data of relevance to this (see below). Of note, in our **discussion**, we stated that the precise mechanism of anti-PD-L1 antibodies remains unclear, but highlight several reports suggesting that reverse signalling via PD-L1 may occur. Moreover, it is clear from previous studies that provision of PD-L1

by CD11c+ cells is essential for effective immunotherapy (Oh *et al.*, 2020; Peng *et al.*, 2020), and our re-analysis of bulk RNA-seq of PD-L1-deficient tumour models (Lau *et al.* 2017, PMID 28220772) show that the immunogenic changes associated with loss of PD-L1 signalling is most strongly associated with knockout of PD-L1 from host immune cells rather than knockout of PD-L1 from tumour cells. This data is shown below but has not been included in the revised manuscript.

Furthermore, using a new co-culture system, we have further shown that expression of PD-L1 by mRegDC is functionally important in anti-PD-L1 therapy, and that the immunogenic changes we observe in DCs following anti-PD-L1 therapy are driven, at least in part, by IFN γ *in vitro* or *ex vivo* (below), which T cells increase secretion of following PD-1/PD-L1 blockade (Garris *et al.*, 2018; Grasso *et al.*, 2020). These new data is elaborated under our response to comment 7 and are included in **Fig. 5 / Ext. Data Fig. 10**. Of note, while these data do not definitively rule out a direct effect of the antibody on mRegDCs, and does not exhaustively address the mechanism of anti-PD-L1 treatment, it suggests that the changes we observed among DCs may be due to altered TME cytokines in immune-checkpoint therapy. We hope these address some of the reviewer’s questions, and have also added a sentence to the **Discussion** to explicitly address the limitations discussed here.

4. The authors claim that cDCs mature towards mRegDCs within the tumor based on the higher ratio of Red mReg DCs compared to cDC1s or cDC2s and based on pseudo-time trajectory analysis. Furthermore, they claim that there are different stages of differentiation of mReg DCs. To claim this, the authors would require a computational approach that actually takes into consideration cell transition (e.g. RNA velocity analysis) and/or confirm their data with experimental approaches (e.g. explore earlier timepoints upon photoconversion and show i) are mReg DC infiltrating only after cDCs? ii) Are green cDCs upon photoconversion changing abundance of markers (e.g. Cc1ta) over time?). In any case and given the authors data, how do the authors explain that most mregDCs express *Irf8* given that their model shows a transition from cDC2?

We thank the reviewer for the helpful suggestions, and recognise that there should be more work done to clarify the cDC to mRegDC transition. We have performed several of the analyses and experiments suggested, as below.

RNA velocity: RNA velocity analyses based on splicing dynamics support our existing proposed pseudotime trajectory, where mRegDC_1 are an intermediate state on the transition to mRegDC_2 or mRegDC_3, which are terminal tumour mRegDC states. This analysis is now included in **Fig. 1J** and **Ext. Data Fig. 3B**.

Early timepoints: To prove that mRegDC indeed arise from cDC, we decided to examine timepoints very early after photoconversion (i.e. 5 hours), per the reviewer’s suggestion. The hypothesis was that, if indeed mRegDC originate from cDC, most of the Kaede-green DCs 5 hours after photoconversion would be cDC, because newly-infiltrated DCs in this timespan would have had little time to acquire the mRegDC programme, but would be higher at later timepoints. Indeed, we found this to be true. At 5h post-photoconversion, the Kaede-green mRegDC:cDC ratio was approximately 1/100, but increased to 1/5 72h post-photoconversion. Hence, mRegDC emerge only after cDC influx. The data is shown below and is now included in **Ext. Data Fig. 3C**.

cDC1/2 ontogeny: Finally, regarding specific cDC1/2 origin of tumour mRegDCs: Difficulties in transcriptionally distinguishing mRegDC based on origin is recognised (Gerhard et al., 2021; Ginhoux et al., 2022; Maier et al., 2020), and hence the transition in scRNA-seq analysis may be difficult to interpret. Specifically, upregulation of the common mRegDC programme upon DC maturation, and downregulation of cDC1- or cDC2- associated transcripts make it difficult to tell cDC1- or cDC2- derived mRegDC apart transcriptionally, but they may maintain some surface expression of cDC1/2 markers. Hence, we chose to address the question of mRegDC origin by investigating surface XCR1 or CD11b expression (data below). These data indicate that XCR1+ and CD11b+ mRegDC both exist, where XCR1+ mRegDC are enriched in the Kaede-red fraction. Hence, we concluded that mRegDC arise from both cDC1 and cDC2, and similar to tumour

cDC2:cDC1 ratios, CD11b+ mRegDC are more frequent than XCR1+ mRegDC. This conclusion could not be made based on the transcript data alone, and is now included in **Ext. Data Fig. 3D**.

mRegDC express high levels of *Irf8* but lower levels of *Irf4* (below), and as the reviewer points out, we highlighted *Irf8* expression in mRegDCs as an indication of cDC1 origin in the original manuscript submitted. However, in light of this new surface phenotyping, it is more likely a large proportion of tumour mRegDC are cDC2 derived, and that *Irf8* may be expressed as part of the mRegDC programme. We have therefore clarified this in the revised text.

In summary, our data supports that mRegDC arise from both cDC1 and cDC2, and most mRegDC in the tumour are cDC2 derived, as indicated by surface cDC1/2 marker expression. The revised manuscript now has a new section in **Results** addressing cDC to mRegDC maturation, and mRegDC ontogeny.

5. What (which mechanism, molecules, signaling) is causing mregDC dwelling in tumors or driving their migration to LNs? Could that be promoted/therapeutically targeted? Could the authors explore the transcriptome of those differential states of mregDCs and functionally test candidate genes to obtain therapeutically useful findings? Or, in other words, what are crucial differences (in signaling and other features) between mregDC1 and 2/3? Can this transition be avoided or mregDC migration to tdLNs induced by targeting certain pathways (therapeutically)?

We thank the reviewer for their comment on mRegDC migration – clearly, as our data indicate, CCR7 upregulation is not sufficient in itself. We have not fully addressed the determinants of mRegDC migration as we it was not within the scope of the present study, but have highlighted this as a limitation/scope for future work in the manuscript discussion.

However, we have included some data below that consider several possibilities.

First, we sought to identify differentially expressed chemokine receptors between mRegDC_1 and tumour-retained mRegDC_2/3 which may “hold” CCR7+ DCs in the tumour. CCR7 itself was not significantly different between subsets, including at the protein level (panels A, C, D). *The CCR7 flow data has now been added to the manuscript (Ext. Data Fig. 4I)*. *Cxcr5* and *Ccr6* were upregulated on tumour mRegDC_2/3, but was only expressed on about 20% of mRegDC or less (A, C). GRK3/6, which desensitise CCR7 signalling following persistent ligand stimulation, was not differentially expressed between tumour mRegDC subsets (B-C). Of note, *Grk3* was highest on migrated mRegDC, which may suggest that retained mRegDCs have received less CCL19/21 ligation.

Since tumour mRegDCs express *Ccr7*, *Cxcr5* or *Ccr6*, we analysed bulk RNA-seq of whole tumours (from our murine model) to identify expression of their cognate ligands within the tumour itself that may act on these receptors (**E**). *Ccl20* (*Ccr6* ligand) was not detected, but there was low expression of *Ccl19/Ccl21a* (*Ccr7* ligands) and *Cxcl13* (*Cxcr5* ligand). We turned to our full scRNA-seq data to identify which cell types may express these ligands (**F**). In this data, *Ccl21a* was not detected, but *Ccl19/Cxcl13* expression was detected (<2%) from stromal cells (**G**). These were likely fibroblasts, given their expression of *Fap*, *Col6a1*, *Dcn* (**H**).

We next used IF microscopy to investigate these ligand/receptor candidates. As shown in the manuscript (**Fig. 1H**), tumour-retained mRegDC express CCR7. We were only able to detect CCL19 staining in tumour dLNs, but not the tumour (data not shown). Some CCL21 staining was detected in the tumour, but these were primarily from LYVE1+ cells (**I-J**), with some adjacent CCR7+ mRegDCs. However, these may represent imminent DC emigrants via the lymphatics, and many mRegDCs were not adjacent to lymphatic structures. Alternatively, it is possible that some CCL21+ LYVE1+ cells do not form patent/functional lymphatics (analogous to aberrant angiogenesis) and retain mRegDCs. Overall, these data suggest it is unlikely that aberrant CCL19/CCL21 expression (eg. by tumour cells) retain CCR7+ DCs. We could not detect CXCL13 or CXCR5 by conventional antibody staining of tumour sections, and these may require reporter mice to further investigate.

R2; comment 5

We have not included the above data in the current manuscript. However, we are happy to reconsider if the reviewer or editor believes this is necessary.

6. *mRegDC and CD8 T cell colocalization via spatial transcriptomics and fluorescence microscopy has to be quantified.*

Co-localisation in spatial transcriptomics is already quantified (Fig. 4D-F, 6I-J, Ext. Data Fig. 12B-C). Values (colour) represent the Pearson correlation coefficient (R-value) of expression values or enrichment scores between voxels in the local ($k=6$) neighbourhood, where only significant correlations are shown. More details are provided under Methods. To make this clearer, we have now added a legend to the colour bars ('Pearson R-value') to relevant figures. We apologise that this was not clearer in the original submission.

mRegDC and CD8T cell co-localisation by fluorescence microscopy has also now been repeated and quantified by spatial Pearson correlation, and added to the revised manuscript (**Ext. Data Fig. 9F** and accompanying text). These data indicate that in regions of the tumour where there are more mRegDC, there are also more CD8 T cells.

7. The role of DCs on anti-PDL1 tumor therapy is already established (PMID: 32973173). Can the authors demonstrate an experimental and selective increase of mReg DC & CD8 T cell interaction upon anti-PDL1 therapy?

We acknowledge that the role of DCs in anti-PD-L1 therapy has been established, and have referenced this in our introduction, including PMID: 32973173. However, the precise DC cell-types involved or the site at which PD-L1+ DCs reside/function have not been explored. Our study provides novelty in showing that, surprisingly, CCR7+ PD-L1+ DCs may be retained in the tumour and modulate effector T cells locally, influenced by anti-PD-L1 therapy.

We have not made the claim that there are increased numbers of mRegDC – CD8 T cell interactions following treatment. In addition, since PD-L1-treated tumours are significantly smaller in size (**Ext. Data Fig. 2A**) and are well known to have increased T cell infiltration, it is difficult to objectively measure whether there is an increase in the frequency of mRegDC-T cell interactions specifically, or if this is confounded by the above factors. However, what we have highlighted in our manuscript is that the precise nature of the ligand-receptor interactions between these cell types are changing (**Fig. 5D**). Specifically, with anti-PD-L1 treatment, maturation of mRegDCs are skewed towards a state with a distinct stimulatory ligand profile.

In our revised manuscript, we have attempted to further assess functional significance of these mRegDC-CD8⁺ T cell interactions, in the context of anti-PD-L1 treatment. While we had initially hoped to perform *ex vivo* experiments (i.e. sorting tumour mRegDCs to culture), this was not feasible due to the low frequency of tumour-residing mRegDC that could be isolated by FACS per tumour after tissue processing in our model. As an alternative, we developed an *in vitro* co-culture system, generating mRegDC from bone marrow-derived DCs (BMDC), since treatment of BMDC with apoptotic tumour cells induces the PD-L1+ mRegDC state (Maier et al., 2020). A graphical summary of the experiment set-up is included below.

Briefly, BMDCs from control or genetically modified (OX40L-deficient) mice were generated and cultured with UV-stressed (apoptotic) Ova-expressing MC38 cells for 8h to induce PD-L1⁺PD-

L2⁺CCR7⁺ “mReg-BMDC” (above right). These were subsequently FACS isolated and co-cultured with naïve OT-I CD8⁺ T cells. Key findings were (selected figures below):

- a) Anti-PD-L1 antibodies increased proliferation and GzmB expression in antigen-specific OT-I cells.
- b) Loss of OX40L from DCs reduced expansion of OT-I cells.
 - i. Of note, we also have new data using OX40L-reporter mice showing that OX40L expression on mRegDC is restricted to tumour mRegDC but not in the dLN *in vivo*. This data is now included in **Fig. 3H / Ext. Data Fig. 6F-G**. These data underline the importance of a tumour-restricted mRegDC state capable of modulating T cell function.
- c) Treatment of mRegDC with recombinant IFN γ but not anti-PD-L1 antibodies *in vitro* or *ex vivo* induced an activated / stimulatory state resembling mRegDC₂.
- d) Pre-treatment of tumour-antigen experienced DCs with IFN γ promoted OT-I T cell expansion, reduced in OX40L-knockout.

We feel that these data provide substantial support for the functional importance of mRegDC-CD8⁺ T cell ligand-receptor interactions we identified, the involvement of mRegDCs in anti-PD-L1 treatment, and provide some clues to the mechanism through which anti-PD-L1 antibodies promote immunogenic mRegDC-CD8⁺ T cell interactions, and we have therefore included them in the main manuscript. A full description is included under **Results** and **Fig. 5 / Ext. Data Fig. 10**.

Additional comments:

- Please, include the source and genes that constitute the “mReg DC Signature” (ExtD Fig. 1)

We used the genes specified in the original manuscript defining mRegDCs (Maier et al., 2020). The list of genes are included in Supplementary table 1.

- Please, show the gatings for cDC1, cDC2 and mRegDC (Fig. 1F)

The gating strategy is shown in **Ext. Data Fig. 2F**.

- What are the mechanisms behind triggering exhaustion in tumor-residing mregDCs? Are other DC subset also suffering a similar exhaustion?

We have not yet deciphered the mechanisms that drive changes to tumour-residing mRegDCs in the local TME, but this will certainly be the aim of future work. We thank the reviewer for the thought-provoking question on the fate of cDCs that do not become mRegDC. Certainly, Kaede-red cDC appear to upregulate genes associated with mRegDC maturation, including *Axl* and *Ciita*, suggesting that these are part of a transitional state towards mRegDC. Consistent with this, PCA analysis of tumour cDC show that from Kaede-green to Kaede-red cDC, there is an upregulation of genes involved in antigen presentation (**Ext. Data Fig. 3G**, below). Downregulation of ‘response to interferon-gamma’ or ‘innate immune response’, etc., may be part of this maturation programme, or may reflect loss of inflammatory gene expression in tumour cDC with time. Unfortunately, our model does not enable us to distinguish between cDC that mature to mRegDC or cDC that remain in the tumour but do not mature, and whether the latter suffer from “exhaustion”. Certainly, CCR7^{neg} cDC are heterogeneous, with variable ability to influence tumour T cells (PMID: 37451271), but we have not sought to address this in the current study.

Of note, by 72h post-photoconversion, only ~10% of cDC are Kaede-red (**Fig. 1G**, below), following a downward trend. This means that cDC are not a major long-dwelling tumour DC subset, and provide a rationale for focusing on mRegDCs in this context.

- Please provide the information which tumor model was used for analysis in every Figure legend, because is not really clear in every case.

We apologise for the lack of clarity and have now included the specific model used in all figure legends as requested.

- Please add a quantification and statistics of Figure 1F, 5G, 6I-J

Fig. 1F is quantified in **1G** and **Ext. Data Fig. 2G**. **Fig. 5G** (now **Fig. 5E**) is a representative inset, but quantification of mRegDC and CD8T co-localisation is now included in **Ext. Data Fig. 9E-F**. **Fig. 6I-J** is already quantified, as explained in response to the reviewer's main comment 6.

- In page 5, where it says Extended Data Fig. 1H-J, it should say Extended data Fig 2H-J.

We apologise. We have noted and corrected this error.

- ExtD Figure 3D – please display differently – the average is not visible, is CCR7 and Fcscn1 expression up or downregulated?

We apologise for the lack of clarity. We have now added arrows to indicate whether expression is upregulated or downregulated in the violin plots. Please note that Ext. Data Fig. 3D is now **Ext. Data Fig. 3E** in the present manuscript.

- In Figure 4, the authors show correlations of CD8 T effector T cells with mregDCs. To put those into context, could the authors also analyze the correlation T cells with the other DC sub-clusters (cDC1s and cDC2s)? That would be useful to place the mregDC state into the context of current knowledge, especially on the potency of cDC1s in cancer (PMID: 30352680)

We thank the reviewer for this suggestion, and have performed the analysis requested. As seen below, there is generally a correlation between T cells and all DC subsets. This is in line with the notion of immune 'hot' or immune 'cold' tumours or that tumour biopsies from which these data are derived are capturing either immune cell-infiltrated regions or immune cell-sparse regions. Regardless, the Pearson correlation coefficient is highest for mRegDCs across these cancer types.

We have not currently included this data in the manuscript (it is difficult to interpret or compare cDC1/2 to mRegDC since both cDC1 and cDC2 mature to mRegDC), but are happy to do so should the reviewer or editor wish to.

- Authors from ref. 37 highlight the interaction of mReg with TILs expressing CD4+PD1+CXCL13+. How do the authors convene this data with the major interaction between mReg and CD8 effector T cells shown by the authors in the present manuscript?

We thank the reviewer for raising this important comment. Cohen *et al.* (Nature Cancer, 2022) use PICseq to investigate myeloid-T cell interactions, a technique that requires strong physical bonds between 2 cells to survive their tissue dissociation protocol, for doublet sequencing. In their data, they highlight interactions between mRegDC and CXCL13⁺ CD4 T cells, but of note, there are also many mRegDC-effector CD8 T cell doublets, which we have used for our analysis. However, we do not know how strongly mRegDC adhere to CD8 T cells versus CD4 T cells relatively, and hence the extent to which respective doublets survive tissue dissociation. We do not think strength of adherence equates to importance of their immunomodulatory interactions – i.e. mRegDC and CD8 T cells may co-localise and have important interactions, but not captured as frequently in PICseq compared to CD4 T cell doublets after dissociation.

Importantly, we are not refuting this data by any means – in fact, we believe these data support our proposal that mRegDCs influence immune cells locally within the tumour, not just the dLN. It is likely that mRegDCs interact with many other immune cell types, including CXCL13⁺ CD4 and regulatory T cells, and may even be present in TLS. We have now highlighted this in our **Discussion** (text below), including the need to further investigate how heterogeneous mRegDC states interact with various cell types, beyond CD8 T cells. The main message of our manuscript was that mRegDCs consist heterogeneous cell states, and we decided to focus on CD8 T cells because we found that they correlate in frequency and co-localise in space (**Fig. 4-5**). However, we do not claim or believe that these interactions are exclusive. We hope that these revisions to the manuscript text help to reconcile these data.

“Recent studies also report that mRegDCs may engage other immune cells in tumours, including CXCL13⁺ CD4⁺ T cells, regulatory T cells and NK cells, and may reside in TLS. How heterogeneous mRegDC states influence the survival or activation of other immune cells, including within TLS, warrants further investigation.”

Response letter to review #3 for:
**Time-, tissue- and treatment-associated heterogeneity in
tumour-residing migratory DCs**

Colin YC Lee *et al.*

Accompanying documents:
Manuscript text; revised v2
Figures; revised v2

Reviewer #3:

This study presents scRNA-seq analysis of DCs in a tumor setting (predominantly MC38-OVA) with or without anti-PDL1 blockade. The authors add the Kaede photo-convertible marker in the mice that allows confirmation that migratory DCs in tumor-draining LNs came into dLNs from flashed tumor or remained in the tumor after the photoflash. Most of the findings are in line with known DC tumor biology, but the study, although extensive in computational approaches, remains descriptive and speculative and lacks any functional tests of the author's interpretations. For example, the authors argue for an 'exhausted' phenotype of DCs based on gene changes that differ from what was used to define exhaustion in T cells. T cell exhaustion was first a functional definition and its definition by molecular markers has evolved beyond the simple notion of expressing some inhibitory markers, such as PD-1, which are now also seen in normally activated T cells. Further, the authors have not documented DC exhaustion functionally or tested it genetically. While the study represents extensive computation description, it falls short in my opinion for publication in Nature Immunology.

We thank the reviewer for their time taken to consider our manuscript and for the constructive feedback provided. We acknowledge the importance of functional testing, and have performed additional experiments using newly-generated OX40L reporter mice and OX40L-deficient DCs, and developed an *in vitro* co-culture system to address some aspects of this. We believe that these additional experiments and analyses further strengthen our data and hope that they address at least some of the reviewer's concerns. Please see below our point-by-point response to each comment, including relevant revisions to the manuscript. Major revisions to the manuscript text have been highlighted in yellow.

FACS enriched red or green myeloid cells and TILs were subjected to scRNAseq and were clustered into 8 groups, including cDC1, 3 groups of cDC2, and 3 groups of so-called "mregDCs" (Fig. 1C). mregDCs were so named by Maier and Merad and they correspond to previously recognized mature CCR7+ DCs, as analyzed by Ardouin and Malissen in Immunity 2016. Previous terminology referred to as immature CCR7-negative DCs, and mature CCR7+ DCs, which Arduin characterized in a resource paper, documenting gene expression changes between unactivated and activated DCs. Many of same genes used to mark "mregDCs" now, such CCR7 and CD274 were also identified by Ardouin and Malissen. The Maier paper did not cite Ardouin/Malissen, and so the mRegDC authors may have been unaware of the extensive characterization of induced and homeostatic DC maturation.

We apologise for our oversight in acknowledging the work done by Ardouin and Malissen 2016 and have now referenced this study in our manuscript. In addition, we have specifically highlighted in the manuscript **introduction** that the CCR7+ PD-L1+ PD-L2+ DC state has been variably named "migratory DCs", "mRegDCs", "cDC3", or "mature DCs" by different authors, but have chosen to adopted the nomenclature "mRegDC" for the remainder of the manuscript. We chose this

terminology because we felt it highlighted the potential regulatory consequences of PD-L1 expression and consequent involvement in immune checkpoint therapy by this mature DC state in the cancer context.

In any case, the current authors rely on the mregDC terminology for activated DCs. The data in Fig. 1 is repetitive of previous work but confirms various patterns of gene expression between CCR7- and CCR7+ DCs using scRNAseq. The authors carry out a pseudo-time analysis which they rely on for subsequent interpretations. The problem is that this approach makes untested assumptions of a unidirectional trajectory determined in the UMAP plot. However, this assumption is never tested and may not be true. The arrow diagram shown in Fig. 1 panel is a hypothesis, not a conclusion. A particular weakness is that this appears to mix cDC1 and cDC2 lineages together in the trajectory into “mregDCs”. Such a trajectory is an unlikely process, given that these are molecularly separate lineages. The authors state “This revealed a trajectory terminating in mRegDC_2/3 that transitioned through the mRegDC_1 state”. It’s not even clear what this sentence means, but it is clear that the authors have no experimental basis to make this claim. Pseudo-time is a computation exercise, but it is not the basis for conclusion. The rest of this paragraph is also speculative based on observed changes in gene expression. What is a “migratory transcript”? And of what consequence is the change in Ciita when the literature invokes other pathways, such as MARCH1, for the regulation of surface MHC expression in DCs? Also, it is not a surprise that a DC might mature more with greater duration in the TME, but even that conclusion was not tested functionally.

We thank the reviewer for their comment on the DC maturation trajectory within tumours, and recognise that there could be more work done to clarify the cDC to mRegDC transition. We have now performed more analyses and experiments to investigate this. Our new data suggest that both cDC1 or cDC2 independently mature to form CCR7+ mRegDC, but a proportion are retained in the tumour and do not migrate. Within tumour mRegDCs, the terminal states are mRegDC_2 and mRegDC_3, and mRegDC_1 is a transitional DC state within this trajectory. These claims are supported by the following revisions:

RNA velocity: RNA velocity analyses based on transcript splicing dynamics support our proposed trajectory, where mRegDC_1 are an intermediate state on the transition to mRegDC_2 or mRegDC_3, which are terminal tumour mRegDC states. This analysis is now included in **Fig 1J** and **Ext. Data Fig 3B**. This approach independently validates the trajectory derived from pseudotime analysis computed on gene expression data. Moreover, the Kaede-green:Kaede-red ratio of mRegDC clusters (**Ext. Data Fig. 2H**) provides further experimental support for progression through this trajectory.

Early timepoints: To show that mRegDC indeed arise from cDC, we examined timepoints very early after photoconversion (i.e. 5 hours). If indeed mRegDC originate from cDC, most of the Kaede-green DCs 5 hours after photo-conversion would be cDC, because newly-infiltrated DCs in this timespan would have had little time to acquire the mRegDC programme, but would be higher at later timepoints. We found this to be true. At 5h post-photoconversion, the Kaede-green mRegDC:cDC ratio was approximately 1/100, but increased to 1/5 72h post-photoconversion. This data proves that

mRegDC emerge only after cDC influx. The data is shown below and is now included in **Ext. Data Fig 3C**.

Surface phenotype indicating cDC1/2 origin: Regarding specific cDC1/2 origin of tumour mRegDCs – difficulties in transcriptionally distinguishing mRegDC based on their origin is recognised (Gerhard et al., 2021; Ginhoux et al., 2022; Maier et al., 2020). Specifically, upregulation of the common mRegDC programme upon DC maturation, and downregulation of cDC1- or cDC2-associated *transcripts* make it difficult to tell cDC1- or cDC2- derived mRegDC apart using in RNA-seq data. Hence, as the reviewer points out, the lineage trajectories in the scRNA-seq data appear to be mixed, with both cDC1 and cDC2 converging on mRegDC single-cell transcriptomes. In reality, the mRegDC clusters likely include independently derived cDC1- and cDC2-mRegDCs, and hence, is probably a source of confusion in the pseudotime plot. However, some surface expression of cDC1/2 markers are retained on mature DCs and may provide clues. We investigated surface XCR1 or CD11b expression on tumour mRegDCs (data below). These data indicated that XCR1+ and CD11b+ mRegDC both exist, where XCR1+ mRegDC was higher in the Kaede-red fraction. Hence, these data support that mRegDC arise from both cDC1 and cDC2, and similar to tumour cDC1:cDC2 ratios, CD11b+ mRegDC are more frequent than XCR1+ mRegDC. This conclusion could not be made based on the transcript data alone, and is now included in **Ext. Data Fig 3D**, including new accompanying text under **Results** and **Discussion** addressing the cDC to mRegDC maturation and mRegDC ontogeny.

Terminal tumour mRegDC are phenotypically distinct: mRegDC_1, which are early in the proposed maturation trajectory, express lower levels of *Cd80* and *Cd86* than mRegDC_2/3 (**Ext. Data Fig. 4B**). Consistent with this, tumour-retained Kaede-red mRegDC upregulate surface CD80/86 compared to newly-formed Kaede-green mRegDC. Hence, high CD80/86 expression is a feature of terminal tumour-retained mRegDCs in the proposed trajectory, and lower in the intermediate/transition state. Furthermore, mRegDC that successfully migrate to the LN, which we have proposed arise from “early” mRegDCs in the trajectory, also express lower levels of CD80/86, and hence are distinct from the tumour-retained state. This data is shown below and is now included in **Fig. 2H**.

“Migratory transcripts” in the text refers to molecules involved in DC migration, as defined in Fig 1c of Maier *et al.*, Nature 2020. We examined *Ciita* because of their well-established role in controlling class-II MHC expression and presentation. However, we have investigated other genes, including *March1*, as in published works, including Ardouin and Malissen 2016. As shown below, there are similar patterns of gene expression. Beyond individual genes, this point has also been further supported in **Ext. Data figure 3G**, where PCA and GO enrichment analysis show that pathways relating to antigen presentation of exogenous/endogenous antigens are upregulated with time in tumour cDC.

Figure 2 is a further description of gene changes in cells from different locations or clusters, but with no experimental test for the consequence of any particular change. Certain observed changes were discussed in terms of possible links to function, such as for CD80/86, ICOS or IL-12, but this is just observational speculation. The last of Figure 2 was another computational exercise in description, integrating tumor-resident DC data with LN DCs, making statements about cluster percentages or transcripts for “DC chemotaxis” and “CCR7 signaling” but without any functional data component. This ends with the speculation “Hence, we propose that mRegDC 2 and mRegDC_3 are terminal, tumour-residing mRegDC states, and DCs that have transited through the intermediate mRegDC_1 state, become increasingly unlikely to egress.” This is a nice proposal, but the authors have not tested it.

We thank the reviewer for these comments. Some of these have been addressed in the preceding response. RNA velocity analysis highlighted previously supports our hypotheses that mRegDC_2/3 are terminal tumour-residing mRegDC states, and that mRegDC_1 is intermediate. Data highlighted in the preceding section on the phenotypic similarity (CD80/86) of migrated mRegDC in the LN to newly-formed Kaede-green mRegDC *in vivo*, rather than Kaede-red DC with prolonged tumour dwell-time, further supports the claim that mRegDC_1 is an intermediate state, and that successful dLN emigrants resemble this intermediate/newly-formed mRegDC state.

We have also provided new experimental data, included in **Fig. 3H/ Ext. Data Fig. 6F-G** of the manuscript, that further supports a terminal tumour mRegDC phenotype absent in LNs. Using new OX40L-reporter mice, we showed that OX40L expression is specific to tumour-retained mRegDCs,

but not expressed by mRegDCs in the dLN *in vivo*. The fact that OX40L expression is not detected in the dLN suggests that this molecule is only upregulated by CCR7+ DC that have failed to migrate, and that OX40L+ tumour-retained mRegDC are highly unlikely to egress to the dLN, or at least is rapidly downregulated upon tumour exit, supporting our conclusions. We have gone on to interrogate the functional importance of OX40L expression using *in vitro* cultures, which is elaborated in response to the following comment. These data address the functional significance of the tumour-specific mRegDC phenotype.

Of note, the primary aim of our study was to highlight that CCR7+ mRegDCs in cancer are heterogeneous, which is related to time, tissue site, and immunotherapy; we have not investigated how mRegDC-mediated T cell activation differs in LNs versus tumours. We believe that the revised data further support the existence of these variable mRegDC states.

Fig 3 continues the description of gene changes between “states” defined in the pseudo-time analysis. Here the authors create a description of “exhaustion” for DCs, but “as defined in T cells, namely reduced expression of molecules enabling effector function(in this case antigen presentation)...” This conclusion is wholly unwarranted. The authors have not tested any functional aspect of any of these speculations. The term “mRegDCs” has not been universally accepted to replace activated CCR7+ DCs, so it seems unwise to make the claim for “exhausted mRegDCs” with not functional evidence.

We thank the reviewer for this comment. Our data shows that activated CCR7+ DCs/mRegDCs, regardless of nomenclature, are heterogeneous, and their ligand expression profile suggests they have unequal capacities to support lymphocyte responses. Ideally, we would sort newly-formed Kaede-green mRegDCs and Kaede-red mRegDCs that have been retained in the tumour for >72h (‘exhausted’) to compare their ability to propagate antigen-specific T cell responses. Unfortunately, despite our best efforts, these functional assays were not feasible with the small number of DCs that can be isolated from tumours (<1000 live Kaede-red mRegDCs per tumour). Instead, we developed an *in vitro* co-culture system (elaborated below) to enable us to investigate the functional importance of individual molecules expressed by these DCs. However, we have added a comment explicitly acknowledging that this system may not recapitulate the whole TME *in vivo*, but do provide supporting data for our use of the term ‘exhausted’ DCs.

In the context of E John Wherry’s definition of T cell exhaustion (PMID: 21739672): (1) poor effector function, (2) sustained expression of inhibitory molecules and (3) a transcriptional state distinct from functional effector cells; we believe that tumour-retained mRegDCs exhibit features of all the above hallmarks. While reduced effector function could not be directly assessed, and remains a limitation as the reviewer points out, mRegDCs with prolonged tumour dwell-time sharply downregulate expression of genes involved in antigen presentation, as well as inflammation, cytokine production, and response to inflammatory stimuli (eg. DAMP sensing), highlighted in **Fig.**

3D; which are all key functions of DCs. Moreover, the supposed effector function of CCR7⁺ DCs is to migrate to LNs and present antigen to surveying T cells, which tumour-retained mRegDCs do not. Next, they sustain high levels of PD-L1/PD-L2 expression, and other inhibitory molecules, shown in **Fig. 1** (hallmark 2). Finally, tumour-retained mRegDCs acquire a phenotype distinct from successful dLN emigrants, as highlighted in **Fig. 2** (hallmark 3). We therefore feel it is reasonable to draw parallels between this DC state and ‘exhausted’ T cells.

One of the potentially deleterious consequences of the prolonged tumour dwell-time of mRegDCs, which maintain high levels of PD-1 ligand expression, and their co-localisation with CD8⁺ T cells, is that they may inhibit anti-tumour cytotoxic T cell function. To investigate the functional importance of tumour-retained mRegDC-expressed ligands in regulating antigen-specific cytotoxic T cell activity, in the context of anti-PD-L1 treatment, we developed the aforementioned *in vitro* co-culture system using “mReg-BMDC” generated from bone marrow-derived DCs. A graphical summary of the experiment set-up is included below.

Briefly, BMDCs from control or genetically modified (OX40L-deficient) mice were generated and cultured with UV-stressed (apoptotic) Ova-expressing MC38 cells for 8h to induce PD-L1⁺PD-L2⁺CCR7⁺ “mReg-BMDC” (above right). These were subsequently FACS isolated and co-cultured with naïve OT-I CD8⁺ T cells. Key findings were (selected figures below):

- Anti-PD-L1 antibodies increased proliferation and GzmB expression in antigen-specific OT-I cells.
- Loss of OX40L from mRegDCs reduced expansion of OT-I cells.
- Treatment of mRegDC with recombinant IFN γ but not anti-PD-L1 antibodies *in vitro* or *ex vivo* induced an activated / stimulatory state resembling mRegDC₂.
- Pre-treatment of tumour-antigen experienced DCs with IFN γ promoted OT-I T cell expansion, reduced in OX40L-knockout.

We feel that these data provide substantial support for the functional importance of mRegDC-CD8⁺ T cell ligand-receptor interactions we identified, including the tumour mRegDC-specific molecule OX40L, the involvement of mRegDCs in anti-PD-L1 treatment, and provide some clues to the mechanism through which anti-PD-L1 antibodies promote immunogenic mRegDC-CD8⁺ T cell interactions. A full description is included under **Results** and **Fig. 5 / Ext. Data Fig. 10**.

Fig. 4 does a de-convolution of transcriptomes from human CRC or melanoma or breast and lung tumors. The authors use xCell to for the deconvolution algorithm to estimate abundance of various cell types in these mixed population of cells. The problem is that deconvolution makes various untested assumptions, for example that expression signatures are unambiguous and distinct for each cell type. However, the data in this paper already shows that there is flexibility in the gene expression pattern of DCs in various activation states, so how can they validate this deconvolution? There may be significant overlap in gene expression profiles between different closely related cell states. In addition, there was no treatment of the technical noise, batch effects, sample variability for this deconvolution. In short, this is another computational approach that lacks specific testing for its predictions. Finally, this approach cannot determine the functional states of the cells that it estimates.

We thank the reviewer for this comment. We would like to highlight that the deconvolution analysis in the early part of figure 4 was only used to make observations on which cell types mRegDC may associate with, but co-localisation with CD8⁺ T cells was independently confirmed using spatial techniques. We felt this ‘top-down’ approach to be necessary, so we could unbiasedly generate hypotheses for targeted testing. The obvious advantage of public repositories such as TCGA is the ability to access thousands of human tumour biopsy samples for highly-powered analyses. Deconvolution (i.e. digital cytometry) remains the gold standard methodology for estimating cell type abundances from bulk transcriptomics data, and is used by many studies.

xCell (PMID: 29141660) is among the most widely used deconvolution softwares and has been cited several thousand times (eg PMID 32803172, 29057876, 34172722, 30894752, 37679568, 37541199,

etc.). The authors have not assumed that expression signatures are unambiguous or distinct for each cell type and apply a novel spill-over compensation technique to reduce dependencies between closely related cell types. Importantly, they showed that the method outperforms other methods in dissecting the TME composition and the output was comprehensively validated with flow cytometry immunophenotyping. Moreover, the authors applied xCell to TCGA in the original manuscript to highlight its performance in deconvolving TME composition of TCGA transcriptomes, as we have done. Finally, as the authors point out, a major advantage of xCell is in its rank-based approach (rather than counts based) to scoring cell type signatures. Consequently, it is agnostic to batch effects, and accounts for both technical and biological noise. Hence, we feel that the application of xCell here is appropriate.

Importantly, in our study, we have not utilised the deconvolution analysis to draw conclusions regarding ‘closely related cell states’ that mRegDC may be interacting with, nor have we discussed the ‘functional’ state of cells it estimates’. We have only noted that mRegDC correlate with CD8 T cells broadly in this transcriptomic data. The specific text is as follows: “*We deconvolved 521 human CRC transcriptomes (The Cancer Genoma Atlas, 2013) and found the highest correlation between mRegDC and CD8⁺ T cell transcripts (Fig. 4A), an observation replicated in melanoma, breast, and lung tumour biopsies (Fig. 4B).*” Given that TCGA data is from tumour biopsies, this implies site-specific co-localisation of mRegDC and CD8⁺ T cells. We have hence interrogated this using Visium data for human tumours and subsequently IF microscopy in murine tumours. Of note, for this revision, we have further quantified the spatial association between mRegDC and CD8⁺ T cells in murine tumours. Furthermore, PICseq data from human lung tumours show that mRegDC and CD8⁺ T cells physically interact. Therefore, our subsequent analyses independently validate the observations we have made from cellular deconvolution of TCGA, allowing us to draw consistent conclusions.

In Fig. 5, the authors analyze doublets by PIC seq from NSCLC, to identify interacting cells. The authors sorted T cell- myeloid cell doublets for this, and concluded that “mRegDC-CD8⁺ T cell doublets were more frequent than other mRegDC-T cell combinations” and that “PICs containing mRegDC 230 (CCR7⁺FSCN1⁺CD274⁺) highly co-expressed CD8B, PRF1, GZMB and PDCD1 (Fig. 4F), confirming that in NSCLC, mRegDC and effector CD8⁺ T cells physically interact.” This and the remainder of the study, which presents transcriptional analysis of DCs from human cancers, also remains at a descriptive and speculative level. The conclusions about conservation of heterogeneity of activated DCs is not novel or surprising.

We re-analysed human RNA-seq data sets to look for evidence that mRegDCs interact with T cells in human tumours, in a similar manner to what we observed in our murine models. In particular, the PICseq data showed that the same molecules are involved in human tumours, between interacting mRegDC and CD8 T⁺ cells. This data, combined with our new murine phenotyping data and *in vitro* cultures, gives us confidence that cellular/molecular interactions between mRegDC and CD8⁺ T cells are conserved and important functionally. Specifically, mRegDCs can directly regulate activate and expansion of antigen-specific T cells, where both PD-L1 and OX40L expression by mRegDC are involved.

In terms of heterogeneity among tumour-retained mRegDC, we contend that several aspects of our study yield previously unappreciated findings. For example, differences between mRegDC within the tumour and those that successfully emigrate to the dLN, which our Kaede model allows us to directly compare, have not been previously delineated. CCR7⁺ DCs as a source of other important ligands which are new targets in recent clinical trials, such as PVR/TIGIT or OX40L/OX40, have also not been described to our knowledge. Moreover, we have now further shown that tumour-

retained mRegDC-expressed ligands are functionally important in regulating antigen-specific T cell responses.

In summary, the use of the Kaede convertible reporter was one bright spot, but in the end did not reveal much more than already known from the presence of CCR7+ activated DCs in the TME. The descriptive quantitation of subsets never went to form a functional result. The proposal for an “exhausted” DC phenotype sounds trendy but isn’t supported by any evidence beyond one interpretation of a pseudo-time computation of scRNA gene expression data.

REVIEWERS' COMMENTS

Reviewer #1 (Remarks to the Author):

The authors have answered most of my questions and concerns, except for the following point: On page 7, line 220-227, the authors listed much transcriptomic evidence for the exhaustion profile of mregDC3, but there were also some results for the opposite. As shown in Extended Data Fig.4B, mregDC3 expresses higher levels of the co-stimulatory molecules CD80 and CD86, as well as IL15 and IL15RA, which are critical for CD8+ T-cell survival, compared to mregDC1, and none of these results support the exhausted features. More convincing experiments are needed to demonstrate the exhausted function of mregDC3, such as their capacity for T cell priming and activation.

Reviewer #2 (Remarks to the Author):

The authors have addressed most of the concerns raised. One aspect that is not elucidated yet are the mechanisms that determine whether mregDC stay in the tumor or migrate to dLNs, although the authors have analyzed several possibilities. In summary, the manuscript has some interesting conclusions, showing a functional heterogeneity of mregDCs that stay in the tumor or migrate to the LN and how the exhaustion phenotype of the tumor-resident mregDC is rescued by anti-PD-L1 treatment.

Reviewer #3 (Remarks to the Author):

After reading the extensive revision, it really seems like the title of the revised study should be "mRegDCs are just an aggregate of activated cDC1 and cDC2 that didn't migrate". When the rebuttal is fully digested, this is the essential message of the paper. This is what many in the DC field have been saying as well.

Since my comments on the rebuttal are a bit long, let me start by saying I vote to accept it at Nature Communications. Now on to the commentary.

The authors acknowledge missing a citation to Ardouin and Malissen's 2016 study in their manuscript and have now included it. They also clarify that CCR7+ PD-L1+ PD-L2+ dendritic cell state is known by various names, but they opted for the "mRegDC" anyway. I think this is not a great choice, but I suppose it's the authors prerogative.

I would note that this issue of nomenclature is being hotly discussed, but I am not hung up on it. This choice by the authors they say is to emphasize the regulatory implications of PD-L1 expression in these mature dendritic cells, particularly regarding their role in immune checkpoint therapy in cancer. However, the study's functional evidence that PD-L1 expression by "mRegDC" plays any regulatory role is not very strong.

The new analyses and experiments indicate that both cDC1 and cDC2 can mature into CCR7+ "mRegDCs" with some cells retained in the tumor without migrating. They identify mRegDC_1 as a transitional state leading to terminal states mRegDC_2 and mRegDC_3 within tumor "mRegDCs". These findings are supported by RNA velocity analyses, which validate the trajectory from gene expression data and are presented in Figure 1J and Extended Data Figure 3B. Additionally, the ratio of Kaede-green to Kaede-red mRegDC clusters provides experimental evidence for progression through this maturation trajectory.

To demonstrate that "mRegDCs" originate from conventional DCs (cDC), the authors analyzed cell populations shortly after photoconversion at 5 hours. They hypothesized that most Kaede-green dendritic cells at this early stage would be cDCs, as newly infiltrated cells would not have had sufficient time to transition into the "mRegDC" program. Their findings confirmed this hypothesis: at 5 hours post-photoconversion, the ratio of Kaede-green mRegDC to cDC was approximately 1/100, which then increased to 1/5 at 72 hours post-photoconversion. This data showing

"mRegDCs" emerge following cDC influx, is included in Extended Data Figure 3C.

The authors acknowledge the difficulty in transcriptionally distinguishing tumor regulatory dendritic cells "mRegDC" based on their cDC1 or cDC2 origins, since both upregulation of a common "mRegDC" program and downregulation of specific cDC transcripts during DC maturation make this differentiation problem. scRNA-seq data show mixed lineage trajectories, both cDC1 and cDC2, converging on "mRegDC" transcriptomes. The authors examined surface expressions of XCR1 and CD11b on tumor mRegDCs, finding both XCR1+ and CD11b+ mRegDCs, with XCR1+ being more prevalent in the Kaede-red fraction. This finding indicating "mRegDCs" originate from both cDC1 and cDC2 is included in Extended Data Figure 3D, with additional discussion on cDC to mRegDC maturation and mRegDC ontogeny in the manuscript.

In the proposed maturation trajectory of tumor "mRegDC" mRegDC_1, which represent an early stage, exhibit lower levels of Cd80 and Cd86 compared to the more mature "mRegDC" 2/3, as shown in Extended Data Figure 4B. Tumor-retained Kaede-red mRegDCs show increased surface expression of CD80/86 relative to the newly-formed Kaede-green mRegDCs. This result indicates that high CD80/86 expression characterizes terminal, tumor-retained mRegDCs, while it is lower in the intermediate/transitional state. Additionally, mRegDCs that successfully migrate to the lymph node (LN), originating from early mRegDCs in the trajectory, also show lower levels of CD80/86, differentiating them from the tumor-retained state. This data is included in Figure 2H.

The study presents data showing that activated CCR7+ dendritic cells (DCs)/mRegDCs are heterogeneous and have varying abilities to support lymphocyte responses based on their ligand expression profiles. Due to the challenges of isolating sufficient DCs from tumors for functional assays, the authors used an in vitro co-culture system to investigate the functional importance of individual molecules expressed by these DCs. They acknowledge this system may not fully replicate the in vivo tumor microenvironment (TME). The data suggest that tumor-retained mRegDCs exhibit features of T cell exhaustion including sustained expression of inhibitory molecules and a distinct transcriptional state, although direct assessment of reduced effector function was not possible. Additionally, these "mRegDCs" sustain high PD-L1/PD-L2 expression and acquire a phenotype distinct from successful lymph node (LN) emigrants. To explore the impact of these mRegDCs on cytotoxic T cell function, especially in the context of anti-PD-L1 treatment, the authors developed a co-culture system with bone marrow-derived DCs.

The study describes an experiment where bone marrow-derived dendritic cells (BMDCs) from control or OX40L-deficient mice were cultured with UV-stressed, Ova-expressing MC38 cells to induce a PD-L1+PD-L2+CCR7+ "mReg-BMDC" phenotype. These cells were then isolated and co-cultured with naïve OT-I CD8+ T cells, leading to several key findings: Anti-PD-L1 antibodies enhanced proliferation and Granzyme B expression in OT-I cells, OX40L loss in mRegDCs reduced OT-I cell expansion, IFN γ treatment, but not anti-PD-L1 antibodies, induced an activated state in mRegDCs resembling mRegDC_2, and pre-treatment with IFN γ promoted OT-I T cell expansion, which was reduced in OX40L-knockout conditions. These results are given to support the functional significance of mRegDC-CD8+ T cell ligand-receptor interactions.

In their experiment, the researchers used bone marrow-derived dendritic cells (BMDCs) from both normal and OX40L-deficient mice, cultured with UV-stressed, Ova-expressing MC38 cells to create a specific "mReg-BMDC" type. They found that anti-PD-L1 antibodies increased OT-I CD8+ T cell proliferation and granzyme B expression, while the absence of OX40L in mRegDCs reduced T cell expansion, and IFN γ treatment activated "mRegDCs", resembling the mRegDC_2 state, thereby highlighting the critical role of mRegDCs in modulating T cell responses. However, the real issue here is that while the changes seen appear "statistically significant", they haven't been shown to be physiologically significant or important. Like the A change from 45% to 30 some percent of CD44+ CD25+ OT-1 or a small change in mean number of divisions, it isn't shown if these amount to any difference in vivo.

The authors used deconvolution analysis primarily to identify possible interactions between mRegDCs and other cell types, particularly noting co-localization with CD8+ T cells, which was later confirmed using spatial techniques. They utilized public repositories like TCGA for accessing large sets of human tumor biopsy samples, applying deconvolution (digital cytometry) for estimating cell type abundances from bulk transcriptomics data. xCell was used for the analysis.

However, one might consider the inherent complexities and potential ambiguities in expression signatures, despite the authors' efforts to minimize overlap between closely related cell types and their validation through flow cytometry immunophenotyping. The study claims that their deconvolution analysis did not infer specific functional states of cells but rather observed correlations between mRegDC and CD8 T cells in CRC transcriptomes from TCGA. While they tried to validate these findings using Visium data for human tumors and IF microscopy in murine tumors, PICseq data from human lung tumors, there remains some significant uncertainty about how far this kind of analysis truly reflects the complex cellular interactions and functional states in the tumor microenvironment.

All in all, I think the authors have done an extensive amount of revision and this probably deserves to be published in Nature Communications. I do regret the use of this term, which many of us in the field think is a bit bogus. This feeling holds out even while the authors have done a good job at their revision, because the text of their study, at least of their rebuttal, acknowledges that "mRegDCs" are just a mixture of activated cDCs of both types, and the evidence supporting a "regulatory" (and implied suppressive) role is really not very strong. But the field can sort this out. It wouldn't surprise me if this paper, despite the use of the term "mRegDC", were used as evidence that the term should be abandoned.

So I am in favor of it being accepted and published.

By the way, I saw this paper at Nature Immunology only upon the original submission and I don't recall seeing it in a revised form there. What's that about? Perhaps this extensive revision wasn't reviewed at Nat. Imm., but that seems weird.

Response letter to reviewer #1 (revision round 2) for:
**Time-, tissue- and treatment-associated heterogeneity in
tumour-residing migratory DCs**

Colin YC Lee *et al.*

Reviewer #1:

The authors have answered most of my questions and concerns, except for the following point:

On page 7, line 220-227, the authors listed much transcriptomic evidence for the exhaustion profile of mregDC3, but there were also some results for the opposite. As shown in Extended Data Fig.4B, mregDC3 expresses higher levels of the co-stimulatory molecules CD80 and CD86, as well as IL15 and IL15RA, which are critical for CD8+ T-cell survival, compared to mregDC1, and none of these results support the exhausted features. More convincing experiments are needed to demonstrate the exhausted function of mregDC3, such as their capacity for T cell priming and activation.

We thank the reviewer for their consideration of our revised manuscript. Please note that we have changed the nomenclature ‘mRegDC’ to ‘(activated) CCR7+ DC’ for the manuscript text, following the suggestion of reviewers and colleagues.

We have used the term “exhausted” here because in our data, the CCR7+ “mRegDCs” with prolonged tumour dwell time i) downregulate class-II MHC expression and antigen presentation machinery, ii) lose immunogenic gene expression including pattern recognition receptors and pro-inflammatory cytokines/chemokines, and iii) are associated with a reduced likelihood to migrate to the tumour-dLN. We acknowledge that further functional investigation to this phenomenon would be ideal, and aim for this to be the scope of future work, including comparing the capacity of tumour-retained DCs versus successful LN emigrants for T cell priming and activation.

However, we do not believe that the results highlighted by the reviewer are necessarily contradictory. Indeed, expression of CD80/86 is higher in mRegDC_3, but the importance of this may be limited, considering the downregulation of other immunogenic programmes and high expression of co-inhibitory molecules. For example, terminally exhausted CD8 T cells express higher levels of cytotoxic granzymes than other CD8 T cell subsets (PMID 30778252), but their collective phenotype and transcriptional programmes distinguish them from other T cell subsets and render them less effective at anti-tumour cytotoxicity. Hence, this description is based on a holistic consideration of the full transcriptome rather than individual molecules. Moreover, the ability for IFN γ to induce an activated “mRegDC” state with increased expression of distinct co-stimulatory ligands and inflammatory cytokines suggest that there is an alternative to this “exhausted” state in terminal CCR7+ DCs. We have now added the following to the Discussion to acknowledge the reviewer’s point. “**While tumour-retained CCR7⁺ DCs increase CD80/86 expression, concomitant downregulation of**

immunogenic transcriptional programmes obscures its functional significance, analogous to the increased expression of granzymes by terminally-exhausted CD8⁺ T cells.”

Response letter to reviewer #2 (revision round 2) for:
Time-, tissue- and treatment-associated heterogeneity in
tumour-residing migratory DCs

Colin YC Lee *et al.*

Reviewer #2:

The authors have addressed most of the concerns raised. One aspect that is not elucidated yet are the mechanisms that determine whether mregDC stay in the tumor or migrate to dLNs, although the authors have analyzed several possibilities. In summary, the manuscript has some interesting conclusions, showing a functional heterogeneity of mregDCs that stay in the tumor or migrate to the LN and how the exhaustion phenotype of the tumor-resident mregDC is rescued by anti-PD-L1 treatment.

We thank the reviewer for their consideration of our revised manuscript. Indeed, we have not elucidated the mechanisms that determine mRegDC retention or migration- this was not within the scope of the present study, but several possible mechanisms are considered in the Discussion. Certainly, the reviewer raises an important question and we hope that our work will motivate further investigation towards addressing mechanisms and conditions that underlie successful tumour DC migration. Please also note that we have changed the nomenclature 'mRegDC' to 'activated CCR7+ DC' for the manuscript text, following the suggestion of reviewers and colleagues.

Response letter to reviewer #3 (revision round 2) for:
**Time-, tissue- and treatment-associated heterogeneity in
tumour-residing migratory DCs**
Colin YC Lee *et al.*

(Reviewer 3 wrote a very lengthy response. We have highlighted what we thought were the key parts in yellow.)

Reviewer #3:

After reading the extensive revision, it really seems like the title of the revised study should be “mRegDCs are just an aggregate of activated cDC1 and cDC2 that didn’t migrate”. When the rebuttal is fully digested, this is the essential message of the paper. This is what many in the DC field have been saying as well.

Since my comments on the rebuttal are a bit long, let me start by saying I vote to accept it at Nature Communications. Now on to the commentary.

The authors acknowledge missing a citation to Ardouin and Malissen's 2016 study in their manuscript and have now included it. They also clarify that CCR7+ PD-L1+ PD-L2+ dendritic cell state is known by various names, but they opted for the “mRegDC” anyway. I think this is not a great choice, but I suppose it’s the authors prerogative.

I would note that this issue of nomenclature is being hotly discussed, but I am not hung up on it. This choice by the authors they say is to emphasize the regulatory implications of PD-L1 expression in these mature dendritic cells, particularly regarding their role in immune checkpoint therapy in cancer. However, the study’s functional evidence that PD-L1 expression by “mRegDC” plays any regulatory role is not very strong.

The new analyses and experiments indicate that both cDC1 and cDC2 can mature into CCR7+ “mRegDCs” with some cells retained in the tumor without migrating. They identify mRegDC_1 as a transitional state leading to terminal states mRegDC_2 and mRegDC_3 within tumor “mRegDCs”. These findings are supported by RNA velocity analyses, which validate the trajectory from gene expression data and are presented in Figure 1J and Extended Data Figure 3B. Additionally, the ratio of Kaede-green to Kaede-red mRegDC clusters provides experimental evidence for progression through this maturation trajectory.

To demonstrate that “mRegDCs” originate from conventional DCs (cDC), the authors analyzed cell populations shortly after photoconversion at 5 hours. They hypothesized that most Kaede-green dendritic cells at this early stage would be cDCs, as newly infiltrated cells would not have had sufficient time to transition into the “mRegDC” program. Their findings confirmed this hypothesis: at 5 hours post-photoconversion, the ratio of Kaede-green mRegDC to cDC was approximately 1/100, which then increased to 1/5 at 72 hours post-photoconversion. This data showing “mRegDCs” emerge following cDC influx, is included

in Extended Data Figure 3C.

The authors acknowledge the difficulty in transcriptionally distinguishing tumor regulatory dendritic cells “mRegDC” based on their cDC1 or cDC2 origins, since both upregulation of a common “mRegDC” program and downregulation of specific cDC transcripts during DC maturation make this differentiation problem. scRNA-seq data show mixed lineage trajectories, both cDC1 and cDC2, converging on “mRegDC” transcriptomes. The authors examined surface expressions of XCR1 and CD11b on tumor mRegDCs, finding both XCR1+ and CD11b+ mRegDCs, with XCR1+ being more prevalent in the Kaede-red fraction. This finding indicating “mRegDCs” originate from both cDC1 and cDC2 is included in Extended Data Figure 3D, with additional discussion on cDC to mRegDC maturation and mRegDC ontogeny in the manuscript.

In the proposed maturation trajectory of tumor “mRegDC” mRegDC_1, which represent an early stage, exhibit lower levels of Cd80 and Cd86 compared to the more mature “mRegDC” 2/3, as shown in Extended Data Figure 4B. Tumor-retained Kaede-red mRegDCs show increased surface expression of CD80/86 relative to the newly-formed Kaede-green mRegDCs. This result indicates that high CD80/86 expression characterizes terminal, tumor-retained mRegDCs, while it is lower in the intermediate/transitional state. Additionally, mRegDCs that successfully migrate to the lymph node (LN), originating from early mRegDCs in the trajectory, also show lower levels of CD80/86, differentiating them from the tumor-retained state. This data is included in Figure 2H.

The study presents data showing that activated CCR7+ dendritic cells (DCs)/mRegDCs are heterogeneous and have varying abilities to support lymphocyte responses based on their ligand expression profiles. Due to the challenges of isolating sufficient DCs from tumors for functional assays, the authors used an in vitro co-culture system to investigate the functional importance of individual molecules expressed by these DCs. They acknowledge this system may not fully replicate the in vivo tumor microenvironment (TME). The data suggest that tumor-retained mRegDCs exhibit features of T cell exhaustion including sustained expression of inhibitory molecules and a distinct transcriptional state, although direct assessment of reduced effector function was not possible. Additionally, these “mRegDCs” sustain high PD-L1/PD-L2 expression and acquire a phenotype distinct from successful lymph node (LN) emigrants. To explore the impact of these mRegDCs on cytotoxic T cell function, especially in the context of anti-PD-L1 treatment, the authors developed a co-culture system with bone marrow-derived DCs.

The study describes an experiment where bone marrow-derived dendritic cells (BMDCs) from control or OX40L-deficient mice were cultured with UV-stressed, Ova-expressing MC38 cells to induce a PD-L1+PD-L2+CCR7+ "mReg-BMDC" phenotype. These cells were then isolated and co-cultured with naïve OT-I CD8+ T cells, leading to several key findings: Anti-PD-L1 antibodies enhanced proliferation and Granzyme B expression in OT-I cells, OX40L loss in mRegDCs reduced OT-I cell expansion, IFN γ treatment, but not anti-PD-L1 antibodies, induced an activated state in mRegDCs resembling mRegDC_2, and pre-

treatment with IFN γ promoted OT-I T cell expansion, which was reduced in OX40L-knockout conditions. These results are given to support the functional significance of mRegDC-CD8⁺ T cell ligand-receptor interactions.

In their experiment, the researchers used bone marrow-derived dendritic cells (BMDCs) from both normal and OX40L-deficient mice, cultured with UV-stressed, Ova-expressing MC38 cells to create a specific "mReg-BMDC" type. They found that anti-PD-L1 antibodies increased OT-I CD8⁺ T cell proliferation and granzyme B expression, while the absence of OX40L in mRegDCs reduced T cell expansion, and IFN γ treatment activated "mRegDCs", resembling the mRegDC_2 state, thereby highlighting the critical role of mRegDCs in modulating T cell responses. However, the real issue here is that while the changes seen appear "statistically significant", they haven't been shown to be physiologically significant or important. Like the A change from 45% to 30 some percent of CD44⁺ CD25⁺ OT-1 or a small change in mean number of divisions, it isn't shown if these amount to any difference in vivo.

The authors used deconvolution analysis primarily to identify possible interactions between mRegDCs and other cell types, particularly noting co-localization with CD8⁺ T cells, which was later confirmed using spatial techniques. They utilized public repositories like TCGA for accessing large sets of human tumor biopsy samples, applying deconvolution (digital cytometry) for estimating cell type abundances from bulk transcriptomics data. xCell was used for the analysis. However, one might consider the inherent complexities and potential ambiguities in expression signatures, despite the authors' efforts to minimize overlap between closely related cell types and their validation through flow cytometry immunophenotyping. The study claims that their deconvolution analysis did not infer specific functional states of cells but rather observed correlations between mRegDC and CD8 T cells in CRC transcriptomes from TCGA. While they tried to validate these findings using Visium data for human tumors and IF microscopy in murine tumors, PICseq data from human lung tumors, there remains some significant uncertainty about how far this kind of analysis truly reflects the complex cellular interactions and functional states in the tumor microenvironment.

All in all, I think the authors have done an extensive amount of revision and this probably deserves to be published in Nature Communications. I do regret the use of this term, which many of us in the field think is a bit bogus. This feeling holds out even while the authors have done a good job at their revision, because the text of their study, at least of their rebuttal, acknowledges that "mRegDCs" are just a mixture of activated cDCs of both types, and the evidence supporting a "regulatory" (and implied suppressive) role is really not very strong. But the field can sort this out. It wouldn't surprise me if this paper, despite the use of the term "mRegDC", were used as evidence that the term should be abandoned.

So I am in favor of it being accepted and published.

By the way, I saw this paper at Nature Immunology only upon the original submission and I

don't recall seeing it in a revised form there. What's that about? Perhaps this extensive revision wasn't reviewed at Nat. Imm., but that seems weird.

We thank the reviewer for their extensive consideration and recommendations towards our revised manuscript, particularly their comments on our new attempts to address “mRegDC” ontogeny, and the functional importance of mRegDC-CD8 T cell interactions. We provide more detailed responses to several of the reviewers' comments below.

Re: Title suggestion and mRegDC nomenclature. We thank the reviewer for these suggestions. After much reflection and consultation with several experts in the DC field, including at the recent EMDS'23 meeting, we now better appreciate that the term “mRegDC” may cause misperceptions and have decided that “**activated CCR7+ DCs**” (abbreviated to CCR7_DC in the scRNA-seq data) is more appropriate and accessible for this manuscript, as the reviewer has suggested. Hence, we have edited this nomenclature. We apologise if our previous use of the “mRegDC” term has been a source of confusion. We have also revised the manuscript title to: “Tumour-retained activated CCR7+ dendritic cells are heterogeneous and regulate local anti-tumour cytolytic activity”.

Re: Physiological significance of *in vitro* observations. We acknowledge the limitations of our *in vitro* model, and agree that the ability to regulate extent of antigen-specific CD8 T cell expansion and activation *in vitro* may not directly translate to different outcomes *in vivo*. The limitations of this system have been acknowledged in our Discussion. The reviewer further highlights that there is limited evidence that PD-L1 expression on activated DCs is suppressive in itself, which we too agree with. However, our *in vitro* work suggests that blocking PD-L1 in an isolated CCR7_DC-CD8 T cell system is sufficient to increase T cell proliferation, suggesting that at the very least, PD-L1 expressing CCR7_DC are more suppressive than in the absence of PD-L1. Furthermore, PD-L1 expression by DCs specifically are a key regulator of anti-cancer T cell immunity *in vivo* (PMID: 35122038), and our data show that CCR7_DCs are the highest expressers of PD-L1 by far.

Re: Computational approaches to investigate CCR7_DC-T cell interactions. We appreciate the limitations of the computational methods used here and the reviewer's concerns, which were necessary due to limited access to human cancer tissues. We hope that our study will motivate a growing body of literature on the complex roles of activated CCR7+ DCs in the local tumour environment, specifically in supporting tumour antigen-specific lymphocyte function.